# New geological constraints on the subsurface structure of the 2022 Fano-Pesaro Mw 5.5 earthquake sequence area (Adriatic Sea, Italy) from legacy seismic reflection images and deep well information.

Elham Safarzadeh<sup>1</sup>, Maurizio Ercoli<sup>1,3</sup>, Filippo Carboni<sup>2,3</sup>, Francesco Mirabella<sup>1</sup>, Assel Akimbekova<sup>4</sup>, Massimiliano Rinaldo Barchi<sup>1</sup>

0 Correspondence to: Safarzadeh, E (elham.safarzadeh@dottarandi.unip.it)

15

<sup>&</sup>lt;sup>1</sup> Department of Physics and Geology, University of Perugia, Italy

<sup>&</sup>lt;sup>2</sup> Institute of Earth and Environmental Sciences (Geology), Albert-Ludwigs-University Freiburg, Germany

<sup>&</sup>lt;sup>3</sup>CRUST Member (Centro interUniversitario per l'analisi SismoTettonica Tridimensionale ConApplicazioni Territoriali), Italy

<sup>&</sup>lt;sup>4</sup> Eni Exploration and Production Division, Via Emilia, 1, 20097 San Donato Milanese, Milan, Italy

## **Abstract**

Studying the subsurface geology in offshore areas is a complex task, as it is impossible or very challenging directly accessing any eventual outcrops at the study site. The integration of key seismic reflection and borehole data is therefore fundamental, even if only available as legacy data on paper hard copy and/or characterized by an apparent low quality. However, such data are often the only ones available, and can still provide a high amount of detailed information for building a reliable geological model to be compared with and discussed about the seismicity distribution in active areas. In this work, legacy seismic reflection profiles calibrated with boreholes are used to propose a new geological model of the frontal part of the Northern Apennines area struck by the 2022 Fano-Pesaro Mw 5.5 earthquake sequence (Adriatic Sea, Italy). The legacy seismic data were digitized and converted to SEG-Y format, and a basic post-stack filtering was applied to enhance data quality. The observed tectonic structures originate from multiple décollements located at different depths and show a strong relationship between the faulting depth and the wavelength of the anticlines. Two structures, namely the Pesaro and the Cornelia anticlines, are interpreted as being related to deep-seated thrusts, showing an en-echelon arrangement and thin-skinned deformation. A smaller wavelength structure, namely the Tamara antiform, is interpreted to be associated with shallow-seated imbricated foreverging thrusts in the forelimb of the Pesaro anticline. We highlight the importance of constructing a well-constrained geological model by integrating legacy geological and geophysical data, aimed at studying offshore seismotectonic settings.

## 40 1. Introduction



Buried and blind thrust faults, particularly those beneath the seafloor, pose considerable challenges for determining their seismic potential and understanding the association between seismic activity and geological structures (Berberian, 1995; Roering et al., 1997; Gunderson et al., 2013; Panara et al., 2021). Despite their hidden nature, they are capable of producing strong earthquakes (≥ Mw 6.0; United States Geological Survey, n.d.) and triggering underwater landslides and tsunamis (Lettis et al., 1997; Ioualalen et al., 2017; Takashimizu et al., 2020; Maramai et al., 2022). As coastal populations and infrastructure continue to expand, understanding the behaviour of these offshore buried faults becomes essential for mitigating both seismic and tsunami risks. Their detection is especially challenging, as it heavily relies on indirect observations such as geophysical data (Roering et al., 1997; Déverchère et al., 2005; Hayes et al., 2010; Sorlien et al., 2013; Franklin et al., 2019). Seismic reflection is one of the most effective geophysical tools able to provide detailed images of the subsurface, illuminating depths where the upper crust earthquakes are located. These data are suitable for identifying faults' geometry, kinematics, hierarchy and dynamics as well as the overall subsurface geological setting and the position of different lithologies (e.g. Barchi et al., 2021).

The Adriatic Sea in central Italy (Fig. 1) is a clear challenging example in terms of risk assessment, as the nearby coastlines are densely populated and many critical infrastructures have been developed over the last decades. In this region, the buried and blind thrust faults, located offshore play a key role in the regional seismotectonic setting, but their detection is particularly

challenging due to the high sedimentation rate of the area (Ricci Lucchi, 1986; Frignani and Langone, 1991; Barbieri et al., 2007; Ghielmi et al., 2013; Amadori et al., 2020) and the generally low quality of the available geophysical data, which are often legacy seismic reflection profiles.

While the axial zone of the Northern Apennines, located about 70 km onshore to the west, is affected by extensional seismicity (Lavecchia et al., 1994; Ciaccio et al., 2005; Chiaraluce et al., 2017; Porreca et al., 2018; Barchi et al., 2021; Sugan et al., 2023), the seismic events recorded in the offshore Marche region (Central Italy, along the Adriatic Sea coast) are mainly compressive and are caused by buried active thrust faults (Argnani, 1998; Maesano et al., 2013; Brancolini et al., 2019; Panara et al., 2021; Montone & Mariucci., 2023; Maesano et al., 2023; Pezzo et al., 2023, Lavecchia et al., 2023). This active contraction is testified by historical seismicity (Boschi et al., 2000; Guidoboni et al., 2019; Rovida et al., 2022), as well as numerous observations derived from geodetic (Bigi et al., 1992; D'agostino et al., 2008; Palano et al., 2020; Pezzo et al., 2020), geological, geophysical (Finetti & Del Ben., 2005; Fantoni & Franciosi, 2010; Ghielmi et al., 2010; Tinterri & Lipparini, 2013; Casero and Bigi, 2013) and seismotectonic studies (Di Bucci and Mazzoli, 2002; Maesano et al., 2013; Brancolini et al., 2019; Panara et al., 2021; Montone & Mariucci, 2023; Carboni et al., 2024).

The subsurface offshore thrust faults and related folds in the study area (Fig. 1) are part of the latest contractional structures associated with the evolution of the Northern Apennines thrust belt. The contractional structures possess a similar geometry to that of the outcropping westward structures, where the chain is exposed (e.g. Mazzanti and Trevisan, 1978; Alvarez, 1999; Barchi, 2010). In the Northern Apennines in particular, previous studies suggested that at least two main sets of structures coexist, namely the Umbria-Marche folds ("deep-seated - large-structures") and shallow imbricates ("shallow-seated - small-structures") (multiple décollements model - Massoli et al., 2006). These two sets of structures have different characteristics and significance. Weak décollements, located at different depths, influence the geometry and kinematics of the thrust systems. Such décollements largely govern the thrust dimensions and evolution, so that the deeper the décollement, the larger the wavelength of the structure (Barchi et al., 1998; Barchi et al., 2010). These considerations are supported by both field observations (e.g., Koopman, 1983; De Feyter, 1986) and former seismic interpretation works in the same region (Pieri and Groppi, 1981; Castellarin et al., 1985; Bally et al., 1986; Barchi et al., 1998; Pauselli et al., 2002) as well as in further areas of the Central Adriatic Sea. (e.g., Carboni et al., 2024).


Understanding the subsurface geological setting in a seismically active area is essential not only for identifying the active causative fault segment, but also for determining the lithologies involved in seismic faulting (e.g. Mirabella et al., 2008). In addition, the spatial distribution of these subsurface geological units also affects the configuration of the seismic velocity models, which are critical for achieving more accurate earthquake location solutions (Latorre et al., 2016).

This study focuses on the recent Fano-Pesaro earthquake sequence that occurred in the southern portion of the Northern Adriatic (NA) Sea, about 25 km offshore from the coastal towns of Fano and Pesaro (Fig. 1). The earthquake caused damage along the entire coast of the Marche Region. This area has experienced significant seismic activity since November 2022, culminating with a Mw 5.5 earthquake on 9 of November 2022. The focal mechanism of the main earthquake indicates almost pure thrust-slip motion along a NW-SE striking fault. No moment tensor solution has been computed for the Mw 5.2 event

due to phase overlap and interference from the two events (Pezzo et al., 2023). By the end of December 2024, this earthquake sequence had recorded over 560 aftershocks larger than Mw 2 (http://terremoti.ingv.it).



In this study, an extensive investigation across an area of about 1400 km2 of the Adriatic Sea offshore Pesaro and Ancona towns has been carried out. A comprehensive data analysis has been accomplished across this region, in order to understand and shed light on the geological and structural settings, aiming to provide insights into its tectono-stratigraphic evolution and seismotectonic character. Therefore, stratigraphic and geophysical analysis, as well as extensive seismic interpretation, were carried out on selected wells and legacy reflection seismic profiles, including both unpublished (commercial) and freely available data stored in public databases (https://www.videpi.com). This study aims to demonstrate the importance of a thoughtful re-use and revision of such offshore data. This workflow is essential to shed light onto the subsurface geological settings of the area that can be compared and integrated with seismicity, particularly because no surface outcrops are clearly available, and there are well-known uncertainties characterizing the offshore earthquakes' locations. The joint use of seismic reflection profiles, calibrated with borehole stratigraphy, provides the necessary framework to mitigate these limitations and improve the accuracy of the geological models.

**Fig.1.** Seismotectonic framework of the Northern Adriatic Sea. Red dots indicate recorded seismicity from 9<sup>th</sup> November 2022 (the Fano-Pesaro earthquake) until 1<sup>th</sup> of January 2025, including events with magnitudes greater than Mw 1.7 (959 events). The orange and yellow stars indicate the main shocks of the 9<sup>th</sup> November 2022 earthquake events, provided by INGV. Blue diamonds indicate seismicity of the region derived from both instrumental and non-instrumental archived earthquakes from years 1269 to 2019, obtained from CPTI15-DBMI15v.4.0 (Rovida et al., 2022 and Locati et al., 2022). The focal mechanisms are from INGV (2022 eq and 2013 eq), Vannoli et al. 2015 (1930 eq). The seismogenic sources are from DISS 3.3.0 (DISS Working Group, 2021), and the fault traces and Fault names (TTS, CTS, PTS and ETS) are from Maesano et al. (2023). The bathymetric contours are based on data from the EMODnet Bathymetry Consortium (2020). The seismic reflection profiles include public tracks from ViDEPI (light gray), representative lines S3 and S4 (green) from ViDEPI, as well as lines S1, S2, and S5 (blue) from ENI.

## 2. Geological, Structural settings and regional seismicity.







The NA Sea is predominantly composed of continental crust (Ollier & Pain., 2009; Piccardi et al., 2011) and represents the deformed foreland of the surrounding orogenic belts, including the Apenninic belt to the West, the Dinarides-Albanides to the East, and the southern Alps to the North (Fig.1). The Adriatic Sea is composed of different stratigraphic units registering the initial drowning and the subsequent emersion of the Tethys margin (e.g., Finetti & Del Ben., 2005; Casero and Bigi, 2013). The initial rifting phase led to the deposition of Permian-Anisian sandstones interbedded with dolostones, limestones, gypsum and salt. During the Late Triassic, the normal faults accommodating the initial Tethys rifting allowed the deposition of evaporitic deposits and shallow-water carbonate sequences (Mattavelli et al., 1991; Geletti et al., 2008; Carminati et al., 2013; Wrigley et al., 2015). The further sea opening promoted the growth of extensive carbonate platforms during the Lower Jurassic, which were subsequently buried by the deposition of Lower Jurassic-Palaeocene intraplatform pelagic carbonate succession (e.g., Centamore et al., 1992; Menichetti and Coccioni, 2013). The closure of Tethys marked the beginning of the compressional phase, which led to the formation of the Alps since the Cretaceous (e.g., Dewey et al., 1989; Schmid et al., 2004; Stampfli & Borel, 2002; Handy et al., 2015), the Dinarides-Albanides since the Palaeocene-Eocene (e.g., Ustaszewski et al., 2010; van Unen et al., 2019; Schmid et al., 2020; van Hinsbergen et al., 2020), and the Northern Apennines since the Oligocene (e.g., Molli, 2008; Molli and Malavieille, 2011; Barchi, 2010; Caricchi et al., 2014; Carboni et al., 2020 a, b). The migration of both the Dinarides and Apennines towards the central axis of the Adriatic Sea (Channell et al., 1979), led to the deposition of upper Eocene-Quaternary sequences on their common foreland basin.

The stratigraphic succession includes a Mesozoic-Paleogene, pre-orogenic, passive margin succession, deposited on the southern side of Western Tethys, and a Neogene-Quaternary, syn-orogenic succession, deposited on the flexured foreland of the Northern Apennine. A reference stratigraphic column is shown in Figure 2, illustrating the main units derived from Pesaro Mare 04 and W1 boreholes drilled in the study area (Fig.1).

The uppermost unit includes up to ~ 3200 meters of Pliocene–Quaternary foreland turbiditic clastic sediments, ranging from Upper/Lower Neritic to Pelagic Platform environments, and includes the Argille del Santerno (AS) and Porto Garibaldi (PG) formations. These sediments transgressively overlie a relatively thin Miocene Marly Group succession, deposited in the distal part of the foreland. This succession includes formations of the Messinian Gessoso Solfifera (GS) (relatively thin), Schlier (SCH) and Bisciaro (BIS) formations. The pre-orogenic multilayer, spanning from the Late Triassic to the Early Miocene, lies beneath the overlying successions. This interval consists of Meso-Cenozoic carbonate deposits alternating between platform and slope facies, indicative of deposition in Lower to Middle Neritic and Pelagic Platform settings. Key formations include the Upper Jurassic to Oligocene Scaglia (SCA), Marne a Fucoidi (FUC), Calcari di Cupello (CDC), and Calcari Diasprigni (CDU), as well as Lower Jurassic dolostones, such as the Calcare Massiccio (MAS) and Dolomie di Castelmanfrino (DCM). Compared to the Umbria-Marche Basin, this succession shows significant differences, notably the interlayering of platform facies with pelagic deposits in the Late Jurassic to Early Tertiary interval. The Triassic succession of the Anidriti di Burano Formation (BF) consists of alternating dolostones, anhydrites, halite, and gypsum, and acts as regional décollement horizons

(Casero and Bigi, 2013). Beneath this succession, the pre-Mesozoic crystalline basement of the Adriatic microplate forms the foundational framework (Vannoli et al., 2014). Due to the limited availability of deep wells, direct data on the thickness and depth of these deeper units remain sparse, necessitating reliance on seismic interpretation.

**Fig. 2.** Reference stratigraphic column for the Pesaro-Fano offshore area (from the Late Jurassic to Holocene sedimentary succession), derived from two representative boreholes (Pesaro Mare 04 and W1, location in Fig. 1).

The NA is characterized by a high sedimentation rate that evolved throughout the Pliocene and Pleistocene, reflecting changes in the depositional environment and regional subsidence. During the Pliocene (5.33–2.58 Ma), sedimentation rates were estimated at 1-2 mm/year in both the Po Plain area and the NA Sea (Ghielmi et al., 2010, 2013; Amadori et al., 2020; Maesano et al., 2023). In the Po Plain area, these rates increased to over 2.5 mm/year during the Calabrian stage (1.8-0.78 Ma) with measured values ranging from  $2.83 \pm 0.19$  mm/year to  $2.14 \pm 0.21$  mm/year (Maesano & D'Ambrogi, 2016). However, sedimentation rates progressively decreased throughout the Middle (0.78-0.126 Ma) and Upper Pleistocene (0.126-0.0117 Ma), reaching a minimum of  $0.39 \pm 0.05$  mm/year in the last 0.45 Myr. This decrease reflects the transition to continental deposition and a general reduction of accommodation space in the basin, while also recording the effect of ongoing regional subsidence during the Pleistocene (Maesano & D'Ambrogi, 2016). The high sedimentation rate and the absence of clear seafloor deformation found on bathymetric and seismic reflection data (Di Bucci & Mazzoli, 2002), along with the generally low-to-moderate magnitude of instrumental seismicity (Mw 

## 3. Fano-Pesaro earthquake: State of the Art







The Fano-Pesaro earthquake sequence began on November 9, 2022, with a Mw 5.5 mainshock. One minute later, a Mw 5.2 earthquake occurred approximately 8 km to the south-southeast of the mainshock. Before this, only one smaller event (ML 2.8) was recorded roughly two months before the mainshock, and no foreshocks immediately preceded the sequence. This abrupt activation caused notable damage along the central Adriatic coastline, drawing significant attention to the area's complex tectonic structures.

Most authors identify that the Adriatic domain is mainly governed by compressive tectonics, with thrust-related deformation playing a dominant role (e.g., Pauselli et al., 2006; Maesano et al., 2013; 2023; Sani et al., 2016; Lavecchia et al., 2023), although others suggest the region is primarily affected by active strike-slip tectonics, with minor thrusts that are occasionally reactivated (e.g., Di Bucci and Mazzoli, 2002; Mazzoli et al., 2015).

Since the Fano-Pesaro 2022 earthquake sequence, several studies have been conducted to better map the regional structures and identify the possible seismogenic faults. These studies have employed various hypotheses and scientific approaches, as well as seismicity relocation, to achieve this goal (Maesano et al., 2023; Pezzo et al., 2023; Lavecchia et al., 2023; Pandolfi et al., 2024; An et al., 2024).

Maesano et al. (2023) were among the first that perform a review and reinterpretation of public seismic reflection profiles (CROP and ViDEPI profiles), alongside comparisons with earthquake locations and aftershock distributions from INGV. These authors suggested that the Fano-Pesaro Offshore earthquake sequence took place along a Cornelia Thrust System (CTS), a buried thrust structure situated at the edge of the Northern Apennines. The affected segment covers an area of approximately 25–40 km² within the larger CTS fault, which itself extends roughly 28 km in length and 10–15 km in width. The CTS is estimated to have a total fault surface area of about 300 km², making it capable of generating earthquakes up to magnitude 6.5. Shortly after, Pezzo et al. (2023) characterized the seismic sequence in space and time, using data from the INGV monitoring system, GNSS-constrained coseismic slip, and public seismic reflection profiles (ViDEPI). Their interpretation identified shallow buried anticlines in the upper 5-6 km of the crust with ramps dipping 20°–35° extending from a deeper, regional basal décollement with a westward dip of 1°–7°. Based on the distribution of relocated aftershock events, the authors interpreted a 15 km long striking seismogenic fault patch, dipping 24° SSW and seismically active at depths of 5–10 km. Using the HypoDD relocation method, they refined the mainshock's position, revealing it to be 4.4 km farther south and at a deeper depth of 8 km than previously reported in the INGV catalogue.

Lavecchia et al. (2023) expanded this picture by investigating the broader lithospheric scale deformation (De Nardis et al. 2022), analyzing the multi-scale geometries of slowly deforming continental regions (SDCR) in eastern Central Italy. They suggested the presence of a shallow megathrust (T1,  $\sim$  20 km to a few km deep) which represents the basal detachment of the external fold-and-thrust domain of the Adriatic Arc. These authors propose the T1 splay, named Bice thrust, extending  $\sim$  30 km with a listric geometry (dip angle  $\sim$  40°–20°, seismogenic depths  $\sim$  7–11 km) and converging at depth with the Cornelia Thrust. Upon associating the first mainshock (Mw 5.5) with the central and southern part of the Bice thrust, they interpret the

second event (Mw 5.2) as due to the subordinate activation of the northern part of the Cornelia Thrust. Following this study,

Pandolfi et al (2024) conducted a probabilistic seismic hazard analysis for the Adriatic Thrust Zone (ATZ).

More recently, An et al. (2024) proposed a new workflow to relocate the Fano-Pesaro seismicity, revealing sharper earthquake clusters between 2–12 km depth. Their analysis estimated an average fault dip of approximately 30° towards the south-southwest. In comparison to the results available in the INGV catalogue, they presented a sharper earthquake cluster closer to the shoreline, mapping a geometry coherent with the available focal mechanisms as well as with the horizons interpreted in seismic reflection profiles.

Finally, Costanzo et al. (2024) presented a new catalogue of the 2022-2023 Adriatic Offshore Seismic Sequence obtained through machine learning-based processing. His relocation placed the ML 5.5 mainshock approximately 0.5 km above the Cornelia fault. compared to the INGV catalogue, this event was shifted 0.44 km southward and 1.2 km deeper. Similarly, the ML 5.2 event was relocated approximately 0.6 km deeper and slightly northwest of its position in the INGV catalogue.

The different interpretations show that the seismogenic structure is not clearly understood at the moment. While Maesano et al. (2023) and Lavecchia et al. (2023) propose new models that differ in identifying the causative faults, other studies (Pezzo et al., 2023; An et al., 2024 and Costanzo, 2024) focused on seismotectonic analysis and the relocation of seismicity, correlating the events with existing models. Despite differences in approaches, results and interpretations on thrust geometries, dimensions, depths and structural relationships, all of the above-mentioned studies agree that the 2022 earthquakes are related to an average ~ 30° dip, southwest-dipping thrust fault, located in the frontal part of the Northern Apennines. However, different opinions remain about which thrust could be the causative structure for the recently recorded seismicity.

## 4. Data and methods





The findings outlined in this paper are based on the interpretation of four deep wells (Table 1) and a set of seismic reflection profiles, comprising 8 crosslines and 3 tielines, covering an area of approximately 1400 km², five of which are described and discussed in detail. Initially, no digital data (e.g. SEG-Y files) were available to be used for enhancing the quality of the dataset. Instead, all the seismic reflection profiles were provided as digital images, scanned from hard paper copy, in PDF format. Three of the selected seismic reflection profiles and a key-borehole, kindly provided by the Italian Energy company Eni S.p.A. under a confidential agreement, are unpublished. The other boreholes and seismic reflection profiles were retrieved from publicly available datasets from ViDEPI databases (https://www.videpi.com; https://www.crop.cnr.it) (Figs. 1 and 2, Table 1), along with industrial exploration reports and maps, which have been deeply reviewed. In this study, these scanned images were digitized to generate SEG-Y files, which were then slightly reprocessed to improve their interpretability (e.g., Barchi et al., 2021; Ercoli et al., 2023; Carboni et al., 2024).

A workflow, including different steps to gather and analyse all the data and ancillary information, has been set up:

1. Data preparation: data organization, quality control (QC), digitalization, georeferencing and importing into a geoscience multi-discipline integration software. 2D and 3D visualization of seismic reflection profiles, wells stratigraphy (formation tops), log images, and seismicity. This workflow incorporates several specialized software

tools: e.g. QGIS for managing geospatial data, a MATLAB code (Sopher, 2018) for digitizing seismic profiles, Petrel and Move platforms for seismic interpretation and velocity modelling, and OpendTect software for conventional data processing. Further details on the processing workflow are illustrated in Figures S1 and S2 of the supplementary materials.

- 2. *Data integration*: stratigraphic correlation among the wells' tops and logs to identify a local seismic stratigraphy, well-to-seismic tie analysis and seismo-stratigraphic interpretation.
- 3. *Velocity model building*: a key well sonic log (Table 1, Fig.3c) was used to extract velocities for Pleistocene and Pliocene formations, whilst literature velocities (Bally et al., 1986; Maesano et al., 2013, 2023; Montone & Mariucci, 2023) were adopted for deeper layers (older than Late Miocene).
- 4. *Time to depth conversion*: horizons, faults and surfaces were converted to depth, and the correlations were extended and verified across a broader area.

**Table 1.** List of datasets (Sp= Spontaneous Potential, Res= Resistivity, Sn = Sonic). The star\* marks the unpublished data, obtained under a confidential agreement, the hashtag# reports the public data downloaded from the Italian database ViDEPI.

| Seismic profiles         |                                |                |                                       |     | Wells                                                       |                                                                                |         |  |
|--------------------------|--------------------------------|----------------|---------------------------------------|-----|-------------------------------------------------------------|--------------------------------------------------------------------------------|---------|--|
| Туре                     | Name                           | Length<br>(Km) | Notes                                 |     | Name                                                        | Depth                                                                          | Logs    |  |
| Crossline<br>(NE-<br>SW) | S1*                            | 18             | Intersected by W1<br>well             | W1* | 3571*                                                       | 4300 m                                                                         | G D     |  |
|                          | S2*                            | 11.5           | Adjacent to the main shock (134 m)    |     | Reached the Lower Cretaceous (Calcari Di Cupello (CDC) Fm). | Sp, Res                                                                        |         |  |
|                          |                                |                |                                       |     | Tamara 01#                                                  | 3191 m<br>Reached the Lower Miocene                                            | Sn, Sp, |  |
|                          | S3 <sup>#</sup><br>(B-402)     | 30             | /                                     |     | Tunkiu 01                                                   | (SCH Fm)                                                                       | Res     |  |
|                          | S4 <sup>#</sup><br>(SV-167-13) | 21             | Intersected by Cornelia well          |     |                                                             | 4258 m<br>Reached the Lower Jurassic<br>Dolostone<br>(MAS Fm).                 | Sp, Res |  |
| Tie line<br>(NW-<br>SE)  | S5*                            | 22             | Adjacent to<br>Pesaro Mare 04<br>well | -   | Pesaro Mare 04#                                             |                                                                                |         |  |
|                          |                                |                |                                       |     | Cornelia 01#                                                | 3976 m Reached the Lower Jurassic Dolostone with Chert (Non defined ~ MAS Fm). | Sp, Res |  |

## 5. Results



## 5.1. Wells' stratigraphy

The wells' stratigraphy was digitized and analysed to identify common geological characteristics (e.g., stratigraphy, lithology, discontinuities, petrophysical properties derived from the logs) and trends (formation thickness, spatial continuity) among the

wells. After reviewing and correlating the lithological and structural information among all the data, a reinterpretation of the wells' stratigraphy has been accomplished and displayed in Figure 3. In the latter, the analysed wells are displayed sequentially, moving from the northwest to the southeast of the study area (Table 1, red arrow in Fig.3a). The data are summarized, aiming to clearly show the tectono-stratigraphic correlation among the four wells, highlighting the spatial variation and gaps due to the presence of erosional and tectonic discontinuities (Fig.3b). To support a deeper understanding of subsurface geology within the study area, such well information was spatially extrapolated along the available seismic reflection profiles, by correlating them with the interpreted TWT (Two-Way Travel Time) seismic horizons ("well- to-seismic tie", Bianco, 2014) and fault sets. From a lithostratigraphic standpoint, five major tectono-stratigraphic units were identified across four wells (Fig. 3):




AS unit (Holocene–Upper Pleistocene): A siliciclastic marine turbidite system composed of fine sandstones, shaly sandstones, and interbedding of shale and silty shale.

PG unit (Lower Pleistocene–Pliocene): This unit is separated from AS by a top-lap unconformity, dated Top Gelasian (Fig. 4), referring to early Pleistocene, older than 1.8 Ma. The Gelasian turbidites within the upper part of PG consist of silty shales with interbedded shales at the top, transitioning to fine sandstones and shaly sandstones in the lower part.

Messinian Marly group (GS-SCH-BIS; Upper Miocene): Comprising shales and marls interbedded with siltstones, carbonates, and minor gypsum deposits associated with the Messinian salinity crisis. The top of this unit is defined by a major unconformity marking the top Messinian surface interpreted as a subaerial exposure surface linked to tectonic uplift or sea-level drop.

SCA group (Oligocene–Upper Cretaceous): Made up of marly limestones interbedded with clays and cherts, attributed to the Scaglia succession. The unit thickens southeastward, consistent with deposition in a flexurally subsiding foredeep setting.

Carbonate platform units (FUC–MAS–DCM; Cretaceous–Jurassic): These include thick intervals of massive limestones and dolostones with variable chert and marl content. Well-by-well stratigraphic characterization reveals distinct stratigraphic and structural features across the study area:

The W1 well intersects the easternmost segment of the seismic profile S1 and contains 160 m of Lower Cretaceous carbonates. Within this well, three erosional unconformities have been identified, corresponding to the top Messinian (between the PG and GS units) at a depth of 3151 m, the lower Oligocene (within the SCA unit) at a depth of 4070 m and the Lower Cretaceous top (between the SCA and FUC units) at a depth of 4154 m (Fig.3).

The Tamara 01 well is located approximately 600 m southeast of seismic profile S2 and near the epicentre of the MW 5.5 mainshock of the 9<sup>th</sup> November 2022. It provides valuable sonic log data for deriving interval velocities and conducting well-to-seismic tie analysis. Projected orthogonally onto the eastern segment of the S1 and S2 seismic profiles, the Tamara 01 well penetrates the upper Miocene SCH Formation for about 176 m. The well exhibits four erosional unconformities and two tectonic boundaries. Erosional unconformities have been identified at several stratigraphic levels: within Lower Pleistocene (between the As and PG units) at a depth of 1217 m, top of the Upper Pliocene (within the PG unit), At 1912 meters, marking

both the top of the Upper Pliocene PG unit and the base of the Lower Pliocene of another PG unit. And the last erosional unconformity occurs top of the Upper Messinian (between GS and SCH units) at a depth of 3015 m. The two tectonic boundaries are recognized from the repetition of the Miocene-Pliocene sequences at depths of 1743 and 2345 m, respectively (Fig.3).


The well Pesaro Mare 04, situated approximately 1 km southwest of the S3 profile, was projected orthogonally onto it. The well penetrates the sequence down to the Lower Jurassic, encompassing 1729 m of dolomitized MAS. Notably, an erosional unconformity corresponding to the top of the Miocene (between SCH and AS units) is documented in the well stratigraphy at a depth of 372 m (Fig.3).

The Cornelia 01 well, located in the southeastern part of our study area, intersects the seismic profile S5. It penetrates Jurassic dolomitized carbonates, which are originally referred to as an *undefined* formation based on the lithological variability and uncertainties in the reported depositional environment. However, considering their stratigraphic position beneath the Marne a Fucoidi and Scaglia Calcarea formations, and their overall characteristics as shallow-water platform carbonates, this unit is interpreted in this study as equivalent to the Dolomie di Castelmanfrino (DCM) Formation. This correlation is consistent with 320 similar successions identified in other Apennine sectors, such as the Montagna dei Fiori area, where comparable dolomitized Jurassic sequences have been described and attributed to the DCM (Ronchi et al., 2003; Murgia et al., 2004; Bencini and Martinuzzi, 2012). The Cornelia 01 well exhibits five erosional unconformities corresponding to the tops of the Upper Pliocene (within PG unit) at a depth of 686 m, Lower Pliocene (within PG unit) at a depth of 738 m, Upper Miocene (between GS and PG units) at a depth of 790 m, Upper Cretaceous (Within SCA unit) at a depth of 1833 m, and Lower Cretaceous (between 325 CDU and FUC units) at a depth of 2478 m. Additionally, a tectonic boundary is reported approximately 30 m from the bottom of the well. It is interpreted as a thrust splay, whose offset results in the repetition of the Lower Cretaceous succession. The well was drilled only into the upper part of this repeated interval (Lower Cretaceous succession), and no data are available for the deeper successions (Fig.3).

**Fig. 3.** a) Location map showing the position of the analysed wells. b) Schematic stratigraphic columns of the wells, reinterpreted from the original data in the ViDEPI database and arranged spatially from northwest to southeast (red arrow in Fig. a).

From the global analysis of the four wells' data across the study area (Fig.3), the Pliocene-Quaternary successions (AS and PG units) show a significant thinning from ~ 3100 m thickness in the northwest to 400–700 m in the southeast, as recorded in Pesaro Mare 04 and Cornelia wells, respectively. Within southeastern wells (Pesaro Mare 04 and Cornelia 01), the Pliocene-Pleistocene sedimentary sequence is frequently incomplete. Notably, in the Pesaro Mare 04, situated on a structural high, the Pliocene succession (PG unit) is entirely absent, with a direct transition from Miocene deposits to Quaternary sediments.

Conversely, in the basin areas, such as the W1 well, a more complete sequence spanning the lower to upper Pliocene is preserved. This sequence is characterized by alternating sandy and clayey layers, often interbedded with marly components. This sequence unconformably overlies the Messinian (GS) evaporites, which are identified exclusively in the northwestern (W1) and southeastern (Cornelia 01) wells of the study area. These evaporites are associated with a Messinian paleo-high that persisted as a subaerially exposed feature for the majority of the Pliocene (Report 1508, ViDEPI).

The lithological analysis of the Meso-Cenozoic carbonate successions within the studied wells reveals a carbonate platform that underwent progressive deepening, testified by the combination of detrital and dolomitic limestones, interspersed with frequent cherty nodules and marly intercalations, particularly in the lower sections. The Triassic succession (BF), which typically consists of evaporites and dolostones in the central Apennines (e.g., Umbria-Marche and Sabina Pelagic Basins), is not intercepted by the studied wells. However, its presence is inferred from nearby Alessandra 01 well (See location in Fig.1), located slightly to the east, which represents the deepest borehole drilled in this region, and is almost entirely composed of dolostone facies reported by Bally (1986), Carminati et al. (2013) and Scisciani & Esestime (2017). As the succession transitions into the Middle Jurassic and extends to the Paleogene, the limestones gradually give way to marly layers.. Additionally, clastic intercalations are observed, suggesting sedimentary inputs from the erosion of adjacent structural highs. Notably, the thickness of the SCA Group increases significantly from the northwestern to the southeastern studied wells (Fig.3).

## 5.2. Seismic stratigraphy and time to depth conversion.



By correlating and calibrating the stratigraphy of Tamara 01 and W1 wells with all the available seismic profiles, we have identified five primary seismic units (SUs), bounded by four prominent, easily traceable key-reflections. These units exhibit distinct geophysical signatures, such as variation in the reflection amplitude, period and geometry. The analyzed seismic profiles follow SEG normal polarity, meaning that an increase in acoustic impedance is represented by a peak, while a decrease corresponds to a trough. The SUs are discussed in the following from top to bottom (Fig.4 and details within the Supplementary Table S1).

SU1 corresponds to the Holocene-Upper Pleistocene turbiditic deposits (AS unit). The uppermost part of SU1 is characterised by continuous to semi-discontinuous, horizontal and parallel reflections, with low to high amplitudes. While the lower part displays continuous to semi-continuous, eastward-dipping reflections, with medium to high amplitudes (Fig.4 and

Supplementary Table S1). The total thickness of SU1 gradually increases north-eastwards, ranging from  $\sim 0.2$  s to 1.5 s TWT across the study area (Fig.4). This thickening pattern is consistently observed in all interpreted seismic profiles (Figs. 5 and 6). SU2 corresponds to the lower Pleistocene turbiditic deposits (PG unit) and is separated from SU1 by a toplap unconformity, dated to Top Gelasian (older than 1.8 Ma; Fig. 4). The thickness of this unit gradually increases from  $\sim 0.2$  s in the SW to 0.6 s in the NE. Similar to SU1, this Thickening pattern is consistently observed in all interpreted seismic profiles (Figs 4, 5 and 6). The uppermost part of this unit displays continuous, NE-dipping parallel reflections with medium to high amplitudes. In contrast, the lower part features semi-continuous, parallel, and sub-horizontal reflections (Supplementary Table S1).






The SU3 represents the Pliocene turbidite deposits and is located in the lower part of PG. The unit displays distinct reflection patterns. The uppermost part of it exhibits continuous, horizontal, parallel reflections with high amplitude, while the middle and lower parts of it show discontinuous to semi-continuous, sub-parallel reflections with low to medium amplitudes. The thickness of this unit varies across different sections, ranging from a few ms to 0.4 s (Figs 4, 5, and 6).

The SU4 represents the complex Miocene succession and is observed within the GS, SCH and BIS Fms. This marly group displays continuous, parallel reflections with high amplitude and high dominant frequency in the narrow uppermost part and creates distinct and sharp reflections in the seismic sections. The rest of the unit presents continuous to discontinuous, subparallel reflections with medium to high amplitude (Supplementary Figs S1, S2 and Table S1). This seismic unit progressively deepens from southwest to northeast (Figs 4, 5 and 6).

The SU5 unit represents the Mesozoic-Paleogene carbonate multilayer and corresponds to the SCA, MAS and DCM Fms and represents the deepest units identified in the study area (Figs 4, 5 and 6). Notably, it exhibits a substantial thickness of over 1 s. The reflections within this unit display a discontinuous, sub-parallel pattern with low to medium amplitude and are marked by some continuous, high amplitude and well-recognizable reflections which are related to the top of the SCA and FUC fms (Mirabella et al., 2008; Porreca et al., 2018; Barchi et al., 2021).

For the depth conversion, a velocity model has been built by integrating new interval velocity values derived from the sonic log of the Tamara 01 well (down to the Late Miocene turbidites) with literature velocity data (e.g., Bally et al., 1986; Maesano et al., 2013, 2023; Montone and Mariucci, 2023). Bi-dimensional velocity models were initially built up along each single profile, with a focus on the shallower area (down to the Top SCA). This workflow was then extended across a tri-dimensional workspace, encompassing later variations driven by all the interpreted horizons and fault surfaces, and correlating with the well data from a broader area. Such a velocity model was later refined in its deeper portion (down to the Jurassic carbonate units) and used to carry out the final conversion from the time to the depth for all the selected seismic profiles. Further details on the velocity models are provided in the Supplementary Table S2.

Fig. 4. a) Seismic stratigraphy of the study area (colored lines) calibrated using the Tamara 01 and W1 wells (see Fig. 3 for stratigraphy column abbreviations). Vp indicates the average P-wave seismic velocity. The displayed values represent averages of the original velocities reported in the last column of Supplementary Table S2, which were derived from sonic log interval averages for the younger succession up to and including the SCH unit, and from published sources for the deeper intervals. b) Digitized sonic log from the Tamara 01 well, showing raw slowness ( $\Delta t$ ,  $\mu s/ft$ ) / row velocity (Km/s).

#### **5.3.** Seismic interpretation




To provide an accurate representation of the subsurface geological and structural features within the research region, five seismic profiles have been selected to carry out the seismic interpretation. Their location and details are shown in Figures 5, 6 and Table 1, while the uninterpreted versions can be found in the supplementary material (Figs. S1 and S2). The dataset

includes four SW-NE-oriented "cross-lines" (S1, S2, S3, and S4) and one NW-SE oriented "tie-line" (S5). The SW-NE profiles cross the two major anticlines present in the area, namely the northern Pesaro Anticline (PA) and the southern Cornelia Anticline (CA), developed at the hanging walls of SW-dipping thrusts, named Pesaro thrust (PT) and Cornelia thrust (CT) (Figs 5 and 6).

The whole interpretation of seismic profiles has been realized by using a pseudo-tri-dimensional correlation of key reflectors picked along the single seismic profiles, by tying reflectors picked on intersecting lines with respect to seismic-stratigraphic units obtained from the well-tie analysis. In this section, the description of the seismic profiles is done from northwest to southeast. The profiles are described considering the increasing TWT (s) and their along-line distance (km).

The seismic profile S1 is dominated by the east-verging PA, characterized by a long wavelength of ~ 12 km (0–12 km distance, Fig. 5a). The PA geometry is traceable from ~ 0.2 s down to ~ 2.5 s, and it is particularly evident following the interpreted Top Jurassic to Top Messinian reflectors (blue and pink colours, respectively). To notice that the Top Messinian reflector is not traceable at the culmination of the PA anticline, due to erosion; in addition, a set of minor folds characterizes the PA forelimb (9–12 km distance range). The more internal minor folds,





closer to the crest zone of PA, affect a thicker succession, ranging from the Jurassic to the Pliocene, and are traceable through key reflectors such as the Top Jurassic (blue colour), Top Lower Cretaceous (dark green colour), Top Oligocene (light green colour), and Top Messinian (pink colour). In contrast, the more external folds deform shallower successions, mainly involving the Messinian units and the overlying Pliocene sediments (Fig. 5a, 5b). Further to the northeast, between 12 and ~ 17 km distance, a complex antiformal structure (wavelength ~ 5 km) folds the Plio-Pleistocene unconformity reflector (dark yellow colour; Fig.5a). This antiformal stack involves a set of minor imbricates, with wavelength 

Fig. 5. Interpretation of S1 and S2 seismic profiles. a) S1, the northernmost section in the study area, crosses the left hinge zone of the PA and reveals variations in its structural style. This profile intersects with seismic section S5 at a  $\sim$  6.1 km. The geometry of the PA is evident by using the Top Messinian, Top Oligocene and Top Lower Cretaceous reflectors. The section also shows the shallow-seated TS developed laterally to the South-Eastern termination of the PA. b) S2 shows the enhanced comprehension of the TS's underlying structure. Four imbricated thrust zones of the PA forelimb and the repetition of Messinian- middle- lower Pliocene successions are observable (uninterpreted images provided in supplementary materials, Fig.S1). Insets a1 and b1 show detailed interpretations.  $\lambda$ s= wavelength of the small structure; PT= Pesaro Thrust; PA= Pesaro Anticline; CA= Cornelia Anticline; TS= Tamara Structure; SU= Seismic Stratigraphic Unit.

The seismic profile S4 (Fig. 6b), located at the southernmost extent of the study area, offers valuable insights into the internal structure of the CA and intersects the Cornelia 01 well, providing key stratigraphic correlations. In contrast to S3, the PA is not present in this seismic section. The CA is represented by an asymmetric NE-verging anticline (as also observed in S3), extending from ~0.5 s to ~3.5 s, and is prominently displayed between 3 km and 13 km distance. This anticline is defined by the folded reflectors from the Top Triassic up to the Pliocene-Pleistocene unconformity (purple to vellow colours), situated 485 within the hanging wall of the underlying CT. The latter, like in S3, offsets the Meso-Cenozoic succession up to the Top lower Pliocene reflector (orange colour); however, in this section, located at the crest zone of the CA (Fig.1), the structure exhibits the maximum height compared to S3, which lies in the northwestern hinge zone, resulting in a larger displacement of the Top Lower Pliocene reflectors (from 1.25 to 2.0 s in S4 versus 2.1-2.2 s in S3; Fig. 6a, b) A small synthetic thrust is again observed in the footwall of the CT, which results in the repetition of the Top Oligocene (Top SCA Group) and Top Messinian reflectors 490 over 9 to 14 km, extending to ~2.7 s. In the northeastern part of this section, the interpreted Pliocene to Quaternary deposits (SU1 and SU2), close to the end of the forelimb of the CA (~13 km distance), are thicker than at the similar location in S3, with the top of the Pliocene reflector located at ~2.5 s in S4 versus ~2.1 s in S3. Additionally, S4 reveals minor fore-verging thrusts in both the southwestern and northeastern sectors of the section (Fig. 6b, at distance ranges 0-3 km and 15-20 km, respectively). While the two west-dipping convergent thrusts observable to the southwest of the CA intersect and slightly 495 displace the Messinian until the Plio-Pleistocene unconformity, the minor thrust to the northeast of CA is detached at the top of the Lower Cretaceous (~ 3.5 s to 2.2 s), displacing the overlying sedimentary successions including deposits from the Upper Cretaceous (dark green) to the Lower Pliocene deposits (orange colour, Fig. 6b).

**Fig. 6.** Interpretation of seismic profiles S3, S4 and S5. a) Section S3 crosses the transition zone of PA and CA structures. The geometries of the anticlines are identified by using the key reflectors (See the legend). However, in this section, the main reflectors in the TS are not clearly traceable (the area marked with a question mark). b) Section S4 is the southernmost section that shows the CA. The doubling of the Mesozoic-Paleogene carbonate multilayer is observable in the frontal part of the CA. c) Section S5 is a tie line, crossing the crest zone of the PA (Uninterpreted images provided in supplementary materials, Fig.S2). λl = wavelength of the large structure; other abbreviations are as in Fig. 5.

The seismic profile S5 (Fig. 6c) serves as a tie line, crossing the crest zone of the PA and situated approximately 500 m from the Pesaro Mare 04 well. This profile provides extensive areal coverage (~36 km), and intersects the S1, S3, and S4 seismic profiles. It is essential for understanding the structure of the PA and for conducting a correlation of interpreted horizons among the aforementioned cross-lines. The geometry of the PA is identifiable from ~0.5 seconds to ~2 seconds, being particularly prominent following the reflectors Top Jurassic (blue) and the Top Messinian (pink). The Top Messinian reflector is visible in the northwestern and southeastern hinge zones of the PA but is clearly absent in the axial zone (~4–20 km distance) due to erosion. The central portion of the PA (~12–29 km distance range) exhibits a stack of imbricate thrusts slices between ~1.5 s and ~2.5 s. These slices are characterized by semi parallel, closely spaced reflectors (Fig. 6c). The PA lies in the hanging wall of the PT, and it is significantly uplifted, forming a semi-symmetrical structure. In contrast, the footwall remains relatively undeformed. These three interpreted thrust faults cut across both the Mesozoic and Cenozoic successions. In the northwestern hinge zone of the PA, no clear displacement has been observed and interpreted within the primary reflectors. Moving to the southeast, starting from ~ 16 km along the profile, growth deposition of the Pliocene-Pleistocene succession (SU1 and SU2) becomes increasingly evident (Fig. 6c). The profile highlights the superimposition of the Meso-Cenozoic sedimentary sequence over the Messinian reflection picked on top of the footwall, with clear evidence of duplication.

The described interpretations carried out on the seismic profiles in TWT, have been then converted to depth, by using the integrated velocity model illustrated in figure 4.

## **6. Discussion**

The integration of both a new set of unpublished and publicly available seismic profiles with borehole data allowed us to highlight the presence of deep-seated and shallow-seated tectonic structures, involving different lithologies and detached along different décollements. This structural setting defines the geometry, dimension and segmentation of the main compressional structures, and ultimately their seismotectonic significance. Depth-converted profiles are used to discuss the possible link between the deep-seated tectonic structures and the seismicity of the area, with a focus on the 2022 seismic sequence (Fig.7). Three depth-converted seismic profiles, S1, S3 and S4 have been selected, being the most representative, based on the achieved geological interpretation and with the aim of building a new geological model of the study area (Fig. 7). These profiles cross the main structures perpendicularly to their strike and extend along the study region from the northwest toward the southeast. This orientation allows to observe the structural relationships between Pesaro Anticline (PA) and Cornelia Anticline (CA) and their thrust faults, Pesaro Thrust (PT) and the Cornelia Thrust (CT), providing a clearer view of the vertical and lateral distribution of the involved key stratigraphic units and the tectonic features within the subsurface of the study area.

## 6.1. Multiple décollements and en echelon folds


In the area covered by this research, variations of mechanical anisotropy strongly influenced the structural setting, forming 540 patterns of interconnected structures, detached along multiple décollements at different depths, corresponding to weak stratigraphic layers. Thus, the recognised tectonic structures have been grouped into two main categories: (i) deep-seated thrusts, represented by the innermost PT and the outermost CT (responsible for the formation of the large-wavelength structures PA and CA), which predominantly affect Mesozoic to Paleogene carbonate sequence; and (ii) shallow-seated thrusts, represented by closely spaced, short-wavelength structures of Tamara structure (TS), affect a limited portion of the Upper 545 Cretaceous and younger sequences, including the Oligocene, the Miocene and the overlying turbidite deposits. Toward the front of the TS, these imbricated shallow-seated thrusts impact even shallower and younger sequences, involving only the Miocene and overlying turbidite deposits (Fig. 5a, 5b). The depth converted profiles S1 and S2 provide a clear view of the spatial relationship among the aforementioned structures (Fig. 7a, 7b). PA is characterized by an NW-SE (along-strike) extent of at least ~ 30 km long and is ~ 12 km wide (along-dip, SW-NE direction, see profile S1 in Figs. 5a,7a, 7d). Its wavelength 550 (λ) as defined by Massoli et al. (2006), thus measured between the PA and CA crests, is ~11 km (Figs. 6a, 7b). Section S1 shows PT as relatively flat in the shallow portion, within the  $\sim 7-12$  km distance range, transitioning to a steeper ramp toward the southwest. It is reasonable to image the PT lower décollement lying at around 9 km depth, possibly on top of the acoustic basement (base of the Triassic evaporites or top of the Permo-Triassic sequence; Mirabella et al., 2008; Barchi et al., 2012; Porreca et al., 2018). However, as profile S1 doesn't extend more to the south-western sector, the interpretation of the deepest structures is poorly constrained, thus based on its interpreted trajectory. Section S1 also shows a series of shallow imbricated, 555 fore-verging and back-verging thrusts in the forelimb of the PA, forming TS, characterized by a length of ~20 km, a width of 7 km and a wavelength λs of ~1.1 km (Figs. 5a, 5b, 7a, 7d). All these structures, including both fore-verging and back-verging thrusts, are associated with the upper, shallower semi-flat segment of the PT, which is detached at multiple stratigraphic levels. These detachments range from ~ 5 km depth within the Jurassic succession in the hanging wall of PT, to a sub-parallel décollement within the Top Messinian (marly group), at roughly 3.5 km depth. The fore-verging imbricated thrusts originate 560 at different levels along this segment, ranging from upper Cretaceous (FUC) in the more internal imbricated thrust to the shallower levels within the weak, Upper Miocene marly rocks in the more external imbricates toward the northeast. These thrusts propagate both eastwards and upwards. This process resulted in multiple repetitions and duplications of the Miocene-Pliocene marly sequences. The nearby Tamara borehole further constraints our interpretation by drilling this shallowest décollement close to the base of the top Miocene "Marly Group" (Fig. 7a) and confirms the depth and the repetition of these 565 sequences across at least three slices. Since the Tamara well was drilled on the outermost part of the Tamara antiformal structure, it does not drill the complete series of imbricated thrusts and duplicated sedimentary sequences mapped in S1 (Figs. 5a,5b and 7a).

The overall analysis and observations of the seismic reflection profiles available on the southeasternmost extent of the study area also allowed for the description of the geometrical characteristics of CA, which are analogous to PA. It results in a NW-

SE striking, ~ 20 km long (possibly extending just a few km further toward the SE) and ~ 12 km wide anticline (profile S4 in Figs. 6b, 7b, 7c) with a wavelength  $\lambda_l$  of ~ 11 km (Figs. 6a, 7b). These structural wavelength values,  $\lambda_l$  and  $\lambda_s$ , are larger than those obtained for corresponding structures in the Umbria-Marche area, where the wavelength range from 3.2 to 7.2 km for  $\lambda_l$  and 0.4 to 2.3 km for  $\lambda_s$  (Massoli et al., 2006). These structures are characterized by lower syn-tectonic sedimentation. Conversely, the observed structural wavelength values are smaller than those observed in the Po Plain, where higher syntectonic sedimentation contributes to even larger structural wavelengths, with  $\lambda_l$  ranging from 15.8 to 33 km and  $\lambda_s$  from 4.5 to 8.2 km (Massoli et al., 2006). This observation is confirmed by the relationship described by Massoli et al. (2006), where variations in structural wavelength are linked to both the depth of the active décollement and the thickness of syn-tectonic sediments.







Our comparative analysis of the PA and CA anticlines, and their related deep-seated thrust systems PT and CT, points out some structural similarities and differences. From the analysis of the profiles S3 and S4 (Fig. 7b, 7c), considering both the geometry of the anticlines and the trajectories of the thrusts, a shared deep decollement level can be inferred at approximately 9 km depth, consistent with results reported in nearby areas (e.g., Pauselli et al, 2006; Lavecchia et al., 2004, 2024). Furthermore, evidence indicates that the thrusting style in this area is a thin-skinned type of deformation, aligning with the observed decollement depth and suggesting tectonic processes that control syn-tectonic sedimentation and accommodate deformation within the overlying sedimentary cover, without involving the basement (Fig. 7). Our interpretation demonstrates that, unlike the PT, the CT lacks an upper shallower décollement. Instead, the ramp of the CT terminates blindly at a depth of 2 km within the base of the upper Pliocene turbiditic successions (Fig. 7b, 7c), and only one imbricated fore-verging thrust has been identified in S4. The latter is also constrained by the Cornelia borehole stratigraphy, evidencing a doubling of the Upper Cretaceous carbonate succession (SCA group) over approximately 3 km between ~16–19 km distance in section S3 and about 4 km between ~ 8-12 km in section S4 (Fig. 7b, 7c). Considering the deeper structures involving the carbonates, this study documents the structural transition between two main compressional structures; the PA (internal) and the CA (external) anticlines. In map view (Fig.7d), these structures are linked to a pair of en-echelon, vicariant, coalescent thrusts, the northernmost PT and the southernmost CT. The interpretation of the seismic lines clearly highlighted that the transition from PT and CT occurs through an intermediate region, where both structures are present (Fig. 7d) and can be viewed as adjacent segments of the outermost thrust of the Northern Apennines. Representative examples of coalescent anticlines extensively crop out also in the Umbria-Marche Apennines (Barchi et al., 1998; Scarsella 1941; Lavecchia, 1981; Lavecchia et al., 1988; Lavecchia et al., 2023), and such examples have been described worldwide since Dahlstrom (1970).

Our investigation shows that the shallow-seated TS structure can be traced only in the southeastern termination of the deep-seated PA up to seismic Profile S3, where both PA and TS overlap on the back limb of the CA (Fig. 7b). However, in the southeastern part of our study area, as seen in seismic profile S4, the shallow-seated imbricated fore-verging thrusts and their related antiformal stacks (TS) are not observable (Fig. 7c). Our investigations indicate that the TS represents the deformed wedge of the frontal part of the PA structure, formed within the hanging wall zone of Pesaro, thus it cannot be considered

originated by a single deep-seated structure such as PT or CT and neither a northwest-eastward continuation of the Cornelia thrust.


Fig. 7. Geological sections derived from (Key figure): a) seismic profile S1, b) seismic profile S3, and c) seismic profile S4. The main shock of 5.5 ML and aftershocks are projected perpendicularly to the section within buffers of 5 km. d) Location map of the interpreted anticlines and thrust faults (this work); the seismicity distribution is sourced from terremoti.ingv.it.  $\lambda_1$ : Wavelength of the large structures;  $\lambda_8$ : Wavelength of the small structures. (Key Figure)

In slightly external sectors, with respect to our studied area, evidence of deep thrusts has been reported from the analysis of low-quality public profiles (Adriatic Arc Front, e.g., Bice thrust, Lavecchia et al., 2023). However, the present study suggests that the PT and associated imbricates did not extend more to the North-East. This consideration is also testified by the presence of a complete sedimentary succession (from Cretaceous carbonates to thick Quaternary sequences). Additionally, in the borehole W1 (drilled in the foreland of the PA), no thrust faults are reported and the Top Messinian reflection correlates well with the corresponding identified erosional boundary. Evidence of deeper, external fronts were not found in the reviewed commercial seismic reflection profiles available across this study area, possibly falling besides the data quality at depth or outside the data coverage.

## 6.2. Seismotectonic implications

The mechanical stratigraphy reveals that both the deep-seated PT and CT ramps cut through the brittle carbonate multilayer, from 3 down to 9 km depth. This range coincides with the depths of most of the seismicity recorded during the Fano-Pesaro 2022 sequence (terremoti.ingv.it), suggesting that these thrusts may potentially serve as seismogenic structures (Fig.8a). Both PT and CT are southwest-dipping thrusts, with an interpreted dip angle of 30°–35°, compatible with the mainshock's focal mechanism (with strike 128°, dip 34° and rake 84°, terremoti.ingv.it).

Given their potential seismogenic role, the relationship between earthquake magnitude and subsurface rupture length for both the PT and CT was analysed using the Empirical relationships for the thrust faults (e.g. Wells and Coppersmith,1994 and Leonard, 2014). Fault length directly influences the maximum possible displacement, and consequently, the potential maximum magnitude (Scholz, 2019). According to the findings of this analysis, the estimated sizes of the PT (~360 km²) and CT (~240 km²) suggest that they are capable of generating seismic events with magnitudes of up to Mw 6.8 and Mw 6.5, respectively. The observed fault lengths are substantial enough to account for both recent and historical seismic activity in the region.

However, determining the exact causative faults for the 2022 November 9th earthquakes remains challenging. It is important to highlight the spatial mismatch, in terms of both location and depth distribution, among the literature interpreted faults and the hypocentral records (terremoti.ingv.it; Table 2) as shown in Figure 1 and Figure 8. Comparing the published earthquake locations and relatively shallow depths (~ 5 km) with our interpretation, seismicity is scarcely distributed across the Cornelia region (Fig. 8a, 8d). The first November 9th, 5.5 Mw main shock appears more closely associated with the PT, other than to the CT (Figs 7d, 8a). The second November 9th mainshock and the aftershocks fall in between the area covered by the seismic profiles S1 and S3, in the interpreted transfer area between PT and CT. These events are close to the PT zone and somewhat far from the CT's main area but occurring at greater depth (~ 8 km) in the footwall of the PT (Figs. 1, 7d). However, it is known that both earthquake hypocentres location and the depth of the "not-relocated" seismicity lack in accuracy (±1 km depth error for the mainshock reported in the INGV catalogue), particularly in the offshore. Recently, several authors have relocated the seismicity recorded during this 2022 sequence. Pezzo et al. (2023), An et al. (2024) and Costanzo (2024) used different relocation methods and methodological approaches, and a significant uncertainty in defining seismic event depth compared to

the location is noticeable. The first relocation by Pezzo et al. (2023) shifted the main shock 1.5 km N-NW, increasing its depth to ~8 km, while aftershocks moved slightly NE and farther offshore (Fig.8b, 8d). The second relocation by An et al. (2024) shifted the main shock 5 km southward, thus closer to the shoreline, with a shallower depth, and relocated the aftershock cluster 6 km S-SE (Fig.8c, 8d). The study also reports error estimations, with maximum values ranging from 0.8 to 3.6 km in all three directions. The spatial distribution of the original (INGV catalogue) and relocated aftershock events, in this area, is farther from the CT and more concentrated around the PT and the transfer zone between the PT and CT (Figs. 1, 7d).



Despite most of the considerations introduced above suggest the recent seismicity related more to PT other than CT (Maesano et al., 2023), this seismicity analysis and whole study underscore the complexity of determining whether the PT or CT served as a primary source of the 2022 seismic activity, or eventually a possible deeper thrust as proposed by other authors (e.g., T1 in Lavecchia et al., 2023). However, such a hypothetical deeper fault is not clearly imaged or visible within available vintage seismic reflection profiles, characterized by a lack of clear reflected signals from deeper reflectors, or just by very weak and poorly continuous reflective patterns embedded within a high level of random noise, typical of legacy profiles (Ercoli et al., 2023).

Table 2. Location and depth parameters of the mainshocks for the 9th November 2022 Fano-Pesaro earthquake, as determined by different sources.

| Event                    | Source             | Depth (km) | Latitude       | Longitude      |
|--------------------------|--------------------|------------|----------------|----------------|
|                          | INGV               | 5.0        | 43°58'59" N    | 13°19'26" E    |
| Main Shock (Mw 5.5)      | An et al., 2024    | 4.40       | 43°56'11"N     | 13°20'20" E    |
|                          | Pezzo et al., 2023 | 7.94       | 43°59'41" N    | 13°18'58" E    |
| Second Shock<br>(Mw 5.2) | INGV               | 7.7        | 43°54'47.88" N | 13°20'40.92" E |
| (IVIW 3.2)               | An et al., 2024    | 8.4        | 43°51'36.36" N | 13°20'16.44" E |

Fig.8. The spatial relationship between seismic events and the structural framework identified in this study, together with the depth distribution of seismicity shown through frequency histograms, seismic events are also projected on the latitude—depth and longitude—depth sections. The orange star marks the mainshock (Mw 5.5) and the yellow star the secondary shock (Mw 5.2) on 9 November 2022. a) Spatial distribution of the main shocks and aftershocks recorded by INGV between 1 November 2022 and 31 January 2023 (<a href="https://terremoti.ingv.it/en">https://terremoti.ingv.it/en</a>, accessed on 1 June 2025). b) Relocated seismicity from Pezzo et al. (2023) between 22 November 2022 and 5
January 2023, including the 9 November mainshock. c) Relocated seismicity from An et al., 2024 (200 seismic events) from 9 to 15 November 2022. d) Combined seismicity from INGV and relocated datasets.

## 5. Conclusions






This paper presents a new geological model of the tectonic structures of the Fano-Pesaro offshore area within the frontal part of the Northern Apennines. Multiple decollements located at different depths have been observed in the study area. These structures show a strong relationship between the depth of faulting and the wavelength of the related anticlines, influencing the kinematics of the thrust system. This study suggests the PT and CT thrusts are possibly detached at depths of  $\sim$  9 km on top of the acoustic basement. The two related PA and CA anticlines can be followed along strike for about 50 km and are characterized by a wavelength in the order of  $\sim$  11 km. The TS, a series of imbricate thrusts, develops along the shallow part of the PT at a depth of 3.5 km, is characterized by a short wavelength ( $\sim$  1.1 km) of the imbricates spread along  $\sim$  5 km in the forelimb of PA, and it can be followed for  $\sim$ 20 km along strike. The PT and CT en-echelon arrangement, the presence of multiple detachments and the thin-skinned deformation (multiple décollements) suggest a geological model for this outermost sector of the Apennines, thus characterized by a thrust system not involving the basement (thin-skinned tectonics).

This study highlights the Cornelia thrust system having a limited extent toward the NW. In addition, the spatial distribution of overall seismicity possibly related to the CT is scarce and cannot be easily linked with it. Although based on its geological, structural and geometrical characteristics, the CT thrust system cannot be completely excluded as a seismogenic source. In the present study, the analysis and the integration of the relocated hypocentres together with the new geological insights suggest that the PT, or a possible deeper easternmost structure, would be a better candidate to be associated with the mainshocks. On the other hand, the relay zone between PT and CT is more coherent with the second main event. The still present uncertainty is mainly due to the low accuracy of the seismicity relocation caused by the lack of seismic stations. On the light of all uncertainties related mainly to the inaccuracy of the offshore seismicity relocation and related depth estimation of the seismic events, it is therefore fundamental to provide solid geological constraints by relying on the unique subsurface data (seismic reflection and wells) available as well as onshore analogues outcropping in the central Apennines. This work aims to remark that defining a solid subsurface geological model by integration of these key data sources (even if legacy) is essential in offshore areas. Building up a reliable, geologically driven model allows for refining not only velocity models to use for more accurate earthquakes' relocation, but also for increasing the reliability of seismotectonic studies and risk assessments.

#### Data availability

Supplementary data associated with this article can be found in the online version

## **Author contribution**

ES, ME, and MB contributed to the conception and design of the study. ES performed data curation, analysis, investigation, methodology, visualization, preparation of tables, maps and figures, and wrote the original draft of the manuscript. ME contributed to investigation, writing and revision. FC contributed to writing, revision, visualization maps and figures. FM and

AA contributed to the investigation and revision. MB was responsible for conceptualization, resources, supervision, and review and editing. All authors contributed to the final revision of the manuscript and approved the version submitted.

## 710 Competing interests

The authors declare that they have no conflict of interest

# **Acknowledgment:**





This research was partially funded by the REDI (REducing risk of natural DIsasters) consortium, University of Camerino (Resp. Prof. Massimiliano Rinaldo Barchi), to which we express our sincere gratitude. We would also like to sincerely thank the anonymous reviewers for their constructive comments and suggestions, which helped improve the quality of the manuscript. We are deeply grateful to Professor Ramon Carbonell (Geo3BCN-CSIC, Spain) for his kind support and thoughtful feedback during the revision of the manuscript. We also warmly thank Dr. Daniel Sopher for his guidance and help during the digitizing process of the seismic profiles. We gratefully acknowledge Schlumberger, Cegal, and PE Limited for providing academic licenses to the University of Perugia, together with their support teams, and we thank the the QGIS and Inkscape teams for developing essential open-source software used in this study. We extend our special thanks to ENI for providing subsurface data under a confidential agreement, and to the Ministero dello Sviluppo Economico DGRME, the Società Geologica Italiana, Assomineraria, and CNR ViDEPI (https://www.arcgis.com/) for granting access to public domain data and services. We are also grateful to An and the Pezzo groups for sharing with us their catalogs of relocated seismicity.

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

## 1035 Web references

https://diss.ingv.it/ last access: 20 January 2025.

https://www.terremoti.ingv.it/ last access: 1 January 2025.

https://www.videpi.com/ last access: 20 January 2025.

https://www.usgs.gov/programs/earthquake-hazards/earthquake-magnitude-energy-release-and-shaking-intensity.