# Peer review of "New geological constraints on the subsurface structure of the 2022 Fano-Pesaro Mw 5.5 earthquake sequence area (Adriatic Sea, Italy) from legacy seismic reflection images and deep well information."

_EGUsphere, 2025_

## Referee Comment (RC2)

**Building up a subsurface geological model in active offshore areas: constraints from legacy seismic reflection profiles and deep wells in the 2022 Fano-Pesaro Mw 5.5 earthquake sequence area (Adriatic Sea, Italy).**

Elham Safarzadeh[1], Maurizio Ercoli[1,3], Filippo Carboni [2,3], Francesco Mirabella[1], Assel Akimbekova[4], Massimiliano R. Barchi[1]

[1] Department of Physics and Geology, University of Perugia, Italy

[2] Institute of Earth and Environmental Sciences (Geology), Albert-Ludwigs-University Freiburg, Germany

[3] CRUST Member (Centro interUniversitario per l'analisi SismoTettonica Tridimensionale ConApplicazioni Territoriali),
Italy

[4] Eni Exploration and Production Division, Via Emilia, 1, 20097 San Donato Milanese, Milan, Italy

*Correspondence to*: Safarzadeh, E (elham.safarzadeh@dottarandi.unip.it)

**Abstract.** Building a geological model in offshore areas is a complex task, due to the obvious absence of outcrops and thus
the inaccessibility to the study site. The integration of key seismic reflection and borehole data is therefore fundamental, even if only available as legacy data on paper hard copy and/or characterized by an apparent low quality. However, such data are often the only ones available, and can still provide a high amount of detailed information for building a reliable geological model to compare and discuss with seismicity distribution in active areas. In this work, legacy seismic reflection profiles calibrated with boreholes are used to propose a new geological model of the frontal part of the Northern Apennines
area struck by the 2022 Fano-Pesaro Mw 5.5 earthquake sequence (Adriatic Sea, Italy). The observed tectonic structures are originated by multiple décollements located at different depths and show a strong relationship between the faulting depth and the anticlines wavelength. Two structures, namely Pesaro and Cornelia anticlines, are interpreted as related to deep-seated thrusts, showing an en-echelon arrangement and thin-skinned deformation. A smaller wavelength structure, namely Tamara antiform, is interpreted to be related to shallow-seated imbricated fore-verging thrusts in the forelimb of the Pesaro anticline.
We highlight the importance of constructing a well-constrained geological model by integrating legacy geological and geophysical data, aimed at offshore seismotectonic studies as well as at industrial applications, particularly in the context of energy transition.

**1. Introduction**

Buried and blind thrust faults, particularly those beneath the seafloor, pose considerable difficulties for the study of global
seismic activity (Berberian, 1995; Roering et al., 1997; Gunderson et al., 2013; Panara et al., 2021). Despite their hidden

nature,  capable to produce ==strong earthquakes== and ==related== underwater landslides, and tsunamis (Lettis et al., 1997; Ioualalen et al., 2017; Takashimizu et al., 2020; Maramai et al., 2022). As coastal populations and infrastructure expand, the understanding of the behaviour of these offshore buried faults becomes essential for mitigating both seismic and tsunami risks. Their detection is especially challenging as it heavily relies on indirect observations such as geophysical data (Roering et al., 1997; Déverchère et al., 2005; Hayes et al., 2010; Sorlien et al., 2013; Franklin et al., 2019). Seismic reflection is one of the best geophysical tools able to provide ==high-resolution images== of the subsurface, being capable of illuminating depths where the upper crust earthquakes are located. These data are suitable to identify the faults geometry, kinematics, hierarchy and dynamics as well as the overall subsurface geological setting and position of the different lithological ==bodies which posses different velocity of seismic waves propagation== (e.g. Chiaraluce et al., 2017).

The Adriatic Sea in central Italy (Fig. 1) is a clear challenging example in terms of risk assessment, as the nearby coastlines are densely populated and many critical infrastructures have been developed during the last tens of years. In this region, the buried and blind thrust faults, present in the offshore area, play a key role in the regional seismotectonic setting, but their detection is particularly challenging due to the high sedimentation rate of the area (Ricci Lucchi, 1986; Frignani and Langone, 1991; Barbieri et al., 2007; Ghielmi et al., 2013; Amadori et al., 2020) and the general low-quality of the available geophysical data, frequently being legacy seismic reflection profiles.

While the axial zone of the ==Northern Apennines, located about 70 km onshore to the West, is affected by extensional seismicity== (Lavecchia et al., 1994; Ciaccio et al., 2005; Chiaraluce et al., 2017; Porreca et al., 2018; Barchi et al., 2021; Sugan et al., 2023), the seismic events recorded ==in the offshore Marche region== are mainly compressive, caused by buried active thrusts faults (Argnani, 1998; Maesano et al., 2013; Brancolini et al., 2019; Panara et al., 2021; Montone & Mariucci., 2023; Maesano et al., 2023; Pezzo et al., 2023, Lavecchia et al., 2023). The related active contraction, ==affecting the Periadriatic region== is testified by historical seismicity (Boschi et al., 2000; Guidoboni et al., 2019; Rovida et al., 2022), and by many observations derived by geodetic (Bigi et al., 1992; D'agostino et al., 2008; Palano et al., 2020; Pezzo et al., 2020), geological, geophysical (Finetti & Del Ben., 2005; Fantoni & Franciosi, 2010; Ghielmi et al., 2010; Tinterri & Lipparini, 2013; Casero and Bigi, 2013) and seismotectonic studies (Di Bucci and Mazzoli, 2002; Maesano et al., 2013; Brancolini et al., 2019; Panara et al., 2021; Montone & Mariucci, 2023; Carboni et al., 2024).

The subsurface offshore thrust faults and related folds ==in the study area== are part of the latest contractional structures associated with the evolution of the Northern Apennines thrust belt. The contractional structures possess similar geometry to that of the outcropping westward structures, where the chain is exposed (e.g. Mazzanti and Trevisan, 1978; Alvarez, 1999; Barchi, 2010). In the Northern Apennines in particular, previous work suggested that at last two main sets of structures, namely the Umbria-Marche folds ("deep-seated - large - structures") and shallow imbricates ("shallow-seated - small - structures") coexist (multiple décollements model - Massoli et al., 2006). These two sets of structures have different characteristics and significance. Weak décollements, located at different depths, influence the geometry and kinematics of the thrust systems. Such décollements largely govern the thrusts dimension and evolution, so that the deeper the décollement,

[Figure]

the larger the wavelength of the structure (Barchi et al., 1998; Barchi et al., 2010). These considerations are supported by both field observations (e.g., Koopman, 1983; De Feyter, 1986) and former seismic interpretation works in the same region (Pieri and Groppi, 1981; Castellarin et al., 1985; Bally et al., 1986; Barchi et al., 1998; Pauselli et al., 2002) and further areas in the Central Adriatic Sea. (e.g., Carboni et al., 2024).

**Fig.1. Seismotectonic framework of the Northern Adriatic Sea. Red dots indicate the recorded seismicity from 9th November 2022 until 1th of January 2025, including magnitudes higher than Mw 1.7 (959 events) and the focal mechanism of the main shock (9th November 2022). The orange and yellow stars indicate the main shocks of the 9th November 2022 earthquake events, provided by INGV (INGV). The blue diamond shapes indicate the seismicity of the region derived from both instrumental and non-**

**instrumental archived earthquakes from years 1269 to 2019, obtained from CPTI15-DBMI15v.4.0 (Rovida et al., 2022 and Locati et al., 2022). Seismogenic sources are from DISS 3.3.0 (DISS Working Group, 2021), while the fault traces are from Maesano et al. (2023).**

[Figure]

The subsurface geological setting of a seismically active area is hence crucial not only for the identification of the active causative fault segment, but also to identify the lithologies involved in seismic faulting (e.g. Mirabella et al., 2008). In addition, the position of the subsurface geological bodies also affects the distribution of the associated different velocity blocks which are fundamental in earthquake location studies (Latorre et al., 2016).

This study focuses on the recent seismic sequence occurred in the southern portion of the Northern Adriatic (NA) Sea, about
25 km offshore from the coastal towns of Fano and Pesaro (Fig. 1), which caused damage along the entire coast of the Marche Region. This area experienced significant seismic activity starting from November 2022, culminating with a Mw 5.5 earthquake on the 9th of November 2022. One minute later, a Mw 5.2 earthquake followed the first event, approximately 8 km more to the south-southeast. The focal mechanisms of both earthquakes indicated almost pure thrust-slip motion along a NW-SE striking fault (Pezzo et al., 2023). This earthquake sequence, at the end of December 2024, recorded over 560
aftershocks larger than Mw 2 (http://terremoti.ingv.it).

[revised manuscript text omitted]

The NA is characterized by high sedimentation rate, that in the Po Plain area reached more than 2.5 mm/year during the Calabrian, decreasing down to ~ 0.4 mm/year in the Upper Pleistocene (Maesano & D'Ambrogi, 2016). In the NA Sea, it is estimated in 1–2 mm/year in the Pliocene (Amadori et al., 2020; Ghielmi et al., 2013, Maesano et al., 2023). The high
sedimentation rate, the absence of a clear seafloor deformation found on bathymetric and seismic reflection data (Di Bucci & Mazzoli, 2002), along with the generally low-to-moderate magnitude of instrumental seismicity (Mw < 4.0, before 2012),

have fueled the scientific debate on the recent activity of the external Northern Apennines. Contrary to slightly more internal sectors (e.g. Conero area, Cuffaro et al., 2010), most authors agree that the tectonic deformation in this external area might be hidden by such a fast sedimentation rate. In the NA Sea, the shortening rate is estimated in 1–2 mm/year until the

Calabrian times, although some studies suggest spatial variations and a progressive temporal decrease (Maesano et al., 2015; Gunderson et al., 2018; Amadori et al., 2020, Panara et al., 2021). Within the same area, some authors, using GNSS data from offshore hydrocarbon seabed-anchored platforms, recently calculated a present-day shortening rate, to be about 1.5 mm/year (Palano et al., 2020, Pezzo et al., 2020; 2023). The offshore tectonic deformation characterizing the study area has been imaged by seismic reflection profiles, showing that the tectonic structures are organized in multiple blind thrusts with associated anticlines (Argnani, 1998; Bigi et al., 1992; Fantoni & Franciosi, 2010; Ghielmi et al., 2010; Maesano et al., 2023). Such reverse faults are buried below thick Plio-Pleistocene marine and continental deposits and likely rooted at depth along a common basal décollement (Bally, 1986, Panara, et al., 2021, De Nardis et al., 2022).

The debate about the recent activity of the external Northern Apennine associated to such blind thrusts has been revived during the last ~ 15 years, as a few important earthquake sequences have been recorded before the 2022 sequence (Maesano et al., 2023; Lavecchia et al., 2023): one in the 2012 and a second in the 2013, onshore in the Pianura Padana (northern Italy) and  offshore southern of Ancona in Marche region, respectively (Mazzoli et al., 2015, Maesano et al., 2013; Burrato et al., 2012; Scognamiglio et al., 2012; Tertulliani et al., 2012; Pezzo et al., 2013; Tizzani et al., 2013; Bonini et al., 2014; Nespoli et al., 2018). Additionally, a revision of the historical seismicity extracted from the available seismic catalogues, reports sequences encompassing mainshock events of Mw>5.5, whose epicentres location is mapped either offshore or onshore the coastline (e.g., 30 October 1930 Mw 5.8 at Senigallia (Vannoli et al, 2015), Fig. 1). These earthquakes have been mainly caused by active thrust faults and produced several induced effects as well as victims and extensive damages within the Marche Region (Guidoboni et al., 2019, Rovida et al., 2022 and Locati et al., 2022). All these recent seismic events stimulated recent studies integrating different disciplines, providing new information, evidence and constraints to the active tectonic setting of the outer Northern Apennines

**3. Fano-Pesaro earthquake: State of the Art**

Most authors identify the Adriatic domain being mainly governed by compressive tectonics, with thrust-related deformation playing a dominant role (e.g., Pauselli et al., 2006; Maesano et al., 2013; 2023; Sani et al., 2016; Lavecchia et al., 2023), although others suggest the region is primarily affected by active strike-slip tectonics, with minor thrusts occasionally reactivated (e.g., Di Bucci and Mazzoli, 2002; Mazzoli et al., 2015).

Since the Fano-Pesaro 2022 earthquake sequence, new research has been conducted to map the existing structures and recognize the possible seismogenic faults through several hypotheses and scientific approaches as well as improving the accuracy of seismicity relocation (Maesano et al., 2023; Pezzo et al., 2023; Lavecchia et al., 2023; Pandolfi et al., 2024; An et al., 2024).

[Figure]

Maesano et al. (2023) performed a review and reinterpretation of public seismic reflection profiles (CROP and ViDEPI

profiles), alongside comparisons with earthquake locations and aftershock distributions from INGV. These authors suggested that the Fano-Pesaro Offshore earthquake sequence took place on a relatively small section (25-40 km²) of the buried Cornelia Thrust System (CTS), situated at the edge of the Northern Apennines (Fig. S1). They also proposed a control by pre-existing normal faults and associated structural highs of the subducting Adria monocline (Amadori et al., 2019; Livani et al., 2018). Their work confirms the CTS being an active fault, It is roughly 300 km² in size, which could produce ruptures up to magnitude 6.5 and may trigger nearby faults.

Pezzo et al. (2023) characterized the seismic sequence in space and time, using data from the INGV monitoring system, GNSS-constrained coseismic slip, and public seismic reflection profiles (ViDEPI). They observed shallow buried anticlines in the upper 5-6 km of the crust with ramps dipping 20°–35° extending from a deeper, regional basal décollement with a westward dip of 1°–7°. Based on the distribution of relocated aftershock events, the authors interpreted a 15 km long striking seismogenic fault patch, dipping 24° SSW and seismically active at depths of 5– 10 km. Their mainshock relocation generated a 4.4 km shift to the south and a depth increase down to 8 km.

Lavecchia et al. (2023) examined the multi-scale geometries of slowly deforming continental regions (SDCR) in eastern Central Italy, focusing on lithospheric-scale deformation (De Nardis et al. 2022). They suggested the presence of a shallow megathrust (T1, ~ 20 km to few km deep) which represents the basal detachment of the external fold-and-thrust domain of the Adriatic Arc. These authors propose the T1 splay, named Bice thrust, extending ~ 30 km with a listric geometry (dip angle ~ 40°– 20°, seismogenic depths ~ 7– 11 km) and converging at depth with the Cornelia Thrust. Upon associating the first mainshock (Mw 5.5) with the central and southern part of the Bice thrust, they interpret the second event (Mw 5.2) due to the subordinate activation of the northern part of the Cornelia Thrust. Following this study, Pandolfi et al (2024) conducted a probabilistic seismic hazard analysis for the Adriatic Thrust Zone (ATZ).

An et al. (2024) proposed a new workflow to relocate the Fano-Pesaro seismicity clusters in a depth range of 2–12 km, with a best-fit dip of about 30° towards the south-southwest. In comparison to the results available in the INGV catalogue, they presented a sharper earthquakes cluster closer to the shoreline, mapping a geometry coherent with the available focal mechanisms as well as with the horizons interpretated in seismic reflection profiles.

While the approaches, results and interpretations on thrust geometries, dimensions, depths and structural relationships might differ, all the above-mentioned studies agree that 2022 earthquakes are related to an averagely ~ 30° dip, southwest-dipping thrust fault, located in the frontal part of the Northern Apennines. However, different opinions remain about which thrust could be the causative structure for the recently recorded seismicity.

**4. Data and methods**

The findings outlined in this paper are based on the interpretation of four deep wells (Table 1) and a set of seismic reflection profiles covering an area of approximately 1400 km², five of which are described and discussed in detail. No digital data (e.g. SEG-Y files) were available to be used for enhancing the quality of the dataset (e.g., Barchi et al., 2021, Ercoli et al.,

2023, Carboni et al., 2024), but all the seismic reflection profiles were provided as digital images, scanned from hard paper copy, in pdf format. Three of the selected seismic reflection profiles and a key-borehole, kindly provided by the Italian Energy company Eni S.p.A. under a confidential agreement, are unpublished. The other boreholes and seismic reflection profiles were retrieved from publicly available datasets from ViDEPI databases (https://www.videpi.com; https://www.crop.cnr.it) (Figs. 1 and 2, Table 1), along with industrial exploration reports and maps, which have been deeply reviewed.

A workflow, including different steps to gather and analyse all the data and ancillary information, has been set up:

1. *Data preparation*: data organization, quality control (QC), digitalization, georeferencing and importing into a
geoscience multi-discipline integration software. 2D and 3D visualization of seismic reflection profiles, wells stratigraphy (formation tops), log images, and seismicity.

2. *Data integration*: stratigraphic correlation among the wells' tops and logs to identify a local seismic stratigraphy, well-to seismic tie analysis and seismo-stratigraphic interpretation.

3. *Velocity model building*: a key well sonic log (Table 1) was used to extract velocities for Pleistocene and Pliocene
formations, whilst literature velocities were adopted for deeper layers (older than Late Miocene).

4. *Time to depth conversion*: horizons, faults and surfaces were converted to depth and the correlations were extended and verified across a broader area.

5.

**Table. 1. List of datasets (Sp= Spontaneous Potential, Res= Resistivity, Sn = Sonic). The star\* marks the unpublished data,**
**obtained under a confidential agreement, the hashtag# reports the public data downloaded from the Italian database ViDEPI.**

| Seismic profiles | | | | Wells | | |
|---|---|---|---|---|---|---|
| Type | Name | Length (Km) | Notes | Name | Depth | Logs |
| Cross line (NE-SW) | S1* | 18 | Intersected by W1 well | W1* | 4300 m Reached the Lower Cretaceous (Calcari Di Cupello (CDC) Fm). | Sp, Res |
| | S2* | 11.5 | Adjacent to main shock (134 m) | Tamara 01# | 3191 m Reached the Lower Miocene (SCH Fm) | Sn, Sp, Res |
| | S3# (B-402) | 30 | / | Pesaro Mare 04# | 4258 m Reached the Lower Jurassic Dolostone (MAS Fm). | Sp, Res |
| | S4# (SV-167-13) | 21 | Intersected by Cornelia well | | | |
| Tie line (NW-SE) | S5* | 22 | Adjacent to Pesaro mare 04 well | Cornelia 01# | 3976 m Reached the Lower Jurassic Dolostone with Chert (Non defined ~ MAS Fm). | Sp, Res |

[Figure]

**5. Results**

**5.1. Wells' stratigraphy**

The wells' stratigraphy was digitized, analysed to identify common geological characteristics (e.g., stratigraphy, lithology,
discontinuities, petrophysical properties derived from the logs) and trends (formation thickness, spatial continuity) among
the wells. After reviewing and correlating the lithological and structural information among all the data, a reinterpretation of
the wells' stratigraphy has been accomplished and displayed in Figure 3. In the latter, the analysed wells are displayed
sequentially, moving from the northwest to the southeast of the study area (Table 1, red arrow in Fig.3a). The data has been
summarized, aiming to clearly show the tectono-stratigraphic correlation among the four wells, highlighting the spatial
variation and gaps due to the presence of erosional and tectonic discontinuities (Fig.3b). Aiming to a deeper understanding of
subsurface geology within the study area, such well information was spatially extrapolated along the available seismic
reflection profiles, by correlating them with the interpreted TWT (Two-Way Travel Time) seismic horizons ("well- to-
seismic tie", Bianco, 2014) and fault sets.

The W1 well intersects the easternmost segment of the seismic profile S1, containing 160 m of Lower Cretaceous
carbonates. Within this well, three erosional boundaries are identified, corresponding to the Messinian, middle-lower
Paleocene, and Lower Cretaceous tops (Fig.3).

The Tamara 01 well, located 600 m southeast of the seismic profile S2 and near the epicenter of the 5.5 Mw mainshock of
the 9th November 2022, provides valuable sonic log data for deriving interval velocities and conducting well-to-seismic tie
analysis. Projected orthogonal onto the eastern segment of the S1 and S2 seismic profiles, Tamara 01 well penetrates the
upper Miocene SCH Formation for about 176 m. The well exhibits four erosional and two tectonic boundaries. The erosional
boundaries are identified within the Lower Pleistocene at depth of 1217 m and at two levels marking the tops of the Upper
Pliocene (1370 and 1912 m) and one level marking the top of the Upper Miocene, located at depths of 3015 m. The two
tectonic boundaries are recognized from the repetition of the Miocene-Pliocene sequences at depths of 1743 and 2345 m,
respectively (Fig.3).

The well Pesaro Mare 04, situated approximately 1 km southwest of the S3 profile, was projected orthogonal onto it. The
well penetrates the sequence down to the Lower Jurassic, encompassing 1729 m of dolomitized MAS. Notably, an erosional
boundary corresponding to the Miocene top is documented in the well stratigraphy at a depth of 372 m (Fig.3).

The Cornelia 01 well, located in the southeastern part of our study area, intersects the seismic profile S5. It penetrates
Jurassic dolomitized carbonates, which are originally referred to an *undefined* formation based on the lithological variations
and on the reported depositional environment; however, it is equivalent to the Dolomie di Castelmanfrino (DCM) formation.
This well exhibits five erosional boundaries corresponding to the tops of the Upper Pliocene (686 m), Lower Pliocene (738
m), Upper Miocene (790 m), Upper Cretaceous (1833 m), and Lower Cretaceous (2478 m).. Additionally, a tectonic
boundary is reported approximately 30 m from the bottom of the well. It is interpreted as a thrust splay, whose offset results
in the repetition of the Early Cretaceous succession (Fig.3).

[Figure]

[Figure]

**Fig. 3. a) Location map showing the position of the analysed wells. b) Schematic stratigraphic columns of the wells, reinterpreted from the original data in the ViDEPI database and arranged spatially from northwest to southeast (red arrow in Fig. a).**

[Figure]

[Figure]

From the global analysis of the four wells' data across the study area (Fig.3), the Pliocene-Quaternary successions show a significant thinning from ~ 3100 m thickness in the northwest to 400– 700 m in the southeast, as recorded in Pesaro Mare 04 and Cornelia wells, respectively. Within this succession, the Pliocene-Pleistocene sedimentary sequence is frequently incomplete. Notably, in the Pesaro Mare 04 well, situated on a structural high, the Pliocene succession is entirely absent, with a direct transition from Miocene deposits to Quaternary sediments. Conversely, in the basin areas, such as the W1 well, a more complete sequence spanning the lower to upper Pliocene is preserved. This sequence is characterized by alternating sandy and clayey layers, often interbedded with marly components. This sequence unconformably overlies the Messinian (GS) evaporites, which are identified exclusively in the northwestern and southeastern wells of the study area. These evaporites are associated with a Messinian paleo-high that persisted as a subaerially exposed feature for the majority of the Pliocene (Report 1508, ViDEPI).

The lithological analysis of the Meso-Cenozoic carbonate successions within the studied wells reveals a carbonate platform that underwent progressive deepening, testified by the combination of detrital and dolomitic limestones, interspersed with frequent cherty nodules and marly intercalations, particularly in the lower sections. The Triassic succession (BF), which typically consists of evaporites and dolostones in the central Apennines (e.g., Umbria-Marche and Sabina Pelagic Basins), in this study area is almost entirely composed of dolostone facies, as reported by the analysed wells. This is also shown by the

Alessandra 1 well, located slightly to the east, which represents the deepest borehole drilled in this region (Bally, 1986; Carminati et al., 2013). As the succession transitions into the Middle Jurassic and extends to the Paleogene, the limestones gradually give way to marly layers, again characterized by typical nodular structures. Additionally, clastic intercalations are observed, suggesting sedimentary inputs from the erosion of adjacent structural highs. Notably, the thickness of the SCA Group increases significantly from the northwestern to the southeastern studied wells

**5.2. Seismic stratigraphy and time-to depth conversion.**

By correlating and calibrating the stratigraphy of Tamara 01 and W1 wells with all the available seismic profiles, we have identified five primary seismic units (SUs), bounded by four prominent, easily traceable key-reflections. These units exhibit distinct geophysical signatures, such as variation in the reflection amplitude, period and geometry. The analyzed seismic profiles follow SEG normal polarity, meaning that an increase in acoustic impedance is represented by a peak, while a decrease corresponds to a trough. The SUs are discussed in the following from top to bottom (Fig.4 and details within the Supplementary Table S1).

The Holocene-Pleistocene turbidites (SU1) comprise fine sandstones, shaly sandstones, and interbedding of shale and silty shale pertaining to AS. SU1 consists of four distinct seismic sub-units (SU1 a, b, c, d), each one characterized by a different seismic signature (see supplementary Table S1). SU1 a, b are characterised by seismic facies from continuous to semi discontinuous horizontal and parallel reflections, with low to high amplitudes; the bottom SU1 c, d display continuous to

semi-continuous E-dipping reflections, with medium to high amplitudes. The total thickness of this unit gradually increases north-eastwards from 0.2 s to 0.3 s (Fig.4 and Supplementary Table S1).

SU2 is separated from SU1 by a top-lap unconformity, dated Top Gelasian (Fig. 4) referring to early Pleistocene older than 1.8 Ma. The Gelasian turbidites within the upper part of PG consist of silty shales with interbedded shales at the top, transitioning to fine sandstones and shaly sandstones in the lower part. The thickness of this unit gradually increases from 0.2 s in the SW to 0.6 s in the NE. Within SU2, we identified two sub-units (SU2 a, b), each one characterized by distinct seismic signature. The uppermost sub-unit (SU2 a) shows continuous E-dipping parallel reflections, with medium to high amplitudes. In contrast, the lower sub-unit SU2 b features semi-continuous, parallel, and sub-horizontal reflections (Supplementary Table S1).

The Pliocene turbidites (unit SU3) within the lower part of PG are composed of silty marls intercalated with medium to very fine-grained sandstones. The subunits (unit SU3 a, b, c) display distinct reflection patterns. The uppermost subunit (SU3 a) exhibits continuous, horizontal, parallel reflections with high amplitude, while the other subunits (SU3 b and SU3 c) show discontinuous to semi-continuous, sub-parallel reflections with low to medium amplitudes. Their thickness variation across different sections is ranging from a few ms to 0.4 s (Fig. 4).

The complex Miocene succession (SU4) found within the SCH, and BIS, are composed of shales and marls interbedded with siltstones, carbonates, and minor gypsum deposits. This marly group displays continuous, parallel reflections with high amplitude and dominant frequency in the narrow uppermost part; the rest of the unit presents continuous to discontinuous, sub-parallel reflections with medium to high amplitude. This seismic unit progressively deepens from southwest to northeast. The high amplitude and dominant frequency within this unit create distinct and sharp reflections in the seismic sections.

The Mesozoic-Paleogene carbonate multilayer (SU5) unit corresponds to the SCA, MAS and DCM. and represent the deepest recognized units. The unit consists of limestone and dolomitized limestone, with intercalations of marls and chert nodules. Notably, it exhibits a substantial thickness of over 1 s. The reflections within this unit display a discontinuous, sub-parallel pattern with low to medium amplitude and are marked by some continuous, high amplitude and well-recognizable reflections which are related to the top of the SCA and FUC fms. (Mirabella et al., 2008; Porreca et al., 2018; Barchi et al.,

2021).

For the depth conversion, a velocity model has been built, by integrating new interval velocity values derived from the sonic log of the Tamara 01 well (down to the Late Miocene turbidites) with literature velocity data (e.g., Bally et al., 1986; Maesano et al., 2013, 2023; Montone and Mariucci, 2023). Bi-dimensional velocity models were initially built up along each single profile, with a focus on the shallower area (down to the Top Scaglia). This workflow was then extended across a tri- dimensional workspace, encompassing later variations driven by all the picked horizons and faults surfaces, and considering some control points corresponding to wells located a broader area. Such a velocity model was later refined in its deeper portion (down to the Jurassic carbonate units) and used to carry out the final conversion from the time to the depth for all the selected seismic profiles. Further details on the velocity models are provided in the Supplementary Table S2.

[Figure]

[Figure]

[Figure]

**Fig. 4.** **Seismic stratigraphy of the study area (colored lines) calibrated using the Tamara 01 and W1 wells, (see Fig. 1 for well locations and Fig. 3 for stratigraphy column abbreviations). Vp indicates the P wave seismic velocity. Additional details are provided in the text and supplementary Table 2.**

**5.3. Seismic interpretation**

To provide an accurate representation of the subsurface geological and structural features within the research region, five seismic profiles have been selected to carried out the seismic interpretation. Their location and details are reported in Figures 5, 6 and Table 1, while the uninterpreted versions can be found in the supplementary material (Figs. S1 and S2). The dataset includes four SW-NE-oriented "*cross-lines*" (S1, S2, S3, and S4) and one NW-SE oriented "*tie-line*" (S5). The SW-NE profiles cross the two major anticlines present in the area, namely the northern Pesaro Anticline (PA) and the southern

[Figure]

Cornelia Anticline (CA), developed at the hanging walls of SW-dipping thrusts, named Pesaro thrust (PT) and Cornelia thrust (CT).

The whole interpretation of seismic profiles has been realized by using a tri-dimensional correlation of key reflections picked along the single seismic profiles, with respect to seismic-stratigraphic units obtained from the well-tie analysis. In this section, the description of the seismic profiles is done from northwest to southeast. The profiles are described considering the increasing TWT (s) and their along line distance (km).

The seismic profile S1 in Figure 5a is dominated by the east-verging PA, characterized by a long wavelength of ~ 13 km (0–12 km distance, Fig. 5a). The PA geometry is traceable from ~ 0.2 s down to ~ 2.5 s, and it is particularly evident following the interpreted Top Jurassic to Top Messinian reflections (blue and pink colours, respectively). To notice that the Top Messinian reflection is not traceable in the culmination of the PA anticline, due to erosion; in addition, a set of minor folds characterize the PA forelimb (9–12 km distance range). Further to the northeast, between 12 and 17 km distance, a complex antiformal structure (wavelength ~ 5 km) folds the Plio-Pleistocene unconformity reflection (dark yellow colour).

This antiformal stack involves a set of minor imbricates, with wavelength < 1 km, detached above the Top Carbonates (Oligocene) reflection (light green colour). The antiformal stack is here referred to as the Tamara structure (TS), drilled by the Tamara 01 well. The PA and TS are separated by a short wavelength (~ 4 km) syncline (~ 9–13 km distance), which is infilled by sub-horizontal reflections interpreted as lower Pliocene sediments, onlapping onto the Plio-Pleistocene unconformity (Fig. 5a). In the northeastern part of the profile (~13–17 km), a clear increase in the apparent dip angle and thickness of the Pliocene succession reflections is visible. Both the PA and the TS are interpreted to be situated in the hanging wall of the SW-dipping main PT thrust. The hinge zone of PA is located on top of the main PT ramp, located within the Mesozoic succession; this ramp links its deepest part with the shallowest, flat portion at ~2.5 s (Fig. 5a). However, in this forelimb sector, a set of imbricate forethrusts and backthrusts have been interpreted departing from PT. These backthrusts have been associated with the minor folds described above on the PA forelimb (~9–13 km distance range). Such backthrusts are detached along the shallower, most internal PT ramp. On the other hand, the set of imbricate forethrusts, build up the shorter wavelength TS and they are all detached along the PT shallower flat. The three imbricates displace up to the Top Messinian and the Top lower Pliocene reflections of at least ~0.1 s TWT, but not the Plio-Pleistocene unconformity, which is only folded. The presence of such imbricates is also interpreted and constrained by the Tamara 01 well stratigraphy, clearly showing two repetitions of the Top Messinian. Further constraints on the PT geometry derive from a set of parallel sub-horizontal reflections observed between 2.5 s and 3.5 s (5–18 km range); they are discordant with the shallower reflections, especially in correspondence with the main ramp, between 3 and 8 km distance, where they look slightly E dipping. These reflections would represent the PT footwall succession, up to the Top Messinian (Fig. 5a).

The seismic profile S2 (Fig. 5b) gives a clearer picture of the TS imbricates. Projecting the Tamara 01 well and picking the Top Messinian reflections, the presence of three imbricates within the TS, which produce three repetitions of the Messinian and Pliocene successions, have been interpreted. The imbricates are detached on the shallow PT flat (~ 2.5 s TWT), which produces a further repetition of the Top Messinian reflection (pink colour). In the south-western part of S2, again the minor

folds driven by the backthrusts mapped in S1 are observable in the north-easternmost part of the PA forelimb. Within S2,
like in S1, the growth deposition of the Pliocene succession is also observed in the northeastern part (apparent E-dip), and
the syncline separating the PA and the TS, again characterized by parallel sub-horizontal reflections associated with the
Pleistocene unit (Fig. 5b).

**Fig. 5.** Interpretation of S1 and S2 seismic profiles. a) S1, the northernmost section in the study area, crosses the left hinge zone of
the PA and reveals variations in its structural style. The geometry of the PA is evident by using the Top Messinian, Top Oligocene
and Top Lower Cretaceous reflections. The section also shows TS shallow seated structure developed laterally to the South-
Eastern termination of the PA. b) S2 shows the enhanced comprehension of the TS's underlying structure. Four imbricated
thrusted zones of the PA forelimb and the repetition of Messinian- middle- lower Pliocene successions are observable
(uninterpreted images provided in supplementary materials, Fig.S1).

[Figure]

[Figure]

The seismic profile S3 (Fig. 6a) provides an excellent view of the structural relationships between the two main structures of the area: the main thrusts PT and CT with their related anticlines PA and CA. The PA is displayed in the southwestern part of the profile (0–10 km distance range). Its geometry can be easily appreciated by following the Top Jurassic to the Top Messinian reflections (blue and pink colours, respectively); the latter is again partially eroded in the axial zone. A few smaller antiformal structures located at the PA forelimb, as already observed in S1 and S2, are again interpreted as being driven by small backthrusts. This profile also shows a strongly reduced size of TS and a steepening of the PT here partially overlies the western flank of another anticline, identified as CA. More north eastwards, the latter appears as an asymmetric NE-verging anticline, traceable from ~ 0.8 s down to ~ 3 s. This anticline is interpreted being related to the underlying CT, whose location is constrained by the Cornelia 01 well. The CT displaces the Meso-Cenozoic succession up to the Top lower

Pliocene reflection (orange colour), while the Plio-Pleistocene unconformity (yellow colour) appears only folded. The CT footwall is recognized following the Top Jurassic to the Top Messinian reflections, which are interpreted slightly parallel and W-dipping until around 18 km distance at ca. 3 s. The CT is interpreted to comprise also a small synthetic thrust, developed at its footwall, which produce a further repetition of the Top Scaglia Group and Top Messinian reflections. More to the northeast, we observe a shallower and thick package of growth strata, interpreted to comprise Pliocene to Quaternary deposits.

[revised manuscript text omitted]

striking ~ 20 km long and ~ 12 km wide anticline (profile S4 in Figs. 6b, 7b, 7c) with a wavelength λl of ~ 11 km (Figs. 6a, 7b). These structural wavelength values, λl and λs, are larger than those obtained for corresponding structures in the Umbria-Marche area, where corresponding structures have wavelengths of 3.2 to 7.2 km for λl and 0.4 to 2.3 km for λs (Massoli et al., 2006), characterized by lower syn-tectonic sedimentation. Conversely, and they are smaller than those observed in the Po Plain, where higher syn-tectonic sedimentation contributes to even larger structural wavelengths, with λl ranging from 15.8

to 33 km and λs from 4.5 to 8.2 km (Massoli et al., 2006).

[Figure]

[Figure]

**Fig. 7.** Geological sections derived from: a) seismic profile S1, b) seismic profile S3, and c) seismic profile S4. The main shock of 5.5 ML and aftershocks are projected normal to the section within buffers of 5, 7, and 10 km, respectively. d) Location map of the interpreted anticlines and thrust faults (this work); the seismicity distribution is sourced from terremoti.ingv.it. $\lambda_l$: Wavelength of the large structures; $\lambda_s$: Wavelength of the small structures.

[Figure]

Our comparative analysis of the PA and CA anticlines, and their related deep-seated thrust systems PT and CT points out some structural similarities and distinction. From the analysis of the profiles S3 and S4 (Fig. 7b, 7c), looking at both anticlines geometry and the thrusts trajectories, it is clear how the thrusts share a common deep décollement level, at approximately 8–9 km depth (comparable to results in nearby areas provided by e.g., Pauselli et al, 2006, or Lavecchia et al., 2004). Furthermore, evidence indicates that the thrusting style in this area is a thin-skinned type of deformation, aligning with the observed decollement depth and suggesting tectonic processes that control syn-tectonic sedimentation and accommodate deformation within the overlying sedimentary cover, without involving the basement (Fig. 7). Our interpretation demonstrates that, unlike the PT, the CT lacks an upper shallower décollement. Instead, the ramp of the CT

terminates blindly at a depth of 2 km within the base of the upper Pliocene turbiditic successions (Figs. 7b, 7c), and only one imbricated fore-verging thrust has been identified in S4. The latter is also constrained by the Cornelia borehole stratigraphy, evidencing a doubling of the early Cretaceous carbonate succession (Scaglia group) over approximately 4 km.

Considering the deeper structures involving the carbonates, this study documents the structural transition between two main compressional structures: the PA (internal) and the CA (external) anticlines. In map view (Fig.7d), these structures are linked to a pair of en-echelon, vicariant, coalescent thrusts, the northernmost PT and the southernmost CT. The interpretation of the seismic lines clearly highlighted that the transition from PT and CT occurs through an intermediate region, where both structures are present  (Fig. 7d) and can be viewed as adjacent segments of the outermost thrust of the Northern Apennines. Representative examples of coalescent anticlines extensively crop out also in the Umbria-Marche Apennines (Barchi et al., 1998; Scarsella 1941; Lavecchia, 1981; Lavecchia et al., 1988; Lavecchia et al., 2023), and such examples have been described worldwide since Dahlstrom (1970).

Our investigation shows that the shallow-seated TS structure can be traced only in the southeastern termination of the deep-seated PA up to seismic Profile S3, where both PA and TS overlap on the back limb of the CA (Fig. 7b). However, in the southeastern part of our study area, as seen in seismic profile S4, the shallow-seated imbricated fore-verging thrusts and their related antiformal stacks (TS) are not observable (Fig. 7c). Our investigations indicate that the TS represents the deformed wedge of the frontal part of the PA structure, thus it cannot be considered originated by a single deep-seated structure such as PT or CT and neither a northwest-eastward continuation of the Cornelia thrust.

In slightly external sectors evidence of deep thrusts has been reported from the analysis of low-quality public profiles (Adriatic Arc Front, e.g., Bice thrust, Lavecchia et al., 2023). However, the present study suggests that the PT and associated imbricates did not extend more to the North-East. This consideration is also testified by the presence of a complete sedimentary succession (from Cretaceous carbonates to thick Quaternary sequences). Additionally, in the borehole W1 (drilled in the foreland of the PA), no thrust faults are reported and the Top Messinian reflection correlates well with the corresponding identified erosional boundary. Evidence of deeper, external fronts were not found in the reviewed commercial seismic reflection profiles available across this study area, possibly falling besides the data quality at depth or outside the data coverage.

[Figure]

[Figure]

**6.2. Seismotectonic implications**

The mechanical stratigraphy reveals that both the deep-seated PT and CT ramps cut through the brittle carbonate multilayer, from 3 down to 9 km depth. This range coincides with the depths of most of the seismicity recorded during Fano-Pesaro 2022 sequence (terremoti.ingv.it, Rovida et al., 2022), suggesting that these thrusts may potentially serve as seismogenic structures. Both PT and CT are southwest-dipping thrusts, with an interpreted dip angle of 30°–35°, compatible with the mainshocks focal mechanism (with strike 128°, dip 34° and rake 84°, terremoti.ingv.it).

Given their potential seismogenic role, the relationship between earthquake magnitude and subsurface rupture length for both the PT and CT was analysed using the regression diagrams (e.g. Wells and Coppersmith,1994 and Leonard, 2014). Fault length directly influences the maximum possible displacement, and consequently, the potential maximum magnitude (Scholz, 2019). According to the findings of this analysis, the observed fault lengths are substantial enough to account for both recent and historical seismic activity in the region.

However, determining the exact causative faults for the 2022 November 9th earthquakes remains challenging. It is important to highlight the spatial mismatch (Fig. 1), in terms of both location and depth distribution, among the literature interpreted faults and the hypocentral records (terremoti.ingv.it). Comparing the published earthquake locations and relatively shallow depths (~ 5 km) with our  interpretation, seismicity is scarcely distributed across the Cornelia region (Fig. 7d). The first November 9th 5.5 Mw main shock appears more closely associated to the PT (extending more to the North-East), other than to the CT, the latter being less extended to the North-West (Fig. 7d). The second November 9th mainshock and the aftershocks fall in between the area covered by the seismic profiles S1 and S3, in the interpreted transfer area between PT and CT. This event is close to the PT zone and somewhat far from the CT's main area but occurring at greater depth (~ 8 km) in the footwall of the PT (Fig. 1, Fig. 7d). However, it is known that both earthquake hypocentres location and the depth of the "not-relocated" seismicity lack in accuracy, particularly in the offshore, due to the limited coverage of seismic stations. Recently, several authors has re-located the seismicity recorded during this 2022 sequence. Pezzo et al. (2023), An et al. (2024) and Costanzo (2024) used different relocation methods and methodological approaches and a significant uncertainty in defining seismic event depth compared to the location is noticeable.

**Table 2.** Location and depth parameters of the mainshocks for the 9th November 2022 Fano-Pesaro earthquake, as determined by different sources.

| Event | Source | Depth (km) | Latitude | Longitude |
|---|---|---|---|---|
| Main Shock (Mw5.5) | INGV | 5.0 | 43°58'59"N | 13°19'26"E |
| | An et al., 2024 | 4.40 | 43°56'11"N | 13°20'20"E |
| | Pezzo et al., 2023 | 7.94 | 43°59'41"N | 13°18'58"E |
| Second Shock (Mw 5.2) | INGV | 7.7 | 43°54'47.88"N | 13°20'40.92"E |
| | An et al., 2024 | 8.4 | 43°51'36.36"N | 13°20'16.44"E |

[Figure]

The first relocation by Pezzo et al. (2023) shifted the main shock 1.5 km N-NW, increasing its depth to ~8 km, while aftershocks moved slightly NE and farther offshore. The second relocation by An et al. (2024) shifted the main shock 5 km southward, thus closer to the shoreline, with a shallower depth, and relocated the aftershock cluster 6 km S-SE. The spatial distribution of the relocated aftershock events, as well as the historical seismicity in this area, is farther from the CT and more concentrated around the PT and the transfer zone between the PT and CT (Figs. 1 and 7d).

This analysis underscores the complexity of determining whether the PT or CT served as a primary source of the 2022 seismic activity or if the latter might be associated to a possible deeper thrust (e.g., T1 as supposed by Lavecchia et al., 2023). However, such a possible causative fault is not imaged at depth within our available seismic reflection data, possibly due to the high level of random noise characterizing the legacy profiles (Ercoli et al., 2023) or due to the lack of reflected signals from deeper structures.

**7. Conclusions**

This paper presents a new geological model of the tectonic structures of the Fano-Pesaro offshore area within the frontal part of the Northern Apennines. Multiple decollements located at different depth have been observed in the study area. These structures show a strong relationship between the depth of faulting and the wavelength of the related anticlines, influencing the kinematics of the thrust system. The PT and CT structures are detached at depths of ~ 9 km on top of the acoustic basement. The two related PA and CA structures can be followed along strike for about 50 km and are characterized by a wavelength in the order of ~ 11 km. The TS develops along the shallow part of the Pesaro thrust at a depth of 3.5 km, is characterized by a short wavelength (~ 1.1 km) of the imbricates spread along ~ 5 km in the forelimb of PA, and it can be followed only for ~10 km along strike. The PT and CT en-echelon arrangement, the presence of multiple detachments and the thin-skinned deformation (multiple décollements) suggest a geological model for this outermost sector of the Apennines characterized by a thrust system not involving the  basement (thin-skinned tectonics).

This study highlights a possible minor role of the Cornelia thrust system during the 2022 earthquakes than previously though due to a more limited extent to the NW. Although based on its geological, structural and geometrical characteristics this thrust system cannot be excluded as a seismogenic source, the historical and recent seismicity directly affecting the CT, with its limited extension toward the north, is scarce and cannot be easily linked with it. The integration of the relocated hypocentres and the new geological model suggests that the PT, or a possible easternmost deeper structure, would be better candidates to be associated with the mainshocks. On the other hand, the relay zone between PT and CT is more coherent with the second main event. The still present uncertainty is mainly due to the low accuracy of the seismicity relocation caused by lack of seismic stations and simplified velocity models used. This work aims to remark that defining a solid subsurface geological model by integration of key reflection seismic profiles and boreholes data (even if legacy) is essential in offshore areas. Building up a reliable, geologically driven model, allows to refine not only velocity models to use for more accurate earthquakes' relocation, but for increasing the reliability of seismotectonic studies and risks assessments. The advancement of geological and geophysical studies might have broader benefits also on other application, such as supporting safer exploration projects of carbon capture and storage along the ==NA== sea region.

**Data availability**

Supplementary data associated with this article can be found, in the online version

**Author contribution**

ES, ME, and MB contributed to the conception and design of the study. ES performed data curation, analysis, investigation, methodology, visualization, preparation of tables, maps and figures, and wrote the original draft of the manuscript. ME
contributed to investigation, writing and revision. FC contributed to writing, revision, visualization maps and figures. FM and AA contributed to the investigation, and revision. MB was responsible for conceptualization, resources, supervision, and review and editing. All authors contributed to the final revision of the manuscript and approved the version submitted.

**Competing interests**

The authors declare that they have no conflict of interest

**Acknowledgment:**

The authors express their gratitude to Schlumberger, Cegal, and PE Limited for providing academic licenses for their software platforms and plugins to the University of Perugia. These software and tools were used for uploading scanned legacy seismic profiles and well data (Blueback tools, Cegal), well digitization, and seismic interpretation with Petrel, as
well as for structural modeling, and velocity model analysis with Petrel and Move. We also acknowledge the support teams of Petrel, Move, and Cegal for their prompt assistance in resolving technical challenges during the study. Additionally, we sincerely appreciate the QGIS and Inkscape Teams for developing free and open-source software, which was essential for data organization, correlation, and analysis (QGIS), as well as for image enhancement and digitization (Inkscape). We extend our special thanks to ENI for providing subsurface data under a confidential agreement, and to the Ministero dello
Sviluppo Economico DGRME, the Società Geologica Italiana, Assomineraria, and CNR ViDEPI (https://www.arcgis.com/) for granting access to public domain data and services. We are also grateful to An and the Pezzo groups for sharing with us their catalogs of the relocated seismicity.

---

## Author Response (AR1)

**Note:** The line numbers correspond to the track-changed version of the manuscript.

**Editor**

Dear Editor,

We sincerely thank you for the opportunity to revise and improve our manuscript based on the insightful comments and suggestions provided by the reviewers. We greatly appreciate the time and effort they dedicated to evaluating our work. We have carefully addressed all comments in the revised version of the manuscript. A detailed, point-by-point response to each remark is provided in the accompanying discussion document.

\_\_\_\_\_\_

**Reviewer 1**

Dear Reviewer 1,

Thank you very much for your thoughtful and constructive comments, which have significantly contributed to enhancing the quality of our manuscript. We have implemented the majority of your suggestions, and for the few cases where changes were not made, we have provided detailed justifications in our response.

**General comments**

1. **First referee comment: Line 1:** "I suggest changing the title since no comprehensive geological model is presented. The authors propose the interpretation of a set of seismic profiles tied with well logs, but they do not propose a general interpretation of the tectonics of the region and/or provide subsurface maps as the reader would expect from the title's incipit."

**Reply:** Thank you for your valuable comment. We have slightly revised the title to better reflect the scope of the study. We would also like to clarify that our paper does not aim to present a comprehensive regional geological model. Instead, it focuses on illustrating a methodological workflow for constructing a geological model in active offshore areas, using the Fano–Pesaro earthquake region. In this study, we introduce a new local structural model characterized by multiple décollement levels, supported by the interpretation of seismic profiles and well log data.

**Changes: Line1:** "New geological constraints on the subsurface structure of the 2022 Fano-Pesaro Mw 5.5 earthquake sequence area (Adriatic Sea, Italy) from legacy seismic reflection images and deep well information."

2. **First referee comment**: The graphical quality of both interpreted and uninterpreted seismic profiles is low. Higher quality versions exist in the literature (e.g., Maesano et al., 2023) and is better to be used

**Reply**: Thank you for your valuable comment. During the submission and revision process of this manuscript, we were working on digitizing our studied profiles. Through filtering and simple post-processing, we were able to achieve higher-resolution seismic data. Based on your suggestion, we have replaced the previously used hard copy versions with these improved profiles. The new data allowed for clearer visualization and more detailed interpretation, which has been incorporated into the revised figures. **Change:** Please See Figures 5 and 6 on pages 25 and 28, respectively, and Figures S1 and S2 in the Supplementary Materials.

3. **First referee comment:** "Regarding profile S1, there are a couple of points concerning the interpretation. At first glance, the chaotic seismic signal within the shallow splays in the TS may also involve some part of the Mesozoic succession to allow for section balancing. Also, concerning the S1 profile, I understand that the seismic image quality at the footwall of the thrust is poor. Still, the way the reflections of the succession are interpreted below the thrust appears inconsistent with the limited available data and could lead to the balancing problem that will be discussed subsequently."

Reply: Thank you for your insightful comments. After digitizing and post-processing the seismic profiles, the resolution and clarity of the reflectors improved slightly. We observe that multiple décollements may be present— one shallower at the top of the Messinian and another at the top of the Fucoidi—similar to the Umbria-Marche region model proposed by Massoli et al. (2006). This interpretation applies at least to the first three imbricated thrusts in sections S1, S2 and two imbricated thrust in S3, involving the Scaglia and a small section of the Fucoidi. However, for the frontal imbricates, the Tamara well drilled close to section S2 confirmed that these frontal thrusts do not involve the older successions (Fig. 3). Solid lines indicate reflectors that are clearly visible, while dashed lines represent inferred continuations based on geometry and thickness. Our interpretation integrates seismic reflections with previous studies, structural contour maps, and nearby well data to ensure consistency with the regional geology.

**Changes:** Interpretations in Figures 5 and 6, as well as the geological sections in Figure 7, have been updated on pages 25, 28, and 33, respectively.

4. **First referee comment:** "Profiles S3 and S4 show a similar issue in interpreting the footwall reflections. Even though the authors map these reflectors as inferred (using dashed lines), they may not fully consider the sparse evidence that could help them produce a more accurate interpretation. Alternatively, they might omit the interpretation in these parts of the profiles if they deem them too speculative."

**Reply:** Thank you very much for your helpful comments. As mentioned before, our interpretation is based on seismic reflections, and also supported by previous studies,

structural contour maps, and nearby well data. For the Top Burano and Top Acoustic Basement, no wells in the area reach those depths. We interpreted these reflectors using regional correlations with nearby deeper wells, previous studies and available geological contour maps. The dashed lines were not due to uncertainty in the tracing of the reflectors, but because we are not fully sure about their nature. Based on your suggestions, we have updated the profiles. We now show solid lines where the reflectors are clearly visible, to better reflect that our interpretation is based on actual seismic evidence, not speculative interpretation. We appreciate your detailed review, which has helped us improve the clarity of our figures and interpretations.

**Change:** Please see Figure 6 on page 28.

**Reviewer 1 Specific comments:**

5. **First referee comment: Line 146:** "Line 130 and Figure 2: "Calcare Di Asprigni" should read "Calcare Diasprigni". Figure 6: On the S4 profile, the intersection with profile S5 is missing. On profile S5, the lower splays of PT between km 14 and km 28 do not have a perfect correspondence on profile S3 and could be improved. Specifically, the splays are traced below the Pesaro Mare 04 well on profile S5, but they appear to terminate before the well projection on profile S3.".

**Reply:** Thank you for your careful observation. We have corrected the term from "Calcare Di Asprigni" to "Calcare Diasprigni" in Figure 2.

Change: Line 147: "Calcari Diasprigni (CDU),", Please see figure 2 on page 8.

6. **First referee comment: Line 595:**" Lines 481-482: "The fore-verging imbricated thrusts originated from the upper décollement of the PT within weak, marly rocks (ranging from the upper Miocene to Pleistocene), propagates both eastwards and upwards". As discussed earlier, after attempting some restoration, I believe that this interpretation, as currently presented, could be one of the causes of the imbalance. I would not rule out the possibility that part of the Mesozoic-Tertiary succession is involved in these imbricate thrusts. While I am largely in agreement with the structural framework proposed by the authors, this is simply a suggestion for revising the interpretation in this particular area"

**Reply:** Thank you for your valuable detailed observations. As we mentioned in our previous response, post-processing improved reflector clarity, revealing multiple décollements, one at the top of the Messinian and another at the top of the Fucoidi, consistent with the Umbria-Marche model (Massoli et al., 2006). This interpretation fits the first three thrusts in S1 and S3. For the frontal imbricates, the Tamara well near S2 confirms they do not involve older successions (Fig. 3).

**Change:** please see figures 5 and 6 on pages 25 and 28, respectively.

7. **First referee comment: Line 630:** "Lines 491-495: These structural wavelength values,  $\lambda_l$  and  $\lambda_s$ , are larger than those obtained for corresponding structures in the Umbria-Marche area, where corresponding structures have wavelengths of 3.2 to 7.2 km for  $\lambda_l$  and 0.4 to 2.3 km for  $\lambda_s$  (Massoli et al., 2006), characterized by lower syntectonic sedimentation. Conversely, and they are smaller than those observed in the Po Plain, where higher syn-tectonic sedimentation contributes to even larger structural wavelengths, with  $\lambda_l$  ranging from 15.8 to 33 km and  $\lambda_s$  from 4.5 to 8.2 km (Massoli et al., 2006). "This sentence is somewhat unclear. Are the authors suggesting a correlation between the different wavelengths and the amount of syn-tectonic sedimentation?"

**Reply:** Thank you for your comment. In this section, our intention was to highlight a similar relationship to that observed by Massoli et al. (2006), showing a correlation between structural wavelength, the depth of the décollements, and the thickness of syntectonic successions. To clarify this point, we have revised the paragraph and added a short explanation to emphasize that greater syn-tectonic sedimentation and deeper décollements are generally associated with larger structural wavelengths, consistent with the findings of Massoli et al. (2006).

Change: Line 630-637: "These structural wavelength values,  $\lambda_l$  and  $\lambda_s$ , are larger than those obtained for corresponding structures in the Umbria-Marche area, where the wavelength range from 3.2 to 7.2 km for  $\lambda_l$  and 0.4 to 2.3 km for  $\lambda_s$  (Massoli et al., 2006). These structures are characterized by lower syn-tectonic sedimentation. Conversely, the observed structural wavelength values are smaller than those observed in the Po Plain, where higher syn-tectonic sedimentation contributes to even larger structural wavelengths, with  $\lambda_l$  ranging from 15.8 to 33 km and  $\lambda_s$  from 4.5 to 8.2 km (Massoli et al., 2006). This observation is confirmed by the relationship described by Massoli et al. (2006), where variations in structural wavelength are linked to both the depth of the active décollement and the thickness of syn-tectonic sediments."

8. **First referee comment: Line 654:** "Lines 508-511: "Our interpretation demonstrates that, unlike the PT, the CT lacks an upper shallower décollement. Instead, the ramp of the CT terminates blindly at a depth of 2 km within the base of the upper Pliocene turbiditic successions (Figs. 7b, 7c), and only one imbricated fore-verging thrust has been identified in S4." This difference is quite evident, and I would expect further consideration regarding the reasons for the absence of the shallow décollement in the CT"

**Reply:** Thank you for your helpful comment and detailed observation. The lack of a shallow décollement in the CT area might be due to several possible reasons, such as: stratigraphic variability. The shallow décollement identified in the PT is likely associated with weak, marly formations, as suggested by data from the Tamara well. The presence of similar weak lithologies (GS group) is also indicated by the Bice well, located in the foreland basin in front of Tamara. However, if these lithologies thin out,

pinch out, or are absent in the CT sector, the development of a comparable shallow detachment would have been less likely. Unfortunately, due to the lack of well data in the CT frontal section, we cannot confirm the lateral variation or presence of these lithologies. mechanical conditions and Syntectonic Sedimentation influence: greater syntectonic loading and sedimentation in the PT area may have helped the development of a shallow detachment, while the CT area may have experienced less sedimentary loading or faster uplift, which could have prevented a similar detachment from forming. Since these remain hypotheses without direct evidence, we have limited our discussion to observations and interpretations based on our available data. Further studies will be needed to explore these possibilities in detail.

9. **First referee comment: Line 683**: "Line 535: "6.2 Seismotectonic implications". In this section, the authors discuss their findings in relation to all previous work published on the 2022 seismic sequence and listed in the "State of the art" section (Line 175), except for the work by Maesano et al. (2023). Is there a specific reason for this omission? I believe a more comprehensive discussion could be beneficial for the readers.

**Reply:** Thank you for this observation. In this section we focused our discussion on studies that specifically address the location and relocation of seismicity and their correlation with our interpreted geological structures in the studied area. The paper by Maesano et al. (2023) is indeed a valuable contribution, but it is more focused on the structural geology of the area rather than on the seismicity relocation or correlation. we mentioned the summary in section 3, and have referred to this study several times throughout other parts of the manuscript. which is why we did not include it in this particular part of the discussion.

10. **First referee comment: Line 760:** "Lines 582-583"This study highlights a possible minor role of the Cornelia thrust system during the 2022 earthquakes than previously thought due to a more limited extent to the NW". This sentence in the conclusion seems at odds with the Seismotectonic implications paragraph, where there is no explicit statement in this direction. Are the authors referring to a particular previous study? Some context might be missing, making this sentence not fully supported by the preceding discussion."

**Reply:** Thank you very much for your thoughtful comment. In this study, we introduced two main thrust faults within the 2022 seismicity area, with the intention of presenting the most possible causative structures for the recent events, rather than focusing on interpretations of specific previous studies. As the context and references to earlier works were already discussed in Section 3, we did not repeat them in the conclusion. We have rephrased the sentence in the conclusion to improve clarity and ensure consistency with the discussion provided in the Seismotectonic Implications section.

**Changes:** Line 760-768: "This study highlights the Cornelia thrust system having a limited extent toward the NW. In addition, the spatial distribution of overall seismicity

possibly related to the CT is scarce and cannot be easily linked with it. Although based on its geological, structural and geometrical characteristics, the CT thrust system cannot be completely excluded as a seismogenic source, in the present study the analysis and the integration of the relocated hypocentres together with the new geological insights suggests that the PT, or a possible deeper easternmost structure, would be a better candidate to be associated with the mainshocks. On the other hand, the relay zone between PT and CT is more coherent with the second main event"

**Reviewer 2**

Dear Reviewer 2,

Thank you for your thoughtful and constructive comments. Your suggestions helped us clarify and improve several parts of the manuscript. We addressed all your points, and when we could not make a change, we provided a clear explanation in our responses.

Second referee comment: Line 17: "absence of outcrops or inaccessibility or difficult
access to them? Maybe the outcrops are there but we cannon access them, isn't it?"
Reply: Thank you for your observation. We accept your suggestion and have revised it
accordingly.

Changes: Line17: "Studying the subsurface geology in offshore areas is a complex task,

as it is impossible or very challenging directly accessing....."

Second referee comment: Line 21: "Compare with and to discuss about?"
 Reply: Thank you for your comment. We have revised the sentence accordingly.
 Changes: Line21: "for building a reliable geological model to be compared with and discussed...."

3. **Second referee comment: Line 36**: "Global? Maybe more than global seismic activity it would be to determine their seismic potential? Association of seismic activity to geological structures?"

**Reply:** Thank you for your observation. We have modified the sentence in line with your suggestion.

Changes: Line36: "for determining their seismic potential and understanding the association between seismic activity and geological structures...."

4. Second referee comment: Line 39: "blind fault may be"

**Reply:** Thank you for your observation. We have modified the sentence in line with your suggestion.

Changes: Line39: Sentence removed

5. **Second referee comment**: Line 39: "What do you mean by "strong"? Could you provide a magnitude that may characterize a "strong" earthquake? Mw6, Mw7...?"

**Reply:** Thank you for your observation. We have modified the sentence in line with your comment.

**Changes:** Line39: " strong earthquakes (≥ Mw 6.0; United States Geological Survey, n.d.) "

6. **Second referee comment:** Line 40: "trigger?"

**Reply:** Thank you for your observation. We have modified the sentence in line with your comment.

Changes: Line 40: " and triggering underwater landslides and tsunamis..."

7. **Second referee comment:** Line 45:"The high-resolution depends on your experiment "parameters.

**Reply**: We have modified the sentence in line with your suggestion.

Changes: Line45: " to provide detailed images..."

8. **Second referee comment**: Line 48: "units?"

**Reply:** Thank you for your suggestion. We have modified the sentence.

**Changes:** Line47-48: " These data are suitable for identifying faults' geometry, kinematics, hierarchy and dynamics as well as the overall subsurface geological setting and the position of different lithologies"

9. **Second referee comment**: Line 48: "Necessary?"

**Reply:** Thank you for your comment, As you mentioned it wasn't necessary sentence and we removed it.

Changes: Line 48: sentence removed

10. **Second referee comment:** Line 55: "Maybe you could consider to include some focal mechanisms on the area in Figure 1?"

**Reply:** Thank you for your suggestion. We have incorporated some of the available focal mechanisms. However, to our knowledge, the focal mechanism of the second event on 9 November 2022 is not reported in the literature (e.g., Costanzo, 2025) nor is it included in the INGV catalogue at https://terremoti.ingv.it/tdmt.

**Changes:** The updated Figure 1 now includes more focal mechanisms.

11. **Second referee comment:** Line 57: "Where is the offshore Marche region? For a non Italian reader this is not ovious."

**Reply:** Thank you for your detailed observation. We have added a brief clarifying sentence (in parentheses).

Changes: Line 57: "Marche region (Central Italy, along the Adriatic Sea coast)"

12. **Second referee comment:** Line 60: "This is the same as the Marche region? If yes, why you change the name?"

**Reply:** The Periadriatic region broadly refers to the area surrounding the Adriatic Sea, encompassing both coastal and offshore zones. It includes parts of Italy on the western side, as well as regions in the Balkans such as Slovenia, Croatia, Montenegro, and Albania on the eastern side. In Italian geological context, the term generally denotes the Adriatic foreland adjacent to the Apennines, particularly along the western margin

of the Adriatic Sea. Since the Marche region is part of this broader Periadriatic area, we initially included the term. However, to prevent any confusion for readers unfamiliar with this terminology, we have simplified the sentence and removed the term "Periadriatic." We appreciate your insight, which helped us improve the clarity of our manuscript.

Changes: Line 60-65: "This active contraction is testified by historical seismicity .... "

13. **Second referee comment**: Line 65: "Refer to any figure?"

**Reply:** Thank you for your suggestion; we have added a reference to Figure 1.

**Changes:** Line 66: "The subsurface offshore thrust faults and related folds in the study area (Fig. 1)"

14. **Second referee comment:** Line 80: "On the legend you indicate that you are plotting earthquakes above 2.0."

**Reply:** Thank you for your detailed observation. The legend has been corrected, and the revised figure has been replaced in Fig. 1.

Changes: please see legend Fig.1, Page4.

15. Second referee comment: Line 95: "units?"

**Reply:** We have modified the sentence.

Changes: Line 92:-93: "these subsurface geological units..."

16. **Second referee comment:** Line 95: "Confusing. What do you want to explain?"

**Reply:** We appreciate your observation regarding the clarity of the original sentence. we have revised the text to more clearly convey the importance of subsurface geology in both identifying active fault segments and understanding the lithological controls on seismicity. We also want to emphasize the role of geological structures in shaping seismic velocity models, which are fundamental for the more accurate recognition of the earthquake location. We hope this revised version addresses your concern and improves the overall clarity of the manuscript.

**Changes:** Line94-95: "In addition, the spatial distribution of these subsurface geological units also affects the configuration of the seismic velocity models, which are critical for achieving more accurate earthquake location solutions (Latorre et al., 2016)."

17. Second referee comment: Line 98: "Senigallia?"

**Reply:** Thank you for your comment. We have specified that Pesaro and Fano are the main cities closest to the mainshocks. Previous studies have referred to these events as the Pesaro–Fano sequence, and we follow this naming in our study to avoid confusion and ensure clarity that we are discussing the same seismic sequence.

**Changes:** No Change.

18. **Second referee comment:** Line 99: "In Figure 1 you just show one focal mechanism. Could you include the second one?"

**Reply:** Thank you for your comment. To our knowledge, the focal mechanism of the second event is not available in the literature (e.g., Costanzo, 2025 – Fig.1, p.3) and is also not included in the INGV catalog (https://terremoti.ingv.it/tdmt). According to Pezzo et al. (2023), no moment tensor solution was computed for this event due to phase overlap and interference with the mainshock. We have corrected the sentence to clarify that only the focal mechanism of the main earthquake is available. Additionally, we have added the available focal mechanisms of other historical seismic events in this area to Figure 1 for better context.

Changes: Please see figure 1, page4.

19. **Second referee comment**: Line 112: "Too strong sentence. There are no other possible approaches to tackle this problem?

**Reply**: Thank you for your feedback. We agree that the original sentence may sound too strong. We have revised it to acknowledge that while this workflow is highly effective and essential for building a reliable geological model.

**Changes**: Line 112-114: "This workflow is essential to shed light onto the subsurface geological settings of the area that can be compared and integrated with seismicity..."

20. Second referee comment: Line 113: "?"

**Reply:** Thank you for your observation. We have now completed the previously incomplete sentence and have revised the paragraph accordingly.

**Changes**: Line 115-117: "The joint use of seismic reflection profiles, calibrated with borehole stratigraphy, provides the necessary framework to mitigate these limitations and improve the accuracy of the geological models."

21. **Second referee comment:** Line 137: "In caption of figures 1 and 2 the boreholes are referred as Pesaro Mare 04 and W1 (not split). Be consistent."

**Reply:** Thank you for your comment. We have corrected the borehole name within the text.

**Changes:** Line 137: "Pesaro Mare 04 and W1 boreholes drilled in the study area" Fig.2 Caption: Line2: "derived from two representative boreholes (Pesaro Mare 04 and W1, location in Fig. 1)"

22. **Second referee comment:** Line 160: "Rewrite. Follow the geological chronology. Could you also provide some time reference? Calabrian (0.78-1.8 Ma), Upper Pleistocene, Pliocene."

**Reply:** Thank you for your helpful suggestion. We have carefully rewritten this section to follow a clear geological chronology.

**Changes:** Line 160-168: "The NA is characterized by a high sedimentation rate that evolved throughout the Pliocene and Pleistocene, reflecting changes in the depositional environment and regional subsidence. During the Pliocene (5.33–2.58 Ma),

sedimentation rates were estimated at 1–2 mm/year in both the Po Plain area and the NA Sea (Ghielmi et al.,2010, 2013; Amadori et al., 2020; Maesano et al., 2023). In the Po Plain area, these rates increased to over 2.5 mm/year during the Calabrian stage (1.8–0.78 Ma) with measured values ranging from  $2.83 \pm 0.19$  mm/year to  $2.14 \pm 0.21$  mm/year (Maesano & D'Ambrogi, 2016). However, sedimentation rates progressively decreased throughout the Middle (0.78–0.126 Ma) and Upper Pleistocene (0.126–0.0117 Ma), reaching a minimum of  $0.39 \pm 0.05$  mm/year in the last 0.45 Myr. This decrease reflects the transition to continental deposition and a general reduction of accommodation space in the basin, while also recording the effect of ongoing regional subsidence during the Pleistocene (Maesano & D'Ambrogi, 2016)."

23. **Second referee comment:** Line 171: "Could you include the bathymetry in Figure 1?" **Reply:** Thank you for your comment. We have updated Figure 1 with bathymetric contours. Initially, we didn't add it because the image was already crowded.

Changes: Please see figure 1, page4.

24. Second referee comment: Line 180: "using?:"

Reply: Thank you for your suggestion. We have corrected the sentence

Changes: Line 180: "imaged by" changed to "Imaged using"

25. Second referee comment: Line 193: "have"

Reply: Thank you for your comment. We have corrected it.

26. **Second referee comment: Line 196:** "You have mentioned this Earthquake previously (lines 84-88) but this is the first time you named the earthquake. I suggest to do it also the first time you describe it. Also in the legend or map, and in the caption of Figure 1 you may have to include the name of the earthquake"

**Reply:** Thank you for your helpful suggestions. As you suggested, we added the name of the earthquake sequence to the introduction, where we first introduced the seismic sequence. We also updated the caption of Fig. 1.

Changes: Please see figure 1 caption, Page 4.

27. **Second referee comment:** Line 196: 1/1/1900 12:00:00 AM "This section must be reorganized: 1. Description of the main shock and the aftershock series. 2. Description of the different proposed fault sources for the EQ."

**Reply: 1)** Thank you for your helpful suggestion. The description of the seismicity sequence was initially in the introduction (Lines 98-100). As you suggested, we expanded this section, providing a more detailed description of the Fano-Pesaro earthquake sequence, which includes the foreshock, followed by the mainshock and aftershock series. This description is now at the beginning of the "State of the Art" section. The discussion of the proposed fault sources follows this contextual overview. **Reply: 2)** Thank you for your comment. We have already summarized the findings of

**Reply: 2)** Thank you for your comment. We have already summarized the findings of each relevant study (Lines 75–210), including those that directly confirm the source of

the seismicity (e.g., Maesano et al., 2023), where such information was available or explicitly stated by the authors. We reorganized this section for more clarification.

**Changes:** Line 197-255: "The Fano-Pesaro earthquake sequence began on November 9, 2022, with a Mw 5.5 mainshock. One minute later, a Mw 5.2 earthquake occurred approximately 8 km to the south-southeast of the mainshock. Before this, only one smaller event (ML 2.8) was recorded roughly two months before the mainshock, and no foreshocks immediately preceded the sequence. This abrupt activation caused notable damage along the central Adriatic coastline, drawing significant attention to the area's complex tectonic structures....."

28. **Second referee comment:** Line 213: "Figure S1 corresponds to Figure 5 without interpretation and there is no reference to CTS. It would be necessary to identify CTS in soe figure, maybe Figure 1?"

**Reply:** Thank you for your valuable comment and detailed observations. We have removed the incorrect reference to the image.

**Changes:** Please see figure 1, page 4.

29. **Second referee comment:** Line 221: "seismic network?"

**Reply:** Thank you for your comment regarding the term "INGV monitoring system." In our manuscript, we used this term exactly as it was presented in the original paper by Pezzo et al. (2023), which refers to the seismic network and GNSS monitoring infrastructure operated by INGV.

Change: No change

30. **Second referee comment:** Line 225: "? Rewrite"

**Reply:** Thank you for your suggestion. We have rewritten the sentence to make our meaning clearer and easier for readers to understand.

**Change:** Line 225-227: "Using the HypoDD relocation method, they refined the mainshock's position, revealing it to be 4.4 km farther south and at a deeper depth of 8 km than previously reported in the INGV catalogue."

31. **Second referee comment:** Line 232: "Identify in some figure."

**Reply:** Thank you for your valuable suggestion regarding the inclusion of the thrust mentioned (T1 splay) in the state-of-the-art section. The only suitable place to include this structure would be Figure 1, which already contains a significant amount of information. We felt that adding another structural element could overcrowd the figure and reduce its readability and clarity for the reader. Furthermore, to show the T1 splay accurately, we would also need to include the main T1 thrust, which overlaps with other interpreted thrusts already presented in Figure 1. This would further complicate the visual layout. Additionally, we would like to note that a detailed comparison of thrust geometries is not the main focus of this study. For these reasons, we were unable to add

the additional thrust directly into the figure. We greatly appreciate your thoughtful input and hope this explanation is acceptable.

Change: No change

- 32. **General Changes:** Changes: Line 242: A brief summary of the most recent published study on our study area has been added to complete this section.
- 33. **Second referee comment**: Line256: "Could you specify the number of profiles you have work with? Also it would be interesting of showing all your used seismic dataset in a figure, maybe Figure 1?"

**Reply:** Thank you for your helpful comment. In this study, we primarily focused on the seismic profiles shown in Figure 1, but also integrated interpretation from an additional 8 seismic lines, both public and industry-provided provided available within the study area. These supplementary profiles were used to better constrain the thrust faults and their associated anticline structures, and include both inline and crossline orientations. To avoid overcrowding, we initially did not include these additional lines in Figure 1 in order to ensure readability. We represented only the key interpreted profiles in a darker colour. Following your valuable comment, we have updated Figure 1 to include the additional public seismic profiles, shown in light grey. This approach allows us to highlight the main dataset used in our analysis, while still indicating the availability of further data for more detailed investigations, most of which are accessible through the ViDEPI website.

**Changes:** Line 257: ", comprising 8 crosslines and 3 tielines, " and Please see Figure 1, page 4.

34. **Second referee comment:** Line 260: "Which ones?"

**Reply:** Thank you for your comment. We would like to clarify that the seismic profiles and Wells provided by the company or derived from the ViDEPI website are already described and shown in Table 1 and Figure 1 of the manuscript. In Table 1, we have highlighted the unpublished data provided by the Company by marking the names of the corresponding wells and seismic profiles with an uppercase star, as also mentioned in the table caption. Also in Figure 1, we differentiated the datasets using color coding:

• Green indicates public datasets sourced from the ViDEPI website • Blue indicates confidential data provided directly by the company (for both seismic profiles and well data) This color scheme is also explained in the figure legend. We hope this clarifies the way the ViDEPI data have been presented and distinguished within the manuscript.

Change: No change.

35. **Second referee comment:** Line 262: "Which ones?"

**Reply:** Thank you for your comment. AS we mentioned in the previous comment's response both unpublished dataset provided by the company and public source dataset already highlighted in the Table 1 and Fig.1 and also in their caption and legend.

Change: No change.

36. **Second referee comment:** Line 269: "Could you specify?"

**Reply:** Thank you for your suggestion. In the revised manuscript, we have mentioned some of the key software tools used in the workflow to make it clearer. We chose not to include the full list, as it would be too long and likely outside the main interest of the readers.

**Change:** Line 271-274: "This workflow incorporates several specialized software tools: e.g. QGIS for managing geospatial data, a MATLAB code (based on the methodology of Sopher, 2018) for digitizing seismic profiles, Petrel and Move platforms for seismic interpretation and velocity modelling, and OpendTect software for conventional data processing. Further details on the processing workflow are illustrated in Figures S1 and S2 of the supplementary materials."

37. **Second referee comment:** Line 277: "Which one of the fourth mentioned in Table 1? Could you show the sonic log in Figure 3 or make Figure 4 as Figure 3 and refer to the Figure"

**Reply:** Thank you for your feedback. We also have added a sonic log graph, including the depth and log range, to Figure 4, in order to improve clarity and provide context.

Change: Please see Figure.4. Page 21.

38. Second referee comment: Line 278: "Reference"

**Reply:** Thank you for your valuable comment. References add it to the text.

**Change**: Line 278: "(Bally et al., 1986; Maesano et al., 2013, 2023; Montone & Mariucci, 2023)"

39. **Second referee comment:** Line 280: "How have you done this conversion? I have never done this, but I think that it is not trivial."

Reply: Thank you for your comment. We performed the time-to-depth conversion using two approaches. One was based on interval velocities derived from sonic log data between interpreted horizons. Manual adjustments were also made in areas with intersecting 3D surfaces to ensure consistency and accuracy in the velocity model, which was then applied to horizons, faults, and seismic profiles in the Petrel software. The second approach was a 2D velocity model created using Move software. This model used both interval velocities obtained from sonic log data and constant velocity values sourced from literature for deeper intervals for which we have no registered sonic log data. We then applied this model to the time-to-depth conversion of our interpreted horizons, faults, and seismic profiles. We then correlated the accuracy of the depth-converted horizons with the depth of the top of the same or similar formation to evaluate the accuracy of our velocity model, which shows a good correlation in 3D with well data. To provide further clarification, we have added an extra table showing the actual velocity model used for this study in the supplementary materials.

**Change:** No change.

40. **Second referee comment:** Line 291: "Necessary?"

**Reply:** Thank you for your observation. We believe that specifying the direction, as it helps the reader follow the spatial arrangement of the wells and better understand the variation in depth and thickness of the formations across the study area.

Change: No change

41. **Second referee comment:** Line 294: "Repetitive"

**Reply:** Thank you for your detail review. We replace the repeated verb with the other appropriate verb.

Change: Line 294: "to support a deeper understanding .....

42. **Second referee comment:** Line 313: "Maybe you could name the units that bound the erosional boundary? Ex: ... Messinian (between units PG and GS),... This would help the reader to identify the unconformities. You use unconformities in the figure legend, be consistent and maybe name it in the text as erosive unconformity instead of erosional boundary?"

**Reply:** Thank you for your valuable suggestion. As recommended, we have replaced the term 'erosional boundary' with 'erosional unconformity'. To better introduce the erosional unconformities, we have aligned them with your advice and added bounded units in the revised manuscript.

**Changes:** Line 313-345: "The W1 well intersects the easternmost segment of the seismic profile S1 and contains 160 m of Lower Cretaceous carbonates. Within this well, three erosional unconformities have been identified, corresponding to the top Messinian (between the PG and GS units) ...."

43. **Second referee comment:** Line 323: "What do you mean by levels? Units, formations,...?"

**Reply:** Thank you for your valuable comment. We have revised the text to provide a clearer explanation, highlighting that the two unconformities at 1370 m and 1912 m are both located within the PG unit but occur at different depths and represent distinct boundaries. Due to faulting, this unit is duplicated in the Tamara 01 well, resulting in a structural repetition in which the older portion of the PG unit is positioned above the younger one. Consequently, the base of the Lower Pliocene appears above its top.

**Change:** Line 320-325: "Erosional unconformities have been identified at several stratigraphic levels: within Lower Pleistocene (between the As and PG units) at a depth of 1217 m, top of the Upper Pliocene (within the PG unit), At 1912 meters, marking both the top of the Upper Pliocene PG unit and the base of the Lower Pliocene of another PG unit. And the last erosional unconformity occurs top of the Upper Messinian (between GS and SCH units) at a depth of 3015 m."

44. **Second referee comment:** Line 325: "unconformity? paraconformity?"

**Reply:** Thank you for your comment. The only available data are scanned images of the wells, which include the stratigraphic column, lithology, and log data. In these well images, unconformities are indicated using standard symbology; however, their specific types (e.g., disconformity, angular unconformity) are not explicitly labelled. Although some inclination marks and angles are shown within parts of the stratigraphic column, they are not continuous or sufficient to confidently determine the relative orientation of the beds across the reported unconformities. For this reason, to avoid introducing potentially inaccurate interpretations, we preferred to refer to them more generally as unconformities in the manuscript.

Change: No change.

45. **Second referee comment:** Line 338: "How do you know this? Previous information? Then refer. Your study? Then explain better. What is the basis (lithology, fossil,...) for this correlation?"

**Reply:** Thank you for the insightful comment. The paragraph has been revised to better explain the reasoning behind the correlation of the undefined dolomitized carbonates with the Dolomie di Castelmanfrino Formation. While we kept the description general in the main text. Relevant references have also been added to support this correlation.

Change: Line 333-339: "However, considering their stratigraphic position beneath the Marne a Fucoidi and Scaglia Calcarea formations, and their overall characteristics as shallow-water platform carbonates, this unit is interpreted in this study as equivalent to the Dolomie di Castelmanfrino (DCM) Formation. This correlation is consistent with similar successions identified in other Apennine sectors, such as the Montagna dei Fiori area, where comparable dolomitized Jurassic sequences have been described and attributed to the DCM (Ronchi et al., 2003; Murgia et al., 2004; Bencini and Martinuzzi, 2012)."

46. **Second referee comment:** Line 343: "Just Early Cretaceous? 6/13/2025 5:43:00 PM In figure 3 you use Upper/Middle/Lower instead of Early/Late. Be consistent."

**Reply 1:** "Just Early Cretaceous?" Thank you for the observation. The available data are limited. The well intersects only the uppermost part of the repeated stratigraphic interval, with approximately 30 meters drilled into it. As a result, our interpretation is constrained to the Early Cretaceous portion that is directly sampled. Due to the lack of deeper penetration, no definitive information is available for older formations potentially involved in the repetition caused by thrust displacement. We have clarified this limitation in the revised text.

**Reply 2:** "In figure 3 you use Upper/Middle/Lower instead of Early/Late. Be consistent." Thank you for your detailed revision. We revised it.

**Change:** Line 343-345: "Additionally, a tectonic boundary is reported approximately 30 m from the bottom of the well. It is interpreted as a thrust splay, whose offset results in the repetition of the Lower Cretaceous succession. The well was drilled only into the upper part of this repeated interval (Lower Cretaceous succession), and no data are available for the deeper successions (Fig.3)."

**Change:** Please see figure 3 on page 17 (The boundary of the Porto Garibaldi Fm in the Chronostratigraphical legend has been modified.

47. **Second referee comment:** Line 349: "I cannot distinguish between the VIDEPI and ENI wells."

**Reply:** Thank you for the comment, we would like to clarify that each well is already labelled at the top, differentiated by distinct colours in the location map, and we also introduced them in the local map legend.

change: No Change

48. **Second referee comment:** Line 349: "In the figure you write VIDEPI."

Reply: Thank you for your detail review. We corrected it.

Change: Fig. 3 has been updated, page 18.

49. **Second referee comment:** Line 356: "Rewrite. In well W1 seems to be complete, or at least you do not indicate any unconformity/disconformity in the Plio-Pleistocene succession."

**Reply:** Thank you for your comment. We have revised the paragraph accordingly to provide better clarification.

**Change:** Line 355-359: "Within southeastern wells (Pesaro Mare 04 and Cornelia 01), the Pliocene-Pleistocene sedimentary sequence, is frequently incomplete. Notably, in the Pesaro Mare 04, situated on a structural high, the Pliocene succession (PG unit) is entirely absent, with a direct transition from Miocene deposits to Quaternary sediments. Conversely, in the basin areas, such as the W1 well, a more complete sequence spanning the lower to upper Pliocene is preserved."

50. **Second referee comment:** Line 360: "Identify the wells."

**Reply:** Thank you for your valuable comment: We added the wells name.

**Changes**: Line 361: "the northwestern (W1) and southeastern (Cornelia 01) wells of the study area."

51. **Second referee comment**: Line 365:This unit is not shown in any well. Clarify

**Reply:** Thank you for your valuable comment. We have clarified in the revised text that the Triassic succession (Burano Formation) is not penetrated by the studied wells, and its presence and lithological characteristics are instead inferred from the nearby Alessandra 1 well and published data from adjacent areas (e.g., Bally, 1986; Carminati et al., 2013). Additionally, we have added a label indicating the location of the Alessandra 1 well in Figure 1. Please note that this well lies slightly outside the map extent and is therefore referenced schematically.

**Change:** See Figure 1. page 4.

**Change:** Line 367-371: "is not intercepted by the studied wells. However, its presence is inferred from nearby Alessandra 01 well (See location in Fig.1), located slightly to the east, which represents the deepest borehole drilled in this region, and is almost

entirely composed of dolostone facies reported by Bally (1986), Carminati et al.(2013) and Scisciani & Esestime (2017). "

52. **Second referee comment:** Line 367: Could you show the location of this well in a figure, for example Figure 1?

**Reply:** Thank you for valuable suggestion. we have added a label indicating the location of the Alessandra 1 well in Figure 1. Please note that this well lies slightly outside the map extent and is therefore referenced schematically.

Change: See Figure 1. page 4.

53. **Second referee comment:** Line 373: "This is just indicated for well Pesaro Mare 04. There are no nodules in Cornelia 01. Then, this is not a general characteristic."

**Reply:** Thank you for your valuable comment and detailed review. We agree with your observation and have revised the sentence and removed the also unnecessary words

**Change:** Line 373: this sentence have been removed: "again characterized by typical nodular structures."

Reply: Thank you for your valuable suggestion. As recommended, we have revised and reorganized the section by moving the geological concepts into Section 5.1, where they are more appropriately discussed. We have also added references to Figures 5 and 6 to clearly support the description of the seismic units.

Changes: Line 383-418: "SU1 corresponds to the Holocene-Upper Pleistocene turbiditic deposits (AS unit). The uppermost part of SU1 is characterised by continuous to semi-discontinuous, horizontal and parallel reflections, with low to high amplitudes. While the lower part displays continuous to semi-continuous, eastward-dipping reflections, with medium to high amplitudes (Fig.4 and Supplementary Table S1). The total thickness of SU1 gradually increases north-eastwards, ranging from  $\sim 0.2$  s to 1.5 s TWT across the study area (Fig.4). This thickening pattern is consistently observed in all interpreted seismic profiles (Figs. 5 and 6). ......."

54. **Second referee comment:** Line 376: "After reading this section I think that you may made some changes in sections 5.1 and 5.2. Use section 5.1 to describe the gelogy corresponding to each unit identified in the wells summarizing the main characteristics observed in all the wells. Also about some chronostratigraphic explanation for some of the boundaries that you are mentioning in this section. Then, in section 5.2 just describe the seismic stratigraphy, but not the geology of the units. There is more geological description in section 5.2 that in section 5.1, where is supposed that is the well geology what you may be explaining. There is need to refer to the seismic profiles in Figures 5 and 6. I assume that some unit descriptions are based on the profiles shown on these figures."

**Reply:** Thank you for your comment. We have modified the text based on your suggestion and rearranged it following your valuable advice, separating the stratigraphy and geological sections into 5.1 and 5.2, respectively.

Changes: Lines 288–435: Sections 5.1 and 5.2 have been fully revised.

55. **Second referee comment:** Line 376:"I would suggest to refer to reflectors that bound units as horizons and name them as H1, H2,... and briefly describe them in a paragraph when starting the section. Then describe each SU. 6/16/2025 12:12:00 AM For the reader it would be very useful to identify each unit (SU1, SU2;...) and horizons (H1, H2,...) in the seismic profiles in Figures 5 and 6."

**Reply:** Thank you for your comment. Regarding the horizons, we prefer to retain the names of the reflectors that are more common in previous studies of this area, as they are more familiar to readers. We believe that adding additional horizons and labels could create confusion in the paper.

Change: No Change

56. **Second referee comment:** Line 384: This is the first time that you mention that AS corresponds to a turbiditic succession. Maybe you could could rewrite all the sentence starting from the description of unit AS, then mentioning that are turbidites and that you identify as the seismostrarigraphic unit S1 or something similar.

**Reply:** Thank you for your valuable and detailed observation. As suggested, we have added a description of each stratigraphic unit in Sections 5.1 and 5.2. In line with your previous recommendation, we focused Section 5.2 specifically on the seismic stratigraphy, to ensure a clearer and more structured presentation.

**Change:** Line 299- Section 5.1: "AS unit (Holocene–Upper Pleistocene): A siliciclastic marine turbidite system composed of fine sandstones, shaly sandstones, and interbedding of shale and silty shale."

57. **Second referee comment:** Line 385: "Two comments here: 6/17/2025 6:39:00 PM 1. Maybe you could identify the subunits in profiles S3 or S4? I am not familiar with the area, but following your description I may be able to indicate the boundaries between some of the units. 2. Are you using these subunits in the analysis of the data? If not, consider to simplify this description to a more general description. It is not necessary to describe something that you will not be using and could be confusing for the reader. This is just a suggestion."

**Reply:** Thank you for your valuable comment. Following your advice, we decided to remove the subunits because as you mentioned we did not use them in our interpretations or in other sections, so they were not useful. We also add the updated Figures 5 and 6 by adding zoom views to better present the seismic units.

**Change:** See Fig.5 and Fig.6 on pages 26 and 29, respectively. Change: Line 84-85: This sentence has been removed: "AS. SU1 consists of four distinct seismic sub-units (SU1 a, b, c, d), each one characterized by a different seismic signature (see supplementary Table S1)."

58. **Second referee comment:** Line 389: "The increase in thickness is not shown in Table S1, or at least I cannot see where."

**Reply**: Thank you for your observation. The increase in thickness of SU1 is not visible in Supplementary Table S1, as this table only provides zoomed-in views of each

seismostratigraphic unit without illustrating their lateral extent. However, the thickening trend is clearly observable in Figure 4 at S2 section ,by comparing the thickness on the left (SW) and right (NE) sides of the profile. To improve visibility in Figure 4, we have added vertical bar on the right side of each presented section to better highlight the thickness variations. Also it is consistently visible across the interpreted seismic profiles in Figures 5 and 6. We have revised the main text to clarify this point and added references to Figures 5 and 6, following your valuable suggestion.

**Change:** Line 395: "The thickness of this unit gradually increases from  $\sim 0.2$  s in the SW to 0.6 s in the NE. Similar to SU1, this Thickening pattern is consistently observed in all interpreted seismic profiles (Figs 4, 5 and 6). "Change: See Figs 4, 5 and 6 on pages 21, 26 and 29, respectively.

59. **Second referee comment:** Line 390:Check TWT in Figure 4 since the units are "ms" and I think it must be "s"? Your seismic profiles are in "s".

Reply: We appreciate your detailed observation. The unit has been corrected

Change: See Figure 4, page 21.

60. **Second referee comment:** Line 392: "Or lower Pleistocene as in Figure 3. Be consistent. Check all the manuscript for similar differences between the text and figures and modify where necessary."

**reply:** Thank you for pointing this out. All necessary modifications have been made for consistency.

**Change:** Line 392-395: "corresponds to the lower Pleistocene turbiditic deposits (PG unit) and is separated from SU1 by a toplap unconformity, dated to Top Gelasian (older than 1.8 Ma; Fig. 4). "

61. **Second referee comment:** Line 392: "In Figure 2 unit PG corresponds to the Pliocene, but here you mention that the boundary between this unit and unit AS is the top of the Gelesian (1.8 Ma). Then, describe the Gelesian sediments within the upper part of PG. Here there is a problem, the top of the Gelesian is not the boundary between the Pleistocene and the Pliocene. This boundary corresponds to the base of the Gelesian (2.58 Ma). This needs to be clarified. PG is Pliocene or there is part Pleistocene? Modify text and figures accordingly."

Reply: Thanks to your detail observation, the well data was collected and interpreted between 1969 and 1989, so they used the older version of the geochronology timescale (prior to 2009). In that version, the Pliocene was divided into three epochs: Lower, Middle, and Upper Pliocene. According to the new classification (since 2009), what was previously called the Upper Pliocene now corresponds to the Lower Pleistocene (Gelasian), while the Middle Pliocene corresponds to the Upper Pliocene (Piacenzian). Based on the well stratigraphy data, the Porto Garibaldi formations were assigned ages from Early Pliocene to Middle and Upper Pliocene, according to the old classification. We updated this in the text, but We initially missed to update the legend. We have fixed this in Figure 3.

**Change:** See Figure 3, on page 17.

62. **Second referee comment**: Line 392: "This unconformity is not shown in Figure 4. Maybe it is seen in some profile in Figure 5 or 6? Could you point to a position where this is seen?"

**Reply:** Thank you for your comment. We have added a focused image in Figures 5 and 6 to better illustrate the unconformity and clearly show its position within the seismic profiles.

**Change:** See focus image al in Figure 5, on page 26.

63. **Second referee comment:** Line 395: "Where this is shown?"

**Reply:** Thank you for your observation. The increase in thickness of SU2 similar to SU1 is observable in Figure 4 at S2 section ,by comparing the thickness on the left (SW) and right (NE) sides of the profile. To improve visibility in Figure 4, we have added vertical bar on the right side of each presented section to better highlight the thickness variations. Also it is consistently visible across the interpreted seismic profiles in Figures 5 and 6. We have revised the main text to clarify this point and added references to Figures 5 and 6, following your valuable suggestion.

**Change:** See Figs 4, 5 and 6 on pages 21, 26 and 29, respectively.

64. **Second referee comment:** Line 396: "Same comment as for SU1, necessary to define the subunits if they are not used or shown anywhere?"

**Reply**: Thank you for your valuable comment. Following your advices, we decided to remove the subunits because as you mentioned we did not use them in our interpretations or in other sections, so they were not useful. We also add the updated Figures 5 and 6 by adding zoom views to better present the seismic units.

**Change:** See Figs 5 and 6 on pages 26 and 29, respectively.

**Change**: Line 396-397: "Within SU2, we identified two sub-units (SU2 a, b), each one characterized by distinct seismic signature." This sentence has been removed.

65. **Second referee comment:** Line 398: "apparent or NE? Your profiles trend SW-NE." **Reply:** hank you very much for your helpful observation. You are absolutely right—the original wording referred to an apparent direction, but to avoid confusion, we have now clarified the text by specifying the actual NE-oriented trend of the seismic profiles. We appreciate your attention to detail, which helped us improve the clarity of the manuscript.

**Change:** Line 398: "The uppermost part of this unit displays continuous, NE-dipping parallel reflections with medium to high amplitudes"

66. **Second referee comment:** Line 400: "It seems that PG unit was deposited between the Pliocene and lower Pleistocene, but that is not what is shown in Figure 3. Could you clarify this. It is confusing."

**Reply:** Thank you for your detailed observation. As mentioned in our response to the first comment regarding the age of the PG unit, we had already updated this in the main

text, but we initially missed updating the figure legend. This has now been corrected in Figure 3.

**Change:** See Figure 3 on page 17, Chronological legend.

67. **Second referee comment**: Line 405:According to Figure 4 the maximum TWT thickness would be around 1 ms as shown in profile S2. It is not clear where you determine the thickness of the different SU units. You may need to clarify this. This thickness is estimated along the different profiles? If yes, then refer to the figures and the position of your measurement along the profiles.

**Reply:** Thank you for your observation. In Section 2 of Figure 4, the maximum thickness of SU3 is about 0.4 s on the right side of the profile (NE). On the left side (SW), SU3 appears three times due to repetition, along with repeated Top Messinian units (marked with pink lines). Each SU3 layer there is about 0.3 s thick, so the total thickness of this unites along with SU4 appears close to 1 second. In the other sections (Fig. 5 and Fig. 6), especially in S1, S3, S4, and S5, SU3 is either missing or much thinner usually just a few milliseconds mainly over the anticline crests and their limbs. **Change:** No Change, For Clarification See Figs. 4, 5 and 6 on pages 21, 26 and 29 respectively.

68. **Second referee comment**: Line 406: "What about GS? This is the top of your Miocene succession in Figures 3 and 4"

**Reply:** Thank you for your helpful comment and detailed observation. We missed mentioning the GS. We have now corrected this and included it in the revised text.

**Changes:** Line 406: "The SU4 represents the complex Miocene succession and is observed within the GS, SCH and BIS Fms"

69. **Second referee comment:** Line 411: "Where this is shown? According to your Figure 3 that's not the case."

**Reply:** This unit represents the Top Messinian, as previously mentioned. In our interpreted seismic profiles, it appears at a shallower depth in the southwestern part because it is located on the hanging wall. Toward the northeast, it lies deeper as it is in the footwall of the thrust. Thanks to its high frequency and strong reflection, this horizon is easily traceable across most sections.

Change: No change

70. **Second referee comment:** Line 411: "Repetitive. Consider to rewrite with the previous sentence describing the upper part of the unit"

**Reply:** Thank you for your valuable suggestion and detailed review. We have merged the two sentences to avoid repetition.

**Change:** Line 407-412: "This marly group displays continuous, parallel reflections with high amplitude and high dominant frequency in the narrow uppermost part and creates distinct and sharp reflections in the seismic sections. The rest of the unit presents continuous to discontinuous, sub-parallel reflections with medium to high amplitude

(Supplementary Figs S1, S2 and Table S1). This seismic unit progressively deepens from southwest to northeast (Figs 4, 5 and 6). "

**71. Second referee comment: Line 411: "?"**

**Reply:** Thank you for your comment. What we meant is that this unit exhibits a relatively high dominant frequency at top, which contributes to producing sharper and more clearly defined reflections in the seismic sections. We have revised the text for clarity accordingly

Change: Line 407-412: "This marly group displays continuous, parallel reflections with high amplitude and high dominant frequency in the narrow uppermost part and creates distinct and sharp reflections in the seismic sections. The rest of the unit presents continuous to discontinuous, sub-parallel reflections with medium to high amplitude (Supplementary Figs S1, S2 and Table S1). This seismic unit progressively deepens from southwest to northeast (Figs 4, 5 and 6). "

72. **Second referee comment:** Line 414: "According to Figures 3 and 4 there are much more units in this succession."

**Reply**: Thank you for your observation. We limited our recognized stratigraphic seismic units to those that could be validated using the available well data. Unfortunately, for the deeper units, we do not have access to well data to support further subdivision or confirmation.

Change: No Change

73. **Second referee comment:** Line 422: "Usually you use de acronym, be consistent."

**Reply**: Thank you for your detail review. We correct it in the text.

Change: Line 422: "area (down to the Top SCA)."

74. **Second referee comment:** Line 422: "Which ones? These data must be included, at least as supplemanetry material."

**Reply:** Thank you for your valuable input. We have revised the sentence for clarity. What we intended to convey is that we carefully verified the correlation between our studied wells and the depths of the top formations after depth conversion. Following your suggestion, we have removed the reference to "control points" and clarified that we extended the model toward within our study area. The control points, as described in the text, correspond to the correlation between the tops of formations in the wells and the depth-converted interpreted main reflectors shown in Figure 7. These tops are also visible in Figures 5 and 6 as projected wells, with further details provided in Figure 3.Readers can observe that the depth-converted reflectors align reasonably well with the original formation tops reported in the wells, confirming the reliability of our velocity model.

**Change:** Line 422-424: "This workflow was then extended across a tri-dimensional workspace, encompassing later variations driven by all the interpreted horizons and fault surfaces, and correlating with the well data from a broader area"

75. **Second referee comment:** Line 426: "There is again some confusion. Table S2 shows that AS is the Pleistocene unit with velocity 2200 m/s, but then in Figure 4 this velocity corresponds to the upper PG unit. This needs to be clarified to avoid confusion and future misunderstunding. 6/18/2025 11:36:00 AM Also according to your Table 2 The Oligocene to Triassic units may have different velocities but in your Figure 4 it seems that you just consider a constant velocity for the whole succession. That is the case or are you using some velocity gradient? This needs to be stated because the thickness/depth calculated may be larger than the real."

Reply: Thank you for your insightful and constructive comment. In Figure 4, we opted to display simplified average velocities to maintain visual clarity and avoid overcrowding the image with numerous numerical values. To prevent any confusion, we clarified in the caption that the velocities shown are representative averages and do not reflect the full velocity model used in our analysis. The actual interval velocities applied in the depth conversion workflow are reported in the last column of Table S2. These values are based on well-tie analysis from the Tamara well's sonic log and on published sources, as cited in the other columns of the table. To support transparency and reproducibility, we have also included the full original velocity models in tabular format in the supplementary materials. Regarding the Pleistocene velocity assignment, the problem was not with the velocity value attributed to the AS unit, but rather with an outdated chronostratigraphic classification. Specifically, the table had not been updated to reflect the current definition of the Lower Pleistocene, which was previously grouped differently in the well data due to the chronological framework available at the time of preparation.

**Changes:** To address this, we have: Updated Table S2 to reflect the new classification by including the Lower Pleistocene as part of the PG unit, not just AS; Clarified the caption of Table S2: "Note: The final velocity model was constructed using average interval velocities derived from well-tie analysis of the Tamara well's sonic log;..." Clarified the caption of Fig. 4 page 22, Line 432-436: "The displayed values represent averages of the original velocities reported in the last column of Supplementary Table S2, ...."

76. **Second referee comment:** Line 431: "Various comments: 1. Tamara 01 log? What is it showing? Needs units and maybe make it larger. 2. TWT in ms when I think it must be in s. 3. What are the pink bounded wedge-bodies in S2? 4. What are the blue triangles with pink dots in profile S3?"

**Reply1:** Thank you for your helpful suggestion. Based on your previous valuable comment, we have added the Tamara 01 sonic log as a separate subsection to Fig. 4, included the appropriate labels, and specified the unit range to improve clarity.

**Reply2:** We appreciate your detailed observation. The unit has been corrected.

Reply3: Thank you for this comment. The pink-bounded wedge-shaped bodies correspond to top Messinian reflectors, which are repeated due to several imbricated

thrusts. These repetitions are also confirmed by well data, which show three distinct occurrences.

**Reply4:** Thank you for the question. The blue triangles indicate normal polarity (black = peak), as already mentioned in the text. For clarity, we have now also added a label and included it in the legend.

Change: See Fig.4 on page 21.

77. **Second referee comment**: Line 446: "Refer to the figures."

**Reply:** Thank you for your comment. The requested reference to the figures has been added in the text.

**Change:** Line 446: "...developed at the hanging walls of SW-dipping thrusts, named Pesaro thrust (PT) and Cornelia thrust (CT) (Figs 5 and 6). "

**Change:** Line 446: "...developed at the hanging walls of SW-dipping thrusts, named Pesaro thrust (PT) and Cornelia thrust (CT) (Figs 5 and 6). "

78. **Second referee comment:** Line 447: "What do you mean? I think that you tie the wells to 2D seismic profiles and then follow the reflectors along the profiles and check or extend your interpretation to other profiles. This is not a 3D correlation. But I could be wrong."

**Reply:** Thank you for your insightful comment. You are correct that the well ties are performed on 2D seismic profiles and that the interpretation follows reflectors along these profiles. However, by "three-dimensional 6/18/2025 10:43:00 PM correlation" we mean the integration and correlation of key reflections across multiple intersecting 2D profiles, allowing us to build a coherent spatial understanding of the seismic-stratigraphic units in three dimensions, even though the data are from 2D lines. We add the word of Pseudo for clarifying that it is not real 3D correlation.

**Change**: Line 447: "The whole interpretation of seismic profiles has been realized by using a pseudo-tri dimensional correlation .... "

79. **Second referee comment:** Line 452: "Consider to delete since you refer to this figure at the end of the sentence."

Reply: Reply: Thank you for the suggestion. We removed the extra reference

**Change**: Line 452: "The seismic profile S1 is dominated by the ...." referring to figure 5 has been removed."

80. **Second referee comment:** Line 452:0-12 is 12 km not 13 km. Can you clarify?

**Reply:** Thank you for pointing this out. The value of 12 km is correct, it was a typing mistake, We corrected it within text.

**Change:** Line 452: "The seismic profile S1 is dominated by the east-verging PA, characterized by a long wavelength of  $\sim 12$  km (0–12 km distance, Fig. 5a)."

81. **Second referee comment:** Line 453: "I would suggest to identify the different seismic units (SU1, SU2,...) in the interpreted profile to help the reader to identify them easily in the figures."

**Reply**: Thank you for the helpful suggestion. We have added labels of the seismic units (SU1, SU2, etc.) to Figures 5 and 6. We also add some zoomed images for showing the detail of the interpretation

Change: See Figures 5 and 6 on pages 25 and 28, respectively.

82. **Second referee comment**: Line 454: "(blue, green and pink reflectors in Figure 5)" **Reply:** Thank you for your observation. Yes, you are correct, there is also a green reflector in between. However, in the text, we mentioned only the blue and pink reflectors because we were specifically referring to the top Jurassic, marked by the blue line, and the top Messinian, marked by the pink lines in Figure 5 and 6. For this reason, we introduced only these two reflectors in the sentence.

Change: No Change

83. **Second referee comment**: Line 457: "Could you mentioned the units deformed by these minor folds?"

**Reply:** Thank you for the suggestion. We have added a brief description of these minor folds to the text.

Change: Line 456-460: "The more internal minor folds, closer to the crest zone of PA, affect a thicker succession, ranging from the Jurassic to the Pliocene, and are traceable through key reflectors such as the Top Jurassic (blue colour), Top Lower Cretaceous (dark green colour), Top Oligocene (light green colour), and Top Messinian (pink colour). In contrast, the more external folds deform shallower successions, mainly involving the Messinian units and the overlying Pliocene sediments (Fig. 5a, 5b). "

84. **Second referee comment:** Line 462: "Refer to the figure."

**Reply:** Thank you for the comment. A reference to the figure has been added.

**Change:** Line 462: "...folds the Plio-Pleistocene unconformity reflector (dark yellow colour; Fig.5a)."

85. **Second referee comment:** Line 463: "Refer to the figure."

**Reply:** Thank you for the comment. We have referred to the figure as suggested. **Change:** Line 463: "...fTop Carbonates (Oligocene) reflector (light green colour; Fig. 5a, b)."

86. Second referee comment: Line: 464: "6 km? From 7 km (PA axis) to 13 km?"

**Reply**: Thank you for your comment. We carefully rechecked our interpreted section S1 in both the previous hard copy (black and white) version and the newly added digitized seismic profile. The wavelength of the syncline is approximately 4 km, located between 9 and 13 km along the profile.

Change: No change. Please see Figure 5 on page 25.

87. **Second referee comment:** Line 466: "Pleistocene?"

**Reply:** Thank you for your detailed review and helpful comments. As you mentioned, 'Pleistocene' is correct, and we have corrected it in the text.

Change: Line 465: ".by sub-horizontal reflectors interpreted as lower Pleistocene sediments"

88. Second referee comment: Line 468: "Pleistocene?"

**Reply:** Thank you for your detailed review and helpful comments. As you mentioned, 'Pleistocene' is correct, and we have corrected it in the text.

**Change:** Line 467: ".a clear increase in the apparent dip angle and thickness of the Pleistocene succession"

89. **Second referee comment:** Line 468: reflectors?

**Reply:** You are correct, thank you for your detailed observation. As you suggested, since we are referring to sedimentary successions, it is more appropriate to use 'reflectors' rather than 'reflections'. We have made this correction in the text.

**Change:** The text has been fully modified and updated between lines 440-559. Example: "The PA geometry is traceable from  $\sim 0.2$  s down to  $\sim 2.5$  s, and it is particularly evident following the interpreted Top Jurassic to Top Messinian reflectors ......"

90. **Second referee comment:** Line 468: "shown?"

**Reply:** Thank you for your comment. We are not sure we fully understand your point, but if you mean referring to the figure, we have added the reference accordingly.

91. **Second referee comment:** Line 472: "Above you say 9-12 km. Be consistent."

**Reply:** Thank you for pointing this out. We have corrected.

**Change:** Line 472: "with the minor folds described above on the PA forelimb ( $\sim$ 9–12 km distance range)."

92. **Second referee comment:** Line 472: "What do refer to? You may consider to rephrase."

**Reply:** Thank you for your observation. We have revised and rephrased the sentence to improve clarity.

**Change:** Line 472-475: "Such backthrusts are imbricated from the innermost secondary ramp of the PT, whereas the forethrusts, builds up the shorter wavelength TS, are all imbricated from and detached along the shallower flat of the PT"

93. **Second referee comment**: Line 486: "Confusing. Explain main thrust and then other thrusts branch to it and displaced units."

**Reply:** Thank you for your helpful comment. In this section, we focused specifically on the TS structure, as the main thrust (PT) and the overlying PA structure are not imaged in this profile (Section 2). For this reason, we could not start the description from the main thrust. To improve clarity, we have added one sentence at the beginning of the paragraph to say that the main thrust PT and full geometry of PA is not imaged in this section.

**Change:** Line 483-486: "The seismic profile S2 (Fig. 5b) covers only a small portion of the PA forelimb, including the shallowest flat of the PT, and provides a clearer picture of the TS imbricates; the PT footwall reflectors, previously described in S1, are also visible here and appear more continuous and better traceable (Fig. 5b, inset b1)."

94. **Second referee comment:** Line 489: Consider to delete since you start the sentence with the localization of the area. Otherwise, you may keep this part of the sentence and delete the beginning.

**Reply:** Thank you for the suggestion and detail review. We have removed the unnecessary part of the sentence to improve clarity.

**Change:** Line 489: Revised version: "The imbricates are detached on the shallow PT flat (~ 2.5 s TWT), which produces a further repetition of the Top Messinian reflector (pink colour). In the south-western part of S2, again, the minor folds driven by the backthrusts mapped in S1 are observable.."

95. **Second referee comment**: Line 506: "Some comments or suggestions on the figure: 1. I would recommend to identify the different SU (SU1, SU2,...) in the seismic profiles. 2. The figure caption needs to be understood by itself. Then define all the acronyms that appear in the figure (PA, TS,...) in the caption. 3. What is lamda-s? And S5?"

**Reply:** Thank you for your helpful comments and suggestions. We have updated the figure accordingly.

Change: Please see Fig.5 on page 25. Caption modified Line 505-510: "Interpretation of S1 and S2 seismic profiles. a) S1, the northernmost section in the study area, crosses the left hinge zone of the PA and reveals variations in its structural style. This profile intersects with seismic section S5 at a  $\sim$  6.1 km. The geometry of the PA is evident by using the Top Messinian, Top Oligocene and Top Lower Cretaceous reflectors. The section also shows the shallow-seated TS developed laterally to the South-Eastern termination of the PA. b) S2 shows the enhanced comprehension of the TS's underlying structure. Four imbricated thrust zones of the PA forelimb and the repetition of Messinian- middle- lower Pliocene successions are observable (uninterpreted images provided in supplementary materials, Fig.S1). Insets a1 and b1 show detailed interpretations.  $\lambda$ s= wavelength of the small structure; PT= Pesaro Thrust; PA= Pesaro Anticline; CA= Cornelia Anticline; TS= Tamara Structure; SU= Seismic Stratigraphy unit"

**Reply:** Thank you for your comment. In this context, the geometry refers to the overall structural configuration of the PA.

Change: No Change

97. **Second referee comment:** Line 521: 6/22/2025 11:39:00 AM Wouldn't you agree that the PA may extend from 0 to 7 km? From 7 km to the NE there is a local sincline related to the TS and backthrusts?

**Reply:** Thank you for your insightful comment and detailed observation. Based on our observations and interpretation, the backthrust developed within the forelimb of the PA and is structurally linked to the main PA structure. For this reason, we consider it part of the PA, which we interpret to extend approximately from 0 to 9.7 9.8 km (rounded to 10 km). The synclinal structure appears further to the NE, starting beyond ~10 km.

Change: No Change

98. **Second referee comment:** Line 523: "profiles S1 and S2 (Figure 5)"

**Reply:** Thank you for your comment. We have added the reference to Fig. 5 within the text as suggested.

99. **Second referee comment:** Line 523: "profiles S1 and S2 (Figure 5)"

**Reply:** Thank you for your comment. We have added the reference to Fig. 5 within the text as suggested.

100. **Second referee comment:** Line 523: "profiles S1 and S2 (Figure 5)"

**Reply:** Thank you for your comment. We have added the reference to Fig. 5 within the text as suggested.

**Change:** Line 523: "A few smaller antiformal structures located at the PA forelimb, as already observed in S1 and S2 (Fig. 5), are again interpreted as being driven by small backthrusts."

101. **Second referee comment:** Line 523:"Following similar criteria to set the length of TS than in Figure 5, TS may extend from 6-7 km up to 13 km, for a total length of 6-7 km, similar or larger than in S1. Then, could you justify your afirmation."

**Reply:** Thank you for your comment. As mentioned in our reply to the previous comment, based on our interpretation, the interval from 0 to ~10 km is part of the main PA structure. The backthrust displaces reflectors within the hanging wall of the anticline and is therefore structurally linked to the PA. 6/22/2025 12:12:00 PM In contrast, the TS is interpreted as the result of thrust imbrication derived from the shallower flat-ramp system of the PA. It forms an antiformal structure composed of several closely spaced, small-wavelength antiforms, reflecting the displacement of shallower reflectors from the Top Fucoidi (limited to an internal imbricate), as well as the Oligocene to Plio–Pleistocene successions. Based on our observations, the TS extends from approximately 10 km to a maximum of ~13 km.

Change: No Change

102. **Second referee comment**: Line 525:"Not quite sure about this. To me both flans dip almost similar. I cannot identify a clear assimmetry."

**Reply:** Thank you for your comment. Based on our observations and interpretation, we consider the backlimb of the CA to be traceable from  $\sim$ 4 to 14 km, with the crest zone between 14 and 16 km. In contrast, the forelimb extends 6/22/2025 11:27:00 PM from  $\sim$ 16 to 20 km, including the area affected by the secondary thrust and displaced reflectors, which we interpret as part of the forelimb. The vertical relief in both limbs is similar ( $\sim$ 1.5 to  $\sim$ 2.5 s), but this change occurs over  $\sim$ 10 km in the backlimb and only  $\sim$ 4 km in the forelimb. This difference in dip supports our interpretation of the CA as an asymmetrical fold

Change: No change

103. **Second referee comment:** Line 531: "Use the acronym."

Reply: Thank you for your comment. We have corrected it

Change: Line 531: "The CT is interpreted to comprise also a small synthetic thrust, developed at its footwall, which produces a further repetition of the SCA Group and Top Messinian reflectors."

104. **Second referee comment:** Line 532: "Could you point to the location as you have done for previous units or structures (TWT and distance)?"

**Reply**: Thank you for your comment. We have specified the location of the structure by indicating both TWT and distance

**Change**: Line 533: "we observe a shallower and thicker package of growth strata, interpreted to comprise Pliocene to Quaternary deposits, traceable from approximately 0.5 s to ~2 s TWT between ~16 and ~23 km distance (Fig. 6a)."

105. **Second referee comment**: Line 536:"In this profile the asymmetry is much more clear. The dip in the NW flank seems higher than in the SE."

**Reply:** Thank you for your comment. We agree and have the same observation for this section (S4): the NE flank of the Cornelia Anticline is steeper than the SW flank (again confirming its asymmetrical geometry). In contrast to S3, the Cornelia structure in S4 is more clearly expressed, as it is not overridden by internal anticlines (e.g., Pesaro). Moreover, this section crosses the crest zone of the Cornelia, and this asymmetry is more evident.

Change: No change

106. **Second referee comment:** Line 536: "I do not agree, as mentioned in a previous comment."

**Reply:** Thank you for your comment, we have provided our detailed observations and supporting evidence in response to the previously mentioned comment.

107. **Second referee comment**: Line 537: "3.0?"

**Reply:** Thank you for your comment. Based on our observations, folding is traceable from approximately 3.5 s, starting at the Top Jurassic reflectors (blue line) within the hanging wall.

Change: No change

**108. **Second referee comment:** Line 537: "12-13?"**

**Reply**: Thank you for your comment. We consider the entire structure, including the splays that repeat the reflectors in the frontal section of the anticline. However, as you suggest, the main structure, excluding these splays, extends approximately up to 13 km. We have revised the text.

**Change:** Line 537: "The CA is represented by an asymmetric NE-verging anticline (as also observed in S3), extending from  $\sim$ 0.5 s to  $\sim$ 3.5 s, and is prominently displayed between 3 km and 13 km distance.

109. **Second referee comment**: Line 538: "Something is missing, maybe you need to state that these units are folded?"

**Reply:** Thank you for your comment. We revised the sentence for better clarity and to address your suggestion.

**Change:** Line 537-539: "This anticline is defined by the folded reflectors from the Top Triassic up to the Pliocene-Pleistocene unconformity (purple to yellow colours), situated within the hanging wall of the underlying CT."

110. **Second referee comment:** Line 540: "Comparing S3 and S4 it is surprising the change in offset along the fault corresponding to the Top Lower Pliocene. In S4 is much more higher than in S3. I do not mean this is wrong, it could be possible, but at least you may mention something here about this difference. You could measure the offsets for the different units, which could be useful to state the time of the activity. Another thing is that the thickness of the unit between the Messinian top and the lower Pliocene top (having the name of the corresponding SU here may avoid large sentences) is larger in S3 than in S4. Again this do not have to be wrong, but may deserve some explanation. I have checked the uninterpreted profile, but it has some interpretation on is and it is difficult to compare facies in the image; however, I would not discard that the top of the lower Pliocene could be located at a lower depth. Anyway, that is your interpretation and you may defend it."

**Reply:** Thank you for your insightful comment. You are correct that there is a significant difference in the offset of the Top Lower Pliocene between sections S3 and S4. However, it is important to note that there is a considerable distance of around 10 km between these two sections. In section S4, we are situated at the crest of the Cornelia anticline, which exhibits the maximum height and displacement, whereas in section S3, we are in the northwestern hinge zone of the Cornelia anticline, near its termination and characterized by a lower elevation. Additionally, the interpretation by the other studies e.g. Maesano et al., 2023, Fig. 3 aligns with our observations and also indicates these

differences in displacement offsets. For better clarity, we have revised and updated the corresponding paragraph.

Change: Line 540-543: "The latter, like in S3, offsets the Meso-Cenozoic succession up to the Top lower Pliocene reflector (orange colour); however, in this section, located at the crest zone of the CA (Fig.1), the structure exhibits the maximum height compared to S3, which lies in the northwestern hinge zone, resulting in a larger displacement of the Top Lower Pliocene reflectors (from 1.25 to 2.0 s in S4 versus 2.1–2.2 s in S3; Fig. 6a, b) "

111. **Second referee comment:** Line 543: "In figure is Top Oligocene, why do you not use the same identification? Mixing names is confusing."

**Reply:** Thank you for your valuable suggestion. We have revised the text.

**Change:** Line 543-544: "A small synthetic thrust is again observed in the footwall of the CT, which results in the repetition of the Top Oligocene (Top SCA Group) and Top Messinian reflectors over 9 to 14 km, extending to ~2.7 s."

112. **Second referee comment:** Line 545: "Which unit is that? The units above the Plio-Pleistocene unconformity or above the Top lower Pleistocene or the top of lower Pliocene? If is the thickness above top of Lower Pliocene there is not so much difference. In S3 it is around 2.3s and in S4 around 2.5 s. But again this depends on your interpretation."

**Reply:** Thank you for your comment. We have updated the paragraph to better clarify our observation. Additionally, we included the corresponding seismic unit name (SU1) both in the text and in the figure to improve clarity.

**Change:** Line 544-547: ". In the northeastern part of this section, the interpreted Pliocene to Quaternary deposits (SU1 and SU2), close to the end of the forelimb of the CA ( $\sim$ 13 km distance), are thicker than at the similar location in S3, with the top of the Pliocene reflector located at  $\sim$ 2.5 s in S4 versus  $\sim$ 2.1 s in S3.", Please see Fig.6 on page 28.

113. **Second referee comment:** Line 548:"These thrust faults could not be related to PT? According to your interpretation of profile S5 it could be. Can you say something?" **Reply:** Thank you for your thoughtful observation regarding the possible connection between these minor thrusts and the Pesaro Thrust system. While we cannot completely exclude this possibility, the minor thrusts observed in the northeastern part of S4 intersect around 2.7 s, but in the overlapping portion of the two profiles, these faults are not clearly traceable and appear deeper than the identified PT. Moreover, we observe the presence of another fold structure (Elga structure) in this area, suggesting that these minor thrusts may belong to the northeastern hinge zone of that anticline. However, this structure lies near the southern termination of the Pesaro anticline and is not well constrained in our dataset. As it falls outside the main scope of our study, we prefer not to expand on this point to avoid unsupported speculation.

Change: No change

114. **Second referee comment:** Line 556:"Comments on Figure 5 may be valid also for this figure."

**Reply:** Thank you for your helpful comments and suggestions. We have updated Figure 6, similar to Figure 5, accordingly.

Change: Please see Fig.6 on page 28.

115. **Second referee comment:** Line 565: "I have some problems to correlate some reflectors between the cross profiles with the tie profile S5. On Figure 6 I have pasted the crossing zones of S1, S3 and S4 on S5 and this not accurate and fast correlations show important discrepancies between interpretations. Authors must check and correct this."

**Reply**: Thank you for your detailed review and valuable observations. The mismatch in the intersection points was due to a slight location shift during the graphical preparation of the seismic profile figures for the paper. We have now corrected the intersection locations and updated the figures accordingly, including adjustments related to the revised seismic interpretations. We appreciate your attention to detail, which has helped us improve the clarity and accuracy of our work.

Change: Please see Figures 5 and 6 on pages 25 and 28, respectively.

116. **Second referee comment**: Line 594: "General comments: 1. The extension from NW to SE of the TS mapped in Figure 7d is not in agreement with the presented data."

Reply 1: We appreciate your observation and fully acknowledge your point. In Figure 7d, we drew the TS structure only up to the area where our data and interpretations provide sufficient confidence. In section S3, it is challenging to trace the reflections and imbricated thrusts clearly enough to confirm the continuation of the TS structure. Consequently, in Figure 6a, we did not interpret these reflections within the middle structure (TS). Furthermore, in the geological section provided (Fig. 7b), there is still uncertainty about the detailed structure. To avoid overinterpretation, we preferred to limit the mapped extension of TS to where we are relatively sure. However, Thanks to your detail observation, we recognize that indicating the possible southeastern continuation of TS with a dashed line in Figure 7d could help convey the uncertainty without compromising the accuracy of our interpretation. We revised the figure accordingly to clarify this point.

**Change:** Please see the Figure 7d on page 33 (location map)

117. **Second referee comment:** Line 594: "2. It is not clear the basis to end CT/CA on PT/PA or below it. Could be the structure mor parallel to PT/PA and localized more to the NE and not imaged in S1 and S2?"

**Reply 2:** Thank you for your helpful comment. In this study, we mapped the structures based on the data available within the presented profiles. Our ongoing work with additional seismic lines suggests a possible northwestward extension of the PT and a

shorter southeastward continuation of the CA. According to the available public data (the VIDEPI dataset), the Cornelia anticline appears to terminate before reaching sections S1 and S2. However, since this is beyond the scope of the current dataset, we preferred not to include the full extent of these structures. Following your suggestion, we have now added dashed lines in the local map to indicate their potential continuities. **Change:** Please see Figure 7d on page 33 (location map).

118. Second referee comment: Line 594: "3. In this sections the authors use another terminology to refer to the geological units. This confusing. Why the authors do not use the same units definition for the whole manuscript? For readers not familiar with the area it generates confussion."

Reply 3: Thank you for your valuable advice and observation. We have revised the text to ensure consistent use of geological unit terminology throughout the manuscript. Change: Line 600- 659 (Section 6-1): "shallow-seated thrusts, represented by closely spaced, short-wavelength structures of Tamara structure (TS), affect a limited portion of the Upper Cretaceous and younger sequences, including the Oligocene, the Miocene and the overlying turbidite deposits. Toward the front of the TS, these imbricated shallow-seated thrusts impact even shallower and younger sequences, involving only the Miocene and overlying turbidite deposits (Fig. 5a, 5b). ......"

119. **Second referee comment:** Line 604: "Figure 7a just shows the PT/PA and TS structures, not the CA/CT."

**Reply:** Thank you for your valuable observation. We agree with your comment and have now included Figure 7b in the referenced figure to complement Figure 7a

**Change:** Line 604: "clear view of the spatial relationship among the aforementioned structures (Fig. 7a, 7b)."

120. **Second referee comment**: Line 605: "At least? It could extend more than mapped?"

**Reply:** Thank you for pointing that out. Based on the mapped geometry, it is reasonable to consider that the structure may extend beyond the currently interpreted area. In the text, we used "at least" to indicate that the mapped extent represents a minimum estimate based on the current dataset. The possibility of a larger extent will be further investigated by expanding the interpretation to a wider area, using available data from the ViDEPI project and existing literature.

**Change:** Line 604-606: "PA is characterized by an NW-SE (along-strike) extent of at least  $\sim 30$  km long and is  $\sim 12$  km wide (along-dip, SW-NE direction, see profile S1 in Figs. 5a,7a, 7d). "

121. **Second referee comment**: Line 605: "What is the difference between what you consider width and wavelength? Could you clarify?"

**Reply:** Thank you for your observation. The theoretical definition of wavelength refers to the distance between two successive peaks or troughs of a fold. In section S1, we

only observe the PA structure, which also includes the TS related anticlinal form in front of it. Since this section does not display a complete fold train or multiple adjacent structures that would allow for a clear measurement of wavelength, we have deliberately avoided using the term "wavelength." Instead, we use the term width to refer to the extension of the structure along the dip direction.

Change: No change

122. **Second referee comment:** Line 607: "Could you specify the range? Between xx km and yy km."

**Reply:** Thank you very much for your valuable suggestion. We have modified the text accordingly and added the specified distance range.

**Change**: Line 607-608: "Section S1 shows PT as relatively flat in the shallow portion, within the  $\sim$ 6-11 km distance range, transitioning to a steeper ramp toward the southwest."

123. **Second referee comment**: Line 608: "? Rewrite"

**Reply:** Thank you for your suggestion. We have modified and clarified the sentence accordingly.

**Change**: Line 607-608: "Section S1 shows PT as relatively flat in the shallow portion, within the  $\sim$ 6-11 km distance range, transitioning to a steeper ramp toward the southwest."

124. **Second referee comment:** Line 609: "What is the acoustic basement? It has not been described anywhere previously."

**Reply:** Thank you for your observation. The text has been modified to clarify the meaning of the acoustic basement, which in this context refers to the top of the Permo-Triassic sequence or the base of the Triassic evaporites.

**Change:** Line 608-610: "It is reasonable to image the PT lower décollement lying at around 9 km depth, possibly on top of the acoustic basement (base of the Triassic evaporites or top of the Permo-Triassic sequence; Mirabella et al., 2008; Barchi et al., 2012; Porreca et al., 2018)."

125. **Second referee comment:** Line 613: "I assume that the length is the mapped extension of the TS. In my opinion and according to the authors interpretation, the extension of TS must be longer. They interpreted the structure in profile S3, and could correspond to the faults that are interpreted in S4, but in the map in figure 7d the TS structure does not reach these profiles. Accordingly, TS could be as large as PT/PA. Authors may explain why TS is not extended towards SE. "

**Reply:** Thank you for your remark. As previously explained in the beginning of this section, we limited the mapped extent of the TS structure in Figure 7d to areas where seismic data and interpretations are reliable. To address your correct suggestion, we have updated the figure by adding a dashed line to indicate the possible southeastward

continuation of TS, and we revised the text for better clarification about the approximate length of the TS based on the new updates in the figure.

**Change:** Please see figure 7d (location map) on page 33 and Line 614-615: "forming TS, characterized by a length of  $\sim$ 20 km, a width of 7 km and a wavelength  $\lambda$ s of  $\sim$ 1.1 km (Figs. 5a, 5b, 7a, 7d)."

126. **Second referee comment:** Line 617:"According to your crossections in the not deformed/uplifted/tectonized area, the Messinian unit is at a depth between 4.5-5 km."

**Reply:** Thank you for your valuable comment. In the text, the 3.5 km depth refers specifically to the décollement of the upper flat segment of the PT within the Messinian. However, to improve clarity based on your suggestion, we have revised the sentence to explicitly mention the full range of detachment depths, from the deeper detachment within the Jurassic succession at around 5 km depth to the upper décollement within the Top Messinian successions at around 3.5 km depth.

**Change:** Line 614-617: "All these structures, including both fore-verging and back-verging thrusts, are associated with the upper, shallower semi-flat segment of the PT, which is detached at multiple stratigraphic levels. These detachments range from  $\sim 5$  km depth within the Jurassic succession in the hanging wall of PT, to a sub-parallel décollement within the Top Messinian (marly group), at roughly 3.5 km depth."

**127. **Second referee comment:** Line 624: "thrust sheets?"**

**Reply:** Thank you for your comment. By referring to the "three slices," we mean the repetition of the Neogene sequences caused by structural stacking and duplication, rather than distinct thrust sheets.

Change: No Change

**128. **Second referee comment:** Line 629: "See my comment about this for PA."**

**Reply:** Thank you for your comment. Although not as extensively as the Pesaro structure, the structure likely extends beyond what is currently mapped. It is estimated that its maximum length reaches approximately 22 km along strike. We have revised the text to clarify that this length is based on the current dataset, and that the structure could potentially extend further to the southwest.

**Change:** Line 628-630: ". It results in a NW-SE striking,  $\sim$  20 km long (possibly extending just a few km further toward the SE) and  $\sim$  12 km wide anticline (profile S4 in Figs. 6b, 7b, 7c) with a wavelength  $\lambda l$  of  $\sim$  11 km (Figs. 6a, 7b)."

**129. **Second referee comment:** Line 633:"? Something missing?"**

**Reply:** The text has been revised.

**Change**: Line 632-637: "These structures are characterized by lower syn-tectonic sedimentation. Conversely, the observed structural wavelength values are smaller than those observed in the Po Plain, where higher syn-tectonic sedimentation contributes to

even larger structural wavelengths, with  $\lambda_l$  ranging from 15.8 to 33 km and  $\lambda_s$  from 4.5 to 8.2 km (Massoli et al., 2006). This observation is confirmed by the relationship described by Massoli et al. (2006), where variations in structural wavelength are linked to both the depth of the active décollement and the thickness of syn-tectonic sediments."

130. **Second referee comment**: Line 640: "Geeral comments: 1. Mapped extension of the TS is not based on the data that authors are presenting. 2. Authors use a third different terminology to refer to the geological units. Confusing."

**Reply:** Thank you for your helpful comment. We have updated the location map by refining the mapped extent of the PA—Ca and TS. The legend has also been revised for better consistency. A single terminology is now used in the legend, with corresponding formation names added in parentheses where appropriate.

**Change**: Please see the Figure 7 on page 33.

131. **Second referee comment:** Line 641: Respectively to what? Maybe to the different corssections? Could you plot the buffers in Figure 7d? I am not sure if it make sense to extend the buffer up to 10 km. Usually you may use a buffer to consider the uncertainty on the earthquake location. What is the uncertainty on the horizontal location of the earthquake catalog?

**Reply:** Thank you for your comment. The caption of Figure 7 has been revised accordingly. We have also updated the buffer zone, now applying a consistent 5 km buffer for all seismic sections. This adjustment better highlights the spatial distribution of seismicity relative to the different studied profiles. Regarding the horizontal location uncertainty in the INGV earthquake catalog, it is typically around 1–2 km onshore, where station coverage is denser. Offshore, the uncertainty is possibly higher due to sparser seismic station distribution and greater depth constraints. This estimate is based on our review of several published studies and methods (e.g., Pezzo et al., 2022; An et al., 2021), as we are not specialists in earthquake location analysis. A more detailed evaluation of location uncertainty falls outside the scope of this study.

Change: Line 641: Please see Figure on page 33.

132. **Second referee comment:** Line 649: "Your data/interpretation suggest that they may have a common detachment, but this is not shown clearly."

**Reply:** Thank you for your valuable comment. We have revised the text to more cautiously suggest the possibility of a shared detachment, as the listric geometry of both faults supports this interpretation.

**Change:** Line 647-650: "From the analysis of the profiles S3 and S4 (Fig. 7b, 7c), considering both the geometry of the anticlines and the trajectories of the thrusts, a shared deep decollement level can be inferred at approximately 9 km depth, consistent with results reported in nearby areas (e.g., Pauselli et al, 2006, or Lavecchia et al., 2004)."

133. **Second referee comment:** Line 657: "?"

**Reply:** The text has been revised for improved clarity.

**Change:** Line 658-659: "a doubling of the Upper Cretaceous carbonate succession (SCA group) over approximately 3 km between  $\sim$ 16–19 km distance in section S3 and about 4 km between  $\sim$  8–12 km in section S4 (Fig. 7b, 7c)."

134. **Second referee comment:** Line 660: "In this paragraph you could also discuss about the timing of the deformation or the thrust in sequence propagation towards the basin. Could the CA be a more recent structure than PA?"

**Reply**: Thank you very much for your thoughtful suggestion. We agree that considering the timing of deformation and the sequence of thrust propagation could offer important insights. Based on the available structural geometries, it is possible that the external Cornelia Anticline (CA) represents a younger structure compared to the internal Anticline (PA). In particular, the apparent overstepping of the PT onto the northwestern hinge zone of CA in Section S3 may suggest a pre- to syn-chronological relationship between PA and CA. However, since we don't have enough data to confirm the timing, we preferred to don't discuss about this possibility in this section.

Change: No change

135. **Second referee comment**: Line 668: "That is not what you show in figure 7d, where TS is not extended to at least profile S3."

**Reply:** As mentioned in our response to your valuable comment at the beginning of this section, Figure 7d and the text have been updated to show the possible extension of the TS structure up to section S3.

Change: Please see Figure 7d on page 33.

136. **Second referee comment**: Line 671: "I may not agree with this afirmation, at least it deserves some discussion. What about the thrust faults interpreted to the SW of profile S4? They may be perfectly correlated to TS, just comparing the structure that is quite similar to the structure observed in S1 (S2) and S3."

Reply: Thank you for your valuable comment. Based on our observations and data, TS is a shallow-seated décollement located in the hanging wall of the Pesaro thrust (PT). It is a more external and shallower structure than PT, but intermediate between PT and Cornelia in section S3, appearing in the overlap zone between PA and CA. The minor converging thrusts seen in sections S4 and at its intersection with S5 affect younger stratigraphy (post Upper Messinian) and intersect S5 at about 2.75 s TWT, within the footwall succession of PT. This suggests they relate to deeper thrusts rather than TS. These thrusts are also located in the overstepping zone of Pesaro on the NW hinge of the Elga anticline, which is outside the focus of this study. Due to data resolution, these minor thrusts could not be clearly traced in S5, but we remain confident they are linked to deeper structures rather than TS. We have clarified this point in the revised manuscript.

Change: Line 671-672: "Our investigations indicate that the TS represents the deformed wedge of the frontal part of the PA structure, formed within the hanging wall

zone of Pesaro, thus it cannot be considered originated by a single deep-seated structure such as PT or CT and neither a northwest-eastward continuation of the Cornelia thrust.

137. **Second referee comment**: Line 675: "To what zone are you referring respect to your working area? Where are located this external sectors?"

**Reply:** Thank you for your comment. The text has been modified accordingly. The "external sectors" refer to the Adriatic Arc Front, as already mentioned in parentheses along with the relevant references (e.g., Bice thrust, Lavecchia et

**Change**: Line 675: "In slightly external sectors, with respect to our studied area, evidence of deep thrusts has been reported from ...."

138. **Second referee comment:** Line 684: "A couple of comments: 1. Most of the seismicity is below 7-8 km. 2. What is the vertical error on the localization of the seismicity? 3. Could you include a figure with two graphs: a) EQ magnitude vs time of the seismic series; and b) histogram of cummulative number of earthquakes with depth. That would help to understand better the seismic sequence and the depth distribution of earthquakes.

**Reply:** Thank you for your valuable suggestions. We addressed your comments as follows: 1) Seismicity Depth: According to different seismicity catalogues (including the original INGV dataset and more recent relocated versions), most earthquakes occurred between 6–10 km depth. For better clarity, we added a new figure illustrating this distribution as suggested. 2) Vertical Error in Localization: We have revised the text to include the vertical and horizontal errors in the earthquake localization. 3) Additional Figure: As per your helpful suggestion, we have added a new figure that includes: a) the spatial distribution of the mainshock and aftershock locations, shown together with our interpreted structural features; and b) a histogram illustrating the cumulative number of earthquakes by depth. This figure provides a clearer overview of the spatial relationship between seismicity and structures, as well as the depth distribution of the events.

Changes for comment number 2: Line 705: "accuracy (±1 km depth error for the mainshock reported in the INGV catalogue), " and Line 712-713: "The study also reports error estimations, with maximum values ranging from 0.8 to 3.6 km in all three directions."

Changes for comment numbers 1 and 3: Please see figure 8 on page 38.

139. **Second referee comment:** Line 685: "The studied earthquake series is not reported in this catalog. According to the webpage it spands up to 2020."

**Reply:** Thank you for pointing this out. You are correct, the cited catalog refers to historical seismic events and was mistakenly included here. We have removed the reference accordingly.

**Change**: Line 686: "seismicity recorded during the Fano-Pesaro 2022 sequence (terremoti.ingv.it)" Reply: Thank you for your helpful comment. We have modified the

text by adding and referring again to the estimated dimensions of the PT and CT structures, the type of empirical relationships used (Wells & Coppersmith, 1994; Leonard, 2014), and the resulting possible magnitudes. As this section only aims to indicate their seismogenic potential, we did not include the full equations or detailed calculations, since a more in-depth analysis of this aspect is beyond the scope of the present study.

140. **Second referee comment:** Line 688: "Could you include the focal mechanisms in Figure 7d?"

**Reply:** Thank you for your helpful suggestion. The focal mechanisms are already shown in Figure 1; however, we agree that including them in Figure 7d improves clarity and spatial correlation with our interpreted structural framework.

Change: Please see figure 7d on page 33.

141. **Second referee comment**: Line 690: "impirical relationships"

Reply: The text has been revised

**Change:** Line 690: "using the Empirical relationships for the thrust faults (e.g. Wells and ....."

142. **Second referee comment:** Line 692: "Where is this shown? What are the magnitude ranges that you get from the fault length? There are different empirical relationships, which ones are you using? MW vs Length for thrust/reverse faults? What fault length are you considering in this calculatins?"

**Reply:** Thank you for your helpful comment. We have modified the text by adding and referring again to the estimated dimensions of the PT and CT structures, the type of empirical relationships used (Wells & Coppersmith, 1994; Leonard, 2014), and the resulting possible magnitudes. As this section only aims to indicate their seismogenic potential, we did not include the full equations or detailed calculations, since a more indepth analysis of this aspect is beyond the scope of the present study.

**Change:** Line 692-694: "According to the findings of this analysis, the estimated sizes of the PT ( $\sim$ 360 km2) and CT ( $\sim$ 240 km2) suggest that they are capable of generating seismic events with magnitudes of up to Mw 6.8 and Mw 6.5, respectively. "

143. **Second referee comment:** Line 701: "What do you want to state? It is not clear to me. What is extending to NE?"

**Reply:** Thank you for your helpful comment. We have removed the unnecessary directional reference to improve the clarity of the sentence, as the spatial mismatches were already evident in the figures.

**Change:** Line 701: This sentence has been removed: "(extending more to the North-East)"

144. **Second referee comment**: Line 698: "I do not understand what you want to state. Clarify."

**Reply:** Thank you for your helpful comment. We have removed the unnecessary directional reference to improve the clarity of the sentence, as the spatial mismatches were already evident in the figures.

**Change**: Line 701: This sentence has been removed: "the latter being less extended to the North-West"

145. **Second referee comment:** Line 707: "Not necessary to change paragraph. This is the continuation of the previous sentence."

**Reply:** The text has been revised.

146. **Second referee comment**: Line 709: "Could you include a new figure with the faults and the relocation of the main shocks and aftershocks proposed by each author? That may help to follow your description"

**Reply**: The new figure has been added **Change:** Please see Figure 8 on page 33.

147. **Second referee comment**: Line 714:"I assume that the seismicity has been projected perpendicularly on the profiles. Then when the seismicity on the maps and on the profiles is compared is a little bit confusing. You must state how you have projected it. Another possibility maybe if you project the seismicity following the trend of the structures on the different zones it could shown a better correlation between structures and seismicity?"

**Reply:** Thank you again for your valuable suggestion. As mentioned earlier, we initially used a maximum buffer of ~10 km to project the seismicity perpendicularly onto the profiles. Based on your helpful comment, we revised this to a narrower buffer of ~5 km to improve spatial accuracy. For example, this adjustment shows that no seismic events are projected onto section S4, which intersects the crest of the CT, and the second shock is no longer projected onto any of the sections, as this event is located between sections S1 and S3. supporting the limited role of the CT in the recent seismicity. Additionally, to better highlight the relationship between seismicity and the thrust trend, we applied a color scale in Figure 8 to show depth distribution. This clearly indicates that seismicity follows the thrust system, becoming progressively deeper toward the onshore area.

**Change**: Please see Figure 7 and its caption on page 33, and Figure 8 on page 38.

148. **Second referee comment:** Line 719: "What is T1? There is a deeper thrust sheet in the region?"

**Reply:** Thank you for your comment. T1 refers to a deeper thrust fault suggested in previous studies in this area (Lavecchia et al., 2023), but this fault is not clearly imaged in our data. We have clarified this point in the revised text to avoid confusion.

**Change:** Line 719-724: "However, such hypothetical deeper fault is not clearly imaged or visible within available vintage seismic reflection profiles, characterized by a lack of clear reflected signals from deeper reflectors, or just by very weak and poorly

continuous reflective patterns embedded within high level of random noise, typical of legacy profiles (Ercoli et al., 2023)"

149. **Second referee comment:** Line 753:"Two decollements, isn't it?"

**Reply:** Thank you for your comment. We identified at least two main décollement levels in the available dataset: a deeper one at the top of the acoustic basement, and a shallower one within the Miocene succession (frontal section TS). Additionally, we interpreted possible décollement levels within the SCA and FUC (top Oligocene, Top Cretaceous) in other sections within the study area, particularly within the Pesaro sector. To reflect the presence of these structural levels without excluding the possibility of further décollements, we chose to use the more general term "multiple décollements" rather than specifying an exact number.

Change: No change

150. **Second referee comment**: Line 757: "The TS, a series of up to three/four imbricate thrusts..."

**Reply:** Thank you for your suggestion. The text has been revised accordingly.

**Change**: Line 757-762: "The TS, a series of imbricate thrusts, develops along the shallow part of the PT at a depth of 3.5 km, is characterized by...

151. **Second referee comment**: Line 759: "According to your map and profile interpretations I may not agree with this conclusion. Review my comments."

**Reply:** Thank you for your detailed review and valuable observations. already mentioned in response to previous comments, we have revised the text and updated the structural map of the area, including the possible length of the TS. The text in the conclusion has also been updated accordingly.

Change: Line 795: ".....can be followed for ~20 km along strike."

152. **Second referee comment**: Line 764: "Northern Apennines"

**Reply:** Thank you for your comment. As already mentioned in the text (introduction section), "NA" referred to the Northern Adriatic Sea.

Change: No change